# Optimization of Cas9 activity through the addition of cytosine extensions to single-guide RNAs

**Masaki Kawamata** [1] ✉, **Hiroshi I. Suzuki**[2,3,4,5] ✉, **Ryota Kimura**[1] & **Atsushi Suzuki** [1] ✉

The precise regulation of the activity of Cas9 is crucial for safe and efficient editing. Here we show that the genome-editing activity of Cas9 can be constrained by the addition of cytosine stretches to the 5′-end of conventional single-guide RNAs (sgRNAs). Such a 'safeguard sgRNA' strategy, which is compatible with Cas12a and with systems for gene activation and interference via CRISPR (clustered regularly interspaced short palindromic repeats), leads to the length-dependent inhibition of the formation of functional Cas9 complexes. Short cytosine extensions reduced p53 activation and cytotoxicity in human pluripotent stem cells, and enhanced homology-directed repair while maintaining bi-allelic editing. Longer extensions further decreased on-target activity yet improved the specificity and precision of mono-allelic editing. By monitoring indels through a fluorescence-based allele-specific system and computational simulations, we identified optimal windows of Cas9 activity for a number of genome-editing applications, including bi-allelic and mono-allelic editing, and the generation and correction of disease-associated single-nucleotide substitutions via homology-directed repair. The safeguard-sgRNA strategy may improve the safety and applicability of genome editing.

The CRISPR-Cas9 system enables efficient genome editing[1–4]. Yet, in addition to well-known off-target effects, recent studies have documented several prevalent adverse effects of the standard CRISPR-Cas9 system in mammalian cells, including frequent p53 activation, cytotoxicity with severe DNA damage, large on-target genomic deletion and chromosomal rearrangement[5–8]. In human induced pluripotent stem cells (hiPSCs), severe cytotoxicity and cell-cycle arrest are induced by DNA double-strand break (DSB)-mediated p53 activation; thus, it is difficult to obtain knockout and homology-directed repair (HDR) clones[6,7].

Considering the potent activity of the current CRISPR-Cas9 and its frequent adverse effects, controlled inhibition of its activity would be a straightforward and powerful approach to improve its safety. For this purpose, various options (for example, anti-Cas9 proteins, small-molecule inhibitors and oligonucleotides) have been demonstrated to limit Cas9 activity[9–16]. A recent report described that programming Cas9 ribonucleoproteins (RNPs) with photocleavable guide RNAs (gRNAs) (that is, pcRNAs) is useful for the controlled deactivation of Cas9[17]. Although these options are well-characterized with respect to reduced efficiency of genome cleavage and editing, methods for

[1]Division of Organogenesis and Regeneration, Medical Institute of Bioregulation, Kyushu University, Fukuoka, Japan. [2]Division of Molecular Oncology, Center for Neurological Diseases and Cancer, Nagoya University Graduate School of Medicine, Nagoya, Japan. [3]Institute for Glyco-core Research (iGCORE), Nagoya University, Nagoya, Japan. [4]Center for One Medicine Innovative Translational Research, Gifu University Institute for Advanced Study, Gifu, Japan. [5]David H. Koch Institute for Integrative Cancer Research, Massachusetts Institute of Technology, Cambridge, MA, USA. ✉e-mail: kawamata@bioreg.kyushu-u.ac.jp; hisuzuki@med.nagoya-u.ac.jp; suzukicks@bioreg.kyushu-u.ac.jp

optimizing these approaches for practical CRISPR application, as well as the extent to which they would improve safety and applicability, remain unexplored.

Complex editing outcomes of the standard Cas9 system and demands for its various applications have further complicated the availability and optimization of Cas9 inhibition approaches. When sufficiently expressed in cells by means of a widely used plasmid-based system, single-guide RNA (sgRNA)-Cas9 complexes typically induce bi-allelic indels[18]. Mono-allelic editing can be obtained by temporarily limiting the duration of Cas9 activity and/or controlling the Cas9 doses delivered into cells through various experimental options, including temporal drug selection, fluorescence-activated cell sorting-based cell selection and the delivery of Cas9 mRNAs but not plasmids[19,20]. Cas9 RNPs are also available for this purpose. However, these approaches require fine optimization to control bi-allelic vs mono-allelic editing, depending on individual gRNA sequences and target cell types. In addition, the Cas9 system can be utilized for HDR-mediated gene cassette knock-in (KI) and target conversion, including mono-allelic single-nucleotide substitution. The most challenging of these editing processes is precise mono-allelic single-nucleotide substitution for the modelling and correction of disease models, which is rarely achieved because of inevitable re-editing and bi-allelic editing events[21,22]. The extent to which Cas9 activity should be limited in such applications and whether such optimization can be easily adapted for various sgRNAs and cell types have not been rigorously established.

Therefore, to determine the feasibility of Cas9 inhibition approaches, it is important to precisely determine the relationships among Cas9 activity strength, allelic configurations for editing, adverse effects and editing outcomes. Although general PCR-based Sanger sequencing and next-generation sequencing (NGS) have been widely used to assess CRISPR-related technologies, they cannot detect the large genomic deletions caused in up to 40% of cells[8,23–25]; therefore, complex genome damage and genotoxicity were underestimated in previous studies. NGS also neglects clonal and allelic editing information, including the prevalence of mosaicism. To overcome these limitations, we developed a convenient but accurate experimental system to visualize genome editing dynamics, including large genomic deletions, in each allele at the single-cell level in living cells. This allele-specific indel monitor system (AIMS) allows the rapid and real-time quantitation of various editing patterns of a pair of alleles in a large number of clones without sequencing analysis.

We adopted AIMS for the systematic analysis of bi-allelic and mono-allelic editing, then re-assessed the sequence configuration of widely used gRNAs to determine whether simple gRNA modification could enable programmable Cas9 inhibition. We demonstrated that adding cytosine stretches to the 5′-end of conventional gRNAs reduces the genome editing efficiency in a length-dependent manner via multiple mechanisms. We also developed computational simulations to obtain an overall snapshot of the relationships among gRNA modification, Cas9 activity, Cas9 specificity, cytotoxicity and HDR efficiency. The results of this study establish distinct optimal windows of Cas9 activity for diverse applications, including safe bi-allelic editing, mono-allelic editing, and HDR-based generation and correction of disease-associated single-nucleotide substitutions free from p53 activation.

## Results

### AIMS
AIMS employs two monitor cassettes that contain 2A self-cleaving peptides (P2A) and two distinct fluorescent proteins (tdTomato and Venus), which are inserted in-frame immediately downstream of coding regions of the same genes at two alleles (Fig. 1a and Extended Data Fig. 1a). We used multiple sgRNAs that target the P2A sequence to analyse the indel induction capacities of these sgRNAs (Fig. 1b). By knocking in the AIMS cassette downstream of the coding regions of various genes with distinct cellular localization patterns, AIMS can distinguish in-frame indels, frameshift indels or large deletions, and a lack of indels (no indel) at each allele, according to changes in fluorescence localization (Fig. 1a). No indel and frameshift indels resulted in diffuse distributions of fluorescent proteins (generated by P2A peptide cleavage) and loss of fluorescence, respectively. When nuclear transcription factors (TF) or membrane proteins (MP) were targeted, in-frame indels disrupted endopeptidase recognition of P2A peptides, yielding fusion proteins that consisted of target proteins and fluorescent proteins, followed by fluorescence localization shifts to the nucleus or membrane, respectively. AIMS is also sensitive to large deletions, which cause loss of fluorescence. In AIMS, verifiable sgRNAs can be expanded through the generation of P2A variants with silent mutations (Fig. 1b). In this study, we used the original P2A sequence (P2A$_1$)[26] and one of its variants (P2A$_2$) to test six sgRNAs that targeted P2A$_1$ or P2A$_2$ (Fig. 1b). We developed AIMS in mouse embryonic stem cells (mESCs) by targeting T-Box transcription factor 3 (*Tbx3*) and membrane protein E-cadherin (*Cdh1*) because they are homogeneously expressed in mESCs under 2iL culture[27,28] (Fig. 1a and Extended Data Fig. 1a). AIMS distinguished nine combinations of fluorescence patterns, which were consistent with sequence validation (Fig. 1c and Extended Data Fig. 1b,c).

### Visualization of frequent mosaicism and AIMS accuracy
To enhance experimental reproducibility, we mainly utilized all-in-one plasmids expressing sgRNA, Cas9 and puromycin-resistant cassette (p:RCP); we performed AIMS analysis in cells selected by puromycin treatment (Fig. 1d). Approximately 30% of primary colonies derived from puromycin-resistant single cells exhibited mosaicism (Fig. 1d). Therefore, primary colonies were dissociated; secondary colonies with homogeneous fluorescent patterns were analysed (Fig. 1d). Bi-allelic indels were induced in >99.4% of mESC clones for all six sgRNAs (Fig. 1e). Similar results were obtained when we targeted endogenous genes including the *Alb* gene, not expressed in mESCs (Fig. 1f). Allelic bias was not evident in either indel induction or frameshift/in-frame indel frequency in bi-allelic indel clones (Fig. 1g and Extended Data Fig. 1d). These results collectively suggest that the current plasmid-based CRISPR-Cas9 system induces bi-allelic DNA cleavage when appropriate sgRNAs are designed and sgRNA-Cas9-introduced cells are sufficiently selected; moreover, allelic selection is stochastic and highly dynamic, leading to frequent mosaicism.

Next, we investigated the accuracy of AIMS data-based indel probability (AIMS[P]) through additional sequence analysis of a rare population of tdTomato$^+$/Venus$^{indel}$ and tdTomato$^{indel}$/Venus$^+$ heterozygous clones (Fig. 1h). Of these ostensibly heterozygous clones, 86% was homozygous, resulting in an error frequency of <0.3% (Fig. 1h). Next, we performed a standard T7E1 survey assay with a bacterial cloning process, determined the indel probability (T7E1-Bac[P]) and estimated the error rates (Fig. 1i). Additional sequence analysis revealed that approximately 8% of indels was not digested by T7E1 (Fig. 1j), suggesting that AIMS was more accurate than the T7E1 assay. Therefore, we performed both a T7E1 assay and sequence analysis to determine bacterial cloning-based indel probability (Bac[P]) in subsequent experiments.

### Fine-tuning of Cas9 activity with cytosine extension on sgRNAs
Consistent with our findings, other methods such as CORRECT have employed bi-allelic editing to generate heterozygous genotypes but incorporated technical approaches such as mixed HDR templates to control the editing outcomes of two alleles for heterozygosity[21,22,29]. Thus, we attempted to maximize mono-allelic genome editing by reducing excessive activity. Reducing the amounts of all-in-one plasmid or sgRNA-expressing plasmid failed to increase clones with mono-allelic indels (Fig. 2a and Extended Data Fig. 2a). This suggested that sufficient expression of Cas9 and sgRNA, which are selected by puromycin, results in bi-allelic indels in most cells even with a lower plasmid amount

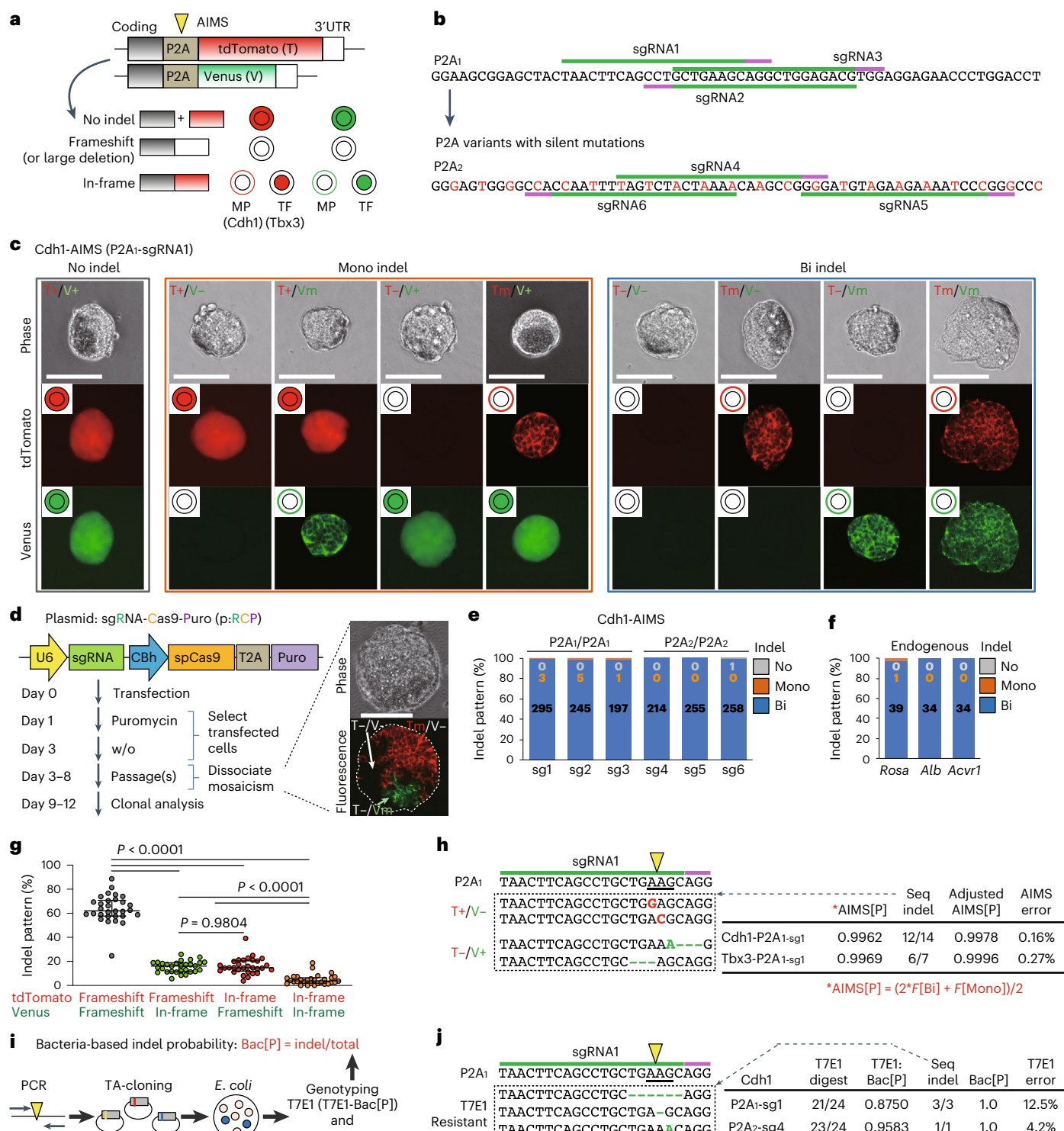

**Fig. 1 | Visualization of allele-specific genome editing events using AIMS.**
**a**, Schematic of AIMS. P2A sites are targeted by sgRNA-Cas9 (yellow pointer). MP, membrane protein; TF, transcription factor. **b**, Target sequences of sgRNAs in the original P2A (P2A₁) and a variant generated by silent mutations (P2A₂, indicated in red). **c**, Results of Cdh1-P2A₁-AIMS in mESCs. T, tdTomato; V, Venus; +, no indel; m, in-frame indel indicated by membrane localization; −, frameshift indel or large deletion indicated by loss of fluorescence. Scale bar, 100 μm. **d**, The protocol with all-in-one CRISPR plasmids (left). The images show an edited mESC colony with mosaicism (right). Scale bar, 100 μm. **e**, Indel patterns measured using Cdh1-AIMS in mESCs. Data are means of 3 independent experiments, except for sg1 (n = 6). The total number of clones analysed is shown in each column

(in **e** and **f**). **f**, Indel patterns for endogenous gene editing in mESCs. Data are means of 3 independent experiments. **g**, Percentages of the four types of bi-allelic indel pattern (n = 30; 6 sgRNAs, Tbx3- and Cdh1-AIMS in mESCs); the median and interquartile ranges are shown. Statistical significance was assessed using Welch's ANOVA and a post hoc Games–Howell test. **h**, Representative indel sequences in the P2A₁ region of a tdTomato or Venus allele in T⁺/V⁻ or T⁻/V⁺ clones, respectively (left), and AIMS error rates (right). Pointers in **h**–**j** indicate DSB sites. Codons are underlined. See Methods for details of the calculation of error rates in **h** and **j**. **i**, Schematic of procedure for calculating bacteria-based indel probability (Bac[P]). **j**, Representative T7E1-insensitive indel sequences (left) and error rate of the T7E1 assay (right). See also Extended Data Fig. 1.

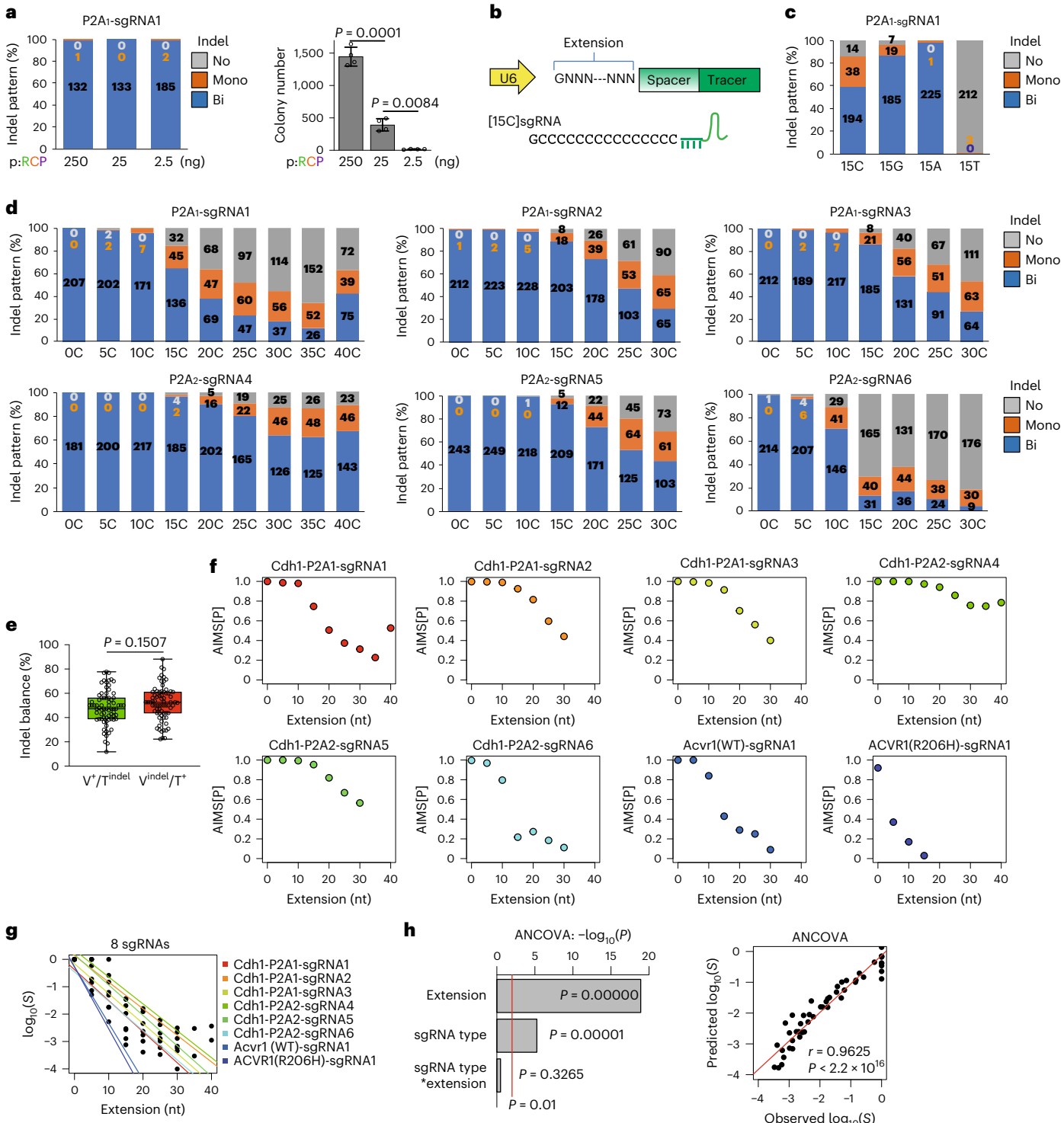

**Fig. 2 | Fine-tuning of editing frequency through the addition of 5′-end cytosine stretches. a,** Analysis of indel patterns (left) and colony numbers (right) after transfection of pRCPs (shown in Fig. 1d) in mESCs with Tbx3-P2A₁-AIMS. Data are means ± s.d. (n = 4 biological replicates). The total number of clones is shown in each column (in **a**, **c** and **d**). Statistical significance was evaluated using Welch's ANOVA and a post hoc Games–Howell test. **b,** Schematic of nucleotide extension at the 5′-end of the spacer. **c,** Effects of 15-base cytosine [15C], guanine [15G], adenine [15A] and thymidine [15T] extension of sgRNA1 in mESCs with Cdh1-P2A₁-AIMS. Data are means of 3 independent experiments (for **c** and **d**). **d,** Indel pattern analysis for Cdh1-AIMS in mESCs. **e** Boxplot showing percentages of mono-allelic indel frequencies for tdTomato and Venus alleles (n = 73;

6 sgRNAs, Tbx3- and Cdh1-AIMS in mESCs). Statistical significance was evaluated using two-tailed Student's t-test. In the boxplots, the centre lines show medians; box limits indicate the 25th and 75th percentiles; whiskers go down to the smallest and up to the largest values. **f,** Relationships between [C] extension length and AIMS[P]. Acvr1(WT)-sgRNA1 and ACVR1(R206H)-sgRNA1 are used in Figs. 6 and 7. **g,** Relationships between [C] extension length and concentrations of effective sgRNA-Cas9 complex (log₁₀(S)). See also Extended Data Fig. 3d. **h,** Left: ANCOVA results for each source of variance. Right: correlations between observed and predicted log₁₀(S). Linear regression line, Pearson's correlation coefficient (r) and P value are shown. See also Extended Data Figs. 2 and 3.

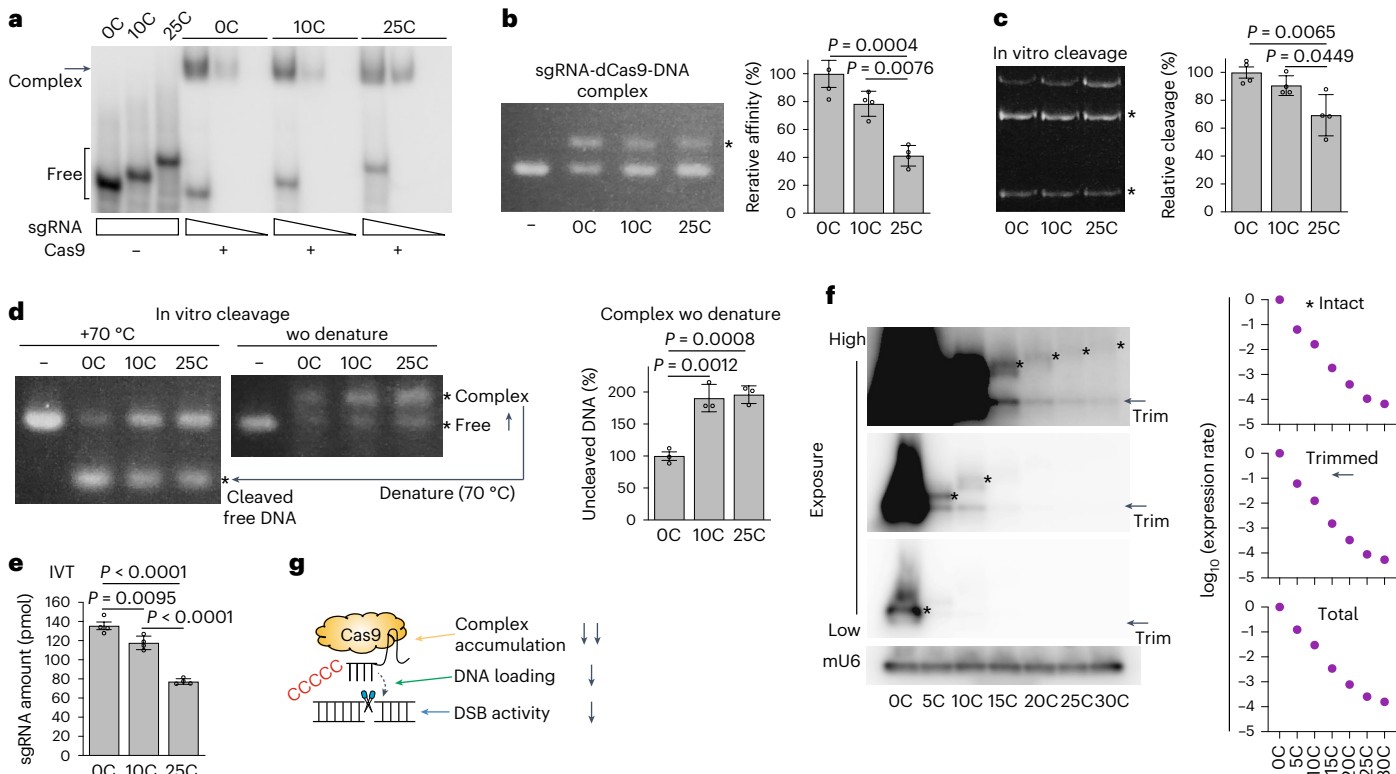

**Fig. 3 | Mechanism for downregulation of Cas9 activity by [C] extension. a**, Gel shift assay to examine the binding affinity of [C]sgRNA to Cas9. **b**, Left: gel shift assay to examine the binding affinity of sgRNA-Cas9 complex to DNA. Asterisk indicates DNA fragments shifted by complex formation. Right: quantitation of results. **c**, Left: in vitro cleavage assay. Asterisks indicate digested DNA products. Right: quantitation of results. **d**, Left: results of a combination DNA gel shift and cleavage assay. Center: accumulation of uncleaved DNA that maintained

an RNP–DNA conformation was detected by omitting the 70 °C denaturation process. Right: quantitation of results. **e**, Results of an IVT assay to evaluate the transcriptional capacity of [C]sgRNA in vitro. **f**, Left: northern blot analysis results for [C]sgRNA expression levels in mESCs. Right: quantitation of results. **g**, Schematic of the mechanism used to reduce Cas9 activity. In **b**–**e**, data are means ± s.d. (n = 4 biological replicates), and statistical significance was assessed using Welch's ANOVA and a post hoc Games–Howell test. Source data.

for transfection. Next, we tested the addition of 15-base stretches of guanine [15G], cytosine [15C], adenine [15A] and thymidine [15T] to the 5′-ends of spacer sites on the basis of previous reports that a few additional guanines at the 5′-end may interrupt sgRNA-Cas9 activity[30,31] (Fig. 2b). Importantly, [15C] extension substantially increased the frequency of mono-allelic indel clones (Fig. 2c); [15T]sgRNA almost completely failed to induce indels, perhaps in relation to sgRNA expression loss, because [15T] contained a 4xT transcription termination signal for the U6 promoter[32]. Therefore, we focused on cytosine ([C]) extension in subsequent experiments.

We investigated the relationships between [C] extension length and bi-/mono-allelic indel patterns by systematically generating all-in-one plasmids that expressed [0C]–[30C]-extended sgRNAs for six different sgRNA sequences (Extended Data Fig. 2b). For all six sgRNA sequences, [C]-extended sgRNAs ([C]sgRNAs) exhibited decreased bi-allelic indels and increased mono-allelic indels in a length-dependent manner, indicating length-dependent editing suppression (Fig. 2d). In addition, length-dependent suppression accompanied increased mosaicism (63% for [20C]sgRNA) (Extended Data Fig. 2c). Allelic bias was not observed in mono-allelic indel induction (Fig. 2e and Extended Data Fig. 2d).

Genome editing efficiency is reportedly influenced by the local genome environment and cell types[33], even when the same sequences are targeted, as confirmed in Extended Data Fig. 3a–c. We determined the probability of single-allele editing (AIMS[P]) on the basis of the frequency of cells with bi-allelic and mono-allelic editing; we observed that the absolute indel probabilities of [C]sgRNAs varied among sgRNAs (Fig. 2f). To separate the variations in sgRNA sequences and the

effects of [C] extension, we determined the relative concentrations of effective sgRNA-Cas9 complexes (S) compared with sgRNAs without [C] extension, for each of the six sgRNAs and other sgRNAs used in this study (Fig. 2g, Extended Data Fig. 3d and Methods). Importantly, we found clear and similar inverse relationships between [C] extension and the relative concentrations of effective sgRNA-Cas9 complexes for all analysed sgRNA sequences (Fig. 2g and Extended Data Fig. 3d). The relative effects of [C] extension differed little among sgRNA sequences (Fig. 2g and Extended Data Fig. 3d), suggesting uniform effects on diverse sgRNA sequences. This finding was supported by analysis of covariance (ANCOVA), which showed that the combinatorial effects of sgRNA sequences and [C] extension were marginal (Fig. 2h). Together, these findings suggest that [C] extension decreases the relative concentrations of effective sgRNA-Cas9 complexes in a length-dependent manner, irrespective of the sgRNA sequence.

## Mechanisms of Cas9 activity reduction via [C]sgRNAs

To elucidate the mechanisms of Cas9 inhibition, we performed in vitro assays to directly test the effects of [C] attachment to sgRNAs on DNA loading, DNA cleavage and sgRNA transcription. A gel shift assay using synthetic sgRNAs and recombinant wild-type (WT) Cas9 proteins showed that Cas9 binds similarly to both standard sgRNAs and [C]sgRNAs (Fig. 3a). Next, we prepared a complex that consisted of [C]sgRNA and catalytically dead Cas9 (dCas9) proteins. The in vitro pre-formed [C]sgRNA-dCas9 complexes exhibited decreased loading capacity to target DNA in a gel shift assay (Fig. 3b). We performed a DNA cleavage assay using the pre-formed [C]sgRNA-WT Cas9 complexes; the efficiency of DNA cleavage was modestly reduced by [C]sgRNAs

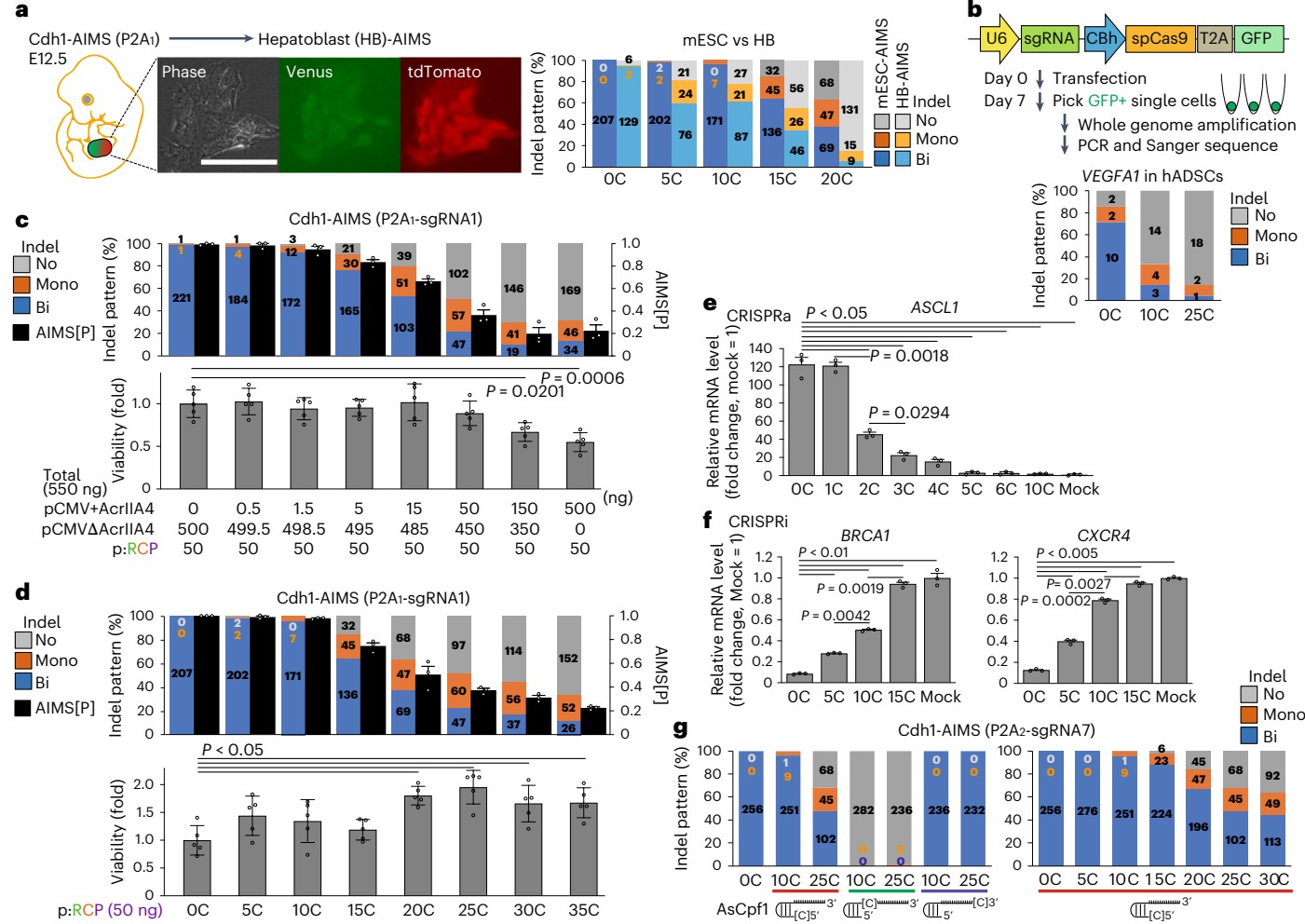

**Fig. 4 | Widespread applicability of the [C]sgRNA system. a**, Left: establishment of an HB-AIMS cell line from Cdh1-P2A₁-AIMS embryos at E12.5 (left). Right: indel analysis using P2A₁-sgRNA1. Data on the indel pattern of mESC-AIMS (Cdh1-P2A₁-sgRNA1) from Fig. 2d are redisplayed for comparison. The total number of clones is shown in each column (for **a**–**d** and **g**). **b**, Single-cell analysis of indel patterns for endogenous *VEGFA1* targeting in hADSCs. Top: scheme. Bottom: the result. **c**,**d**, Effects of AcrIIA4 (**c**) and [C]sgRNA (**d**) on Cas9 activity. Top: indel pattern analysis and AIMS[P] (*n* = 3 independent experiments). Bottom: cell viability (*n* = 5 biological replicates) in mESC-AIMS (Cdh1-P2A₁-sgRNA1). In AcrIIA4 co-transfection experiments (**c**), total plasmid amounts were kept constant. Data from Fig. 2d and Extended Data Fig. 3a are redisplayed

for comparison. pCMV+AcrIIA4, AcrIIA4-expressing plasmid. pCMVΔAcrIIA4, AcrIIA4 truncation plasmid derived from AcrIIA4-expressing plasmid. **e**,**f**, Effects of [C]sgRNAs on CRISPRa (**e**) and CRISPRi (**f**) platform in HEK293T cells (*n* = 3 technical replicates). Mock indicates spacerless all-in-one plasmids for CRISPRa/i and is used as a reference value (1.0) to define a fold change. **g**, Effects of [C]-extension site. Left: red, 5'-end; green, between hairpin and spacer sites; blue, 3'-end. Right: length on AsCpf1 in mESC-AIMS (Cdh1-P2A₂-sgRNA7, *n* = 3 independent experiments). Statistical significance was assessed using one-way ANOVA and a post hoc Tukey–Kramer test (**c** and **d**) or Welch's ANOVA and a post hoc Games–Howell test (**e** and **f**). Data are means ± s.e.m. (**c** and **d** top) or means ± s.d. (**c** and **d** bottom, **e** and **f**). See also Extended Data Fig. 4.

(Fig. 3c). These effects were produced in a length-dependent manner (Fig. 3b,c). We also performed a DNA cleavage assay with and without denaturing the DNA-RNP complex; [C] extension resulted in the accumulation of uncleaved DNA-RNP complexes under non-denaturing conditions (Fig. 3d).

An in vitro transcription assay showed a modest [C] length-dependent decrease in gRNA synthesis (Fig. 3e). In contrast, northern blot analysis showed that expression levels in mESCs were dramatically decreased by [C] extension in a length-dependent manner (Fig. 3f). Importantly, we observed a large dynamic range of suppression, which was comparable with the inferred range of relative concentrations of effective sgRNA-Cas9 complexes (Fig. 2g). We also observed comparable levels of intact [C]sgRNAs and trimmed sgRNAs, which were consistent with previous findings regarding trimming of extended sgRNAs in cells[34]. Considering that Cas9/sgRNA complex formation enhances sgRNA stability[35,36], the stark contrast between the modest

suppression of in vitro transcription and the dramatic decrease in in vivo sgRNA accumulation suggests decreased efficiency of functional complex formation in vivo. This result may partly be explained by competition among [C]sgRNAs, Cas9, other RNA-binding proteins (for example, poly(rC)-binding proteins) and cellular RNAs because assembly of the gRNA-Cas9 complex is reportedly influenced by non-specific RNA competitors and cellular RNAs[37,38]. Together, these results suggest that [C] extension reduces the intracellular fitness between gRNA and Cas9 and the formation of effective Cas9-gRNA complex via multiple mechanisms, finally leading to inhibition over a large dynamic range (Fig. 3g).

### Widespread applicability of the [C]sgRNA system

We further verified the applicability of the [C]sgRNA system by testing in other cell lines, comparing with other Cas9 inhibition approaches and extending into other CRISPR-related technologies. For this purpose,

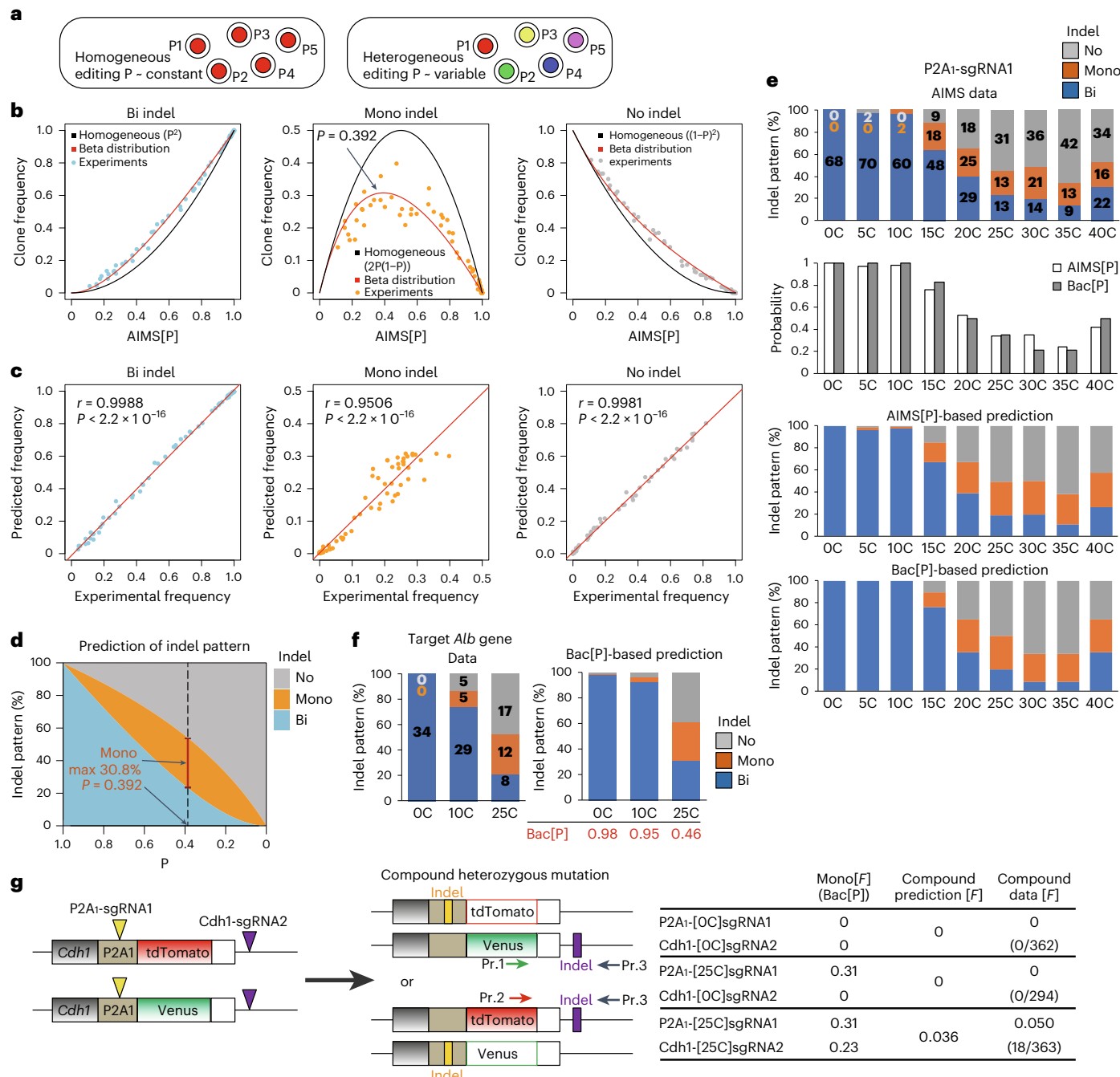

**Fig. 5 | Computational prediction of single-cell genome editing heterogeneity and bi-/mono-allelic indel frequency. a**, Homogeneity vs heterogeneity of genome editing frequency at the single-cell level. **b**, Relationships between AIMS[P] and clone frequency for bi-, mono- or no-indel status ($n = 64$; 6 sgRNAs, Tbx3-AIMS and Cdh1-AIMS in mESCs). **c**, Correlation between experimental data and beta distribution-based prediction. Linear regression curves, Pearson's correlation coefficients ($r$) and $P$ values are shown. **d**, Simulated relationships between indel probability (P) and allelic indel pattern using the beta distribution model. **e**, Comparison of AIMS experimental results using Cdh1-P2A₁-sgRNA1 in mESCs and predictions based on AIMS[P] and

Bac[P]. **f**, Comparison of experimental data (left) and Bac[P]-based indel pattern predictions (right) for endogenous *Alb* (*Albumin*) gene targeting in mESCs. In **e** and **f**, data are means of 3 independent experiments and the total number of clones analysed is shown in each column. **g**, Prediction and generation of compound heterozygous mutation clones using Cdh1-P2A₁-AIMS in mESCs. Mono[*F*] (Bac[P]), frequency of mono-allelic indel Mono[*F*] predicted from Bac[P]. Numbers of compound heterozygous mutation clones generated in 3 independent experiments are indicated in brackets. Pointers and boxes indicate DSB sites. Pr, genotyping primer. See also Extended Data Fig. 5 and Supplementary Tables.

we generated AIMS mouse and established hepatoblast (HB) cell line (HB-AIMS) to compare editing outcomes under the same conditions of mESC-AIMS experiments and analysis protocols (Fig. 4a). HB was chosen on the basis of homogeneous expression of the *Cdh1* gene. The HB-AIMS showed similar bi- to mono-indel shift patterns along with

[C] extension, while the Cas9 activity rapidly decreased in HB-AIMS compared with mESC-AIMS (Fig. 4a). In addition, we confirmed that mono-allelic editing was efficiently induced by [C] extension in both knockout experiments in human adipose-derived stem cells (hADSCs) (Fig. 4b) and ssODN-mediated knock-in experiments in

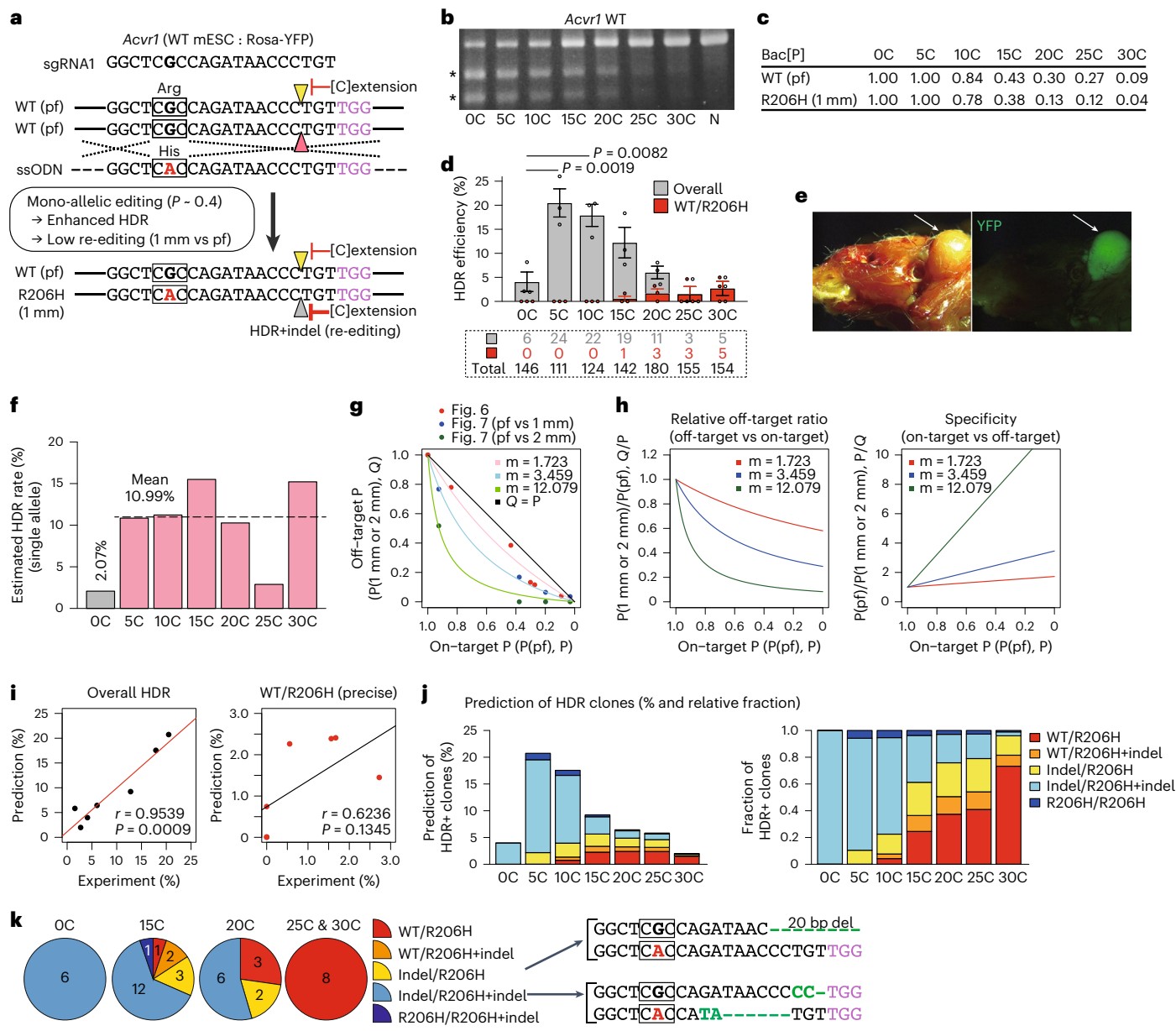

**Fig. 6 | Generation of an FOP disease model and computational modelling.**
**a**, Schematic of precise HDR for mono-allelic G>A replacement in WT/WT mESCs. Pointers indicate DSB sites. Squares indicate codons. pf, perfect match; 1 mm, 1-bp mismatch. **b**, T7E1 assay results. Asterisks indicate PCR products digested by T7E1. N, PX459 plasmid without spacer. **c**, Bac[P] values for both WT and R206H alleles. The on-target and off-target activities were measured by T7E1-based Bac[P] assays. **d**, Clonal analysis of overall HDR and precise HDR (WT/R206H) efficiencies. Overall HDR comprises precise HDR and other HDRs with indels. Numbers of clones analysed are shown in dotted rectangle. Data are means ± s.e.m. of 3 independent experiments. Statistical significance for overall HDR was assessed using one-way ANOVA and a post hoc Tukey–Kramer test. **e**, Generation of an FOP mouse model. Arrows indicate areas of ectopic ossification with mESC contribution, traced by the Rosa-YFP reporter.

**f**, Computational estimation of HDR rates at the single-allele level. **g**, Relationships between on-target (pf, P) and off-target editing probability (1 mm or 2 mm, Q). Computational fitting results are shown. Red, blue and green dots indicate experimental data shown in Figs. 6c and 7c (pf vs 1 mm) and Fig. 7c (pf vs 2 mm), respectively. **h**, Computational analysis of the decrease in relative off-target editing (left), increase in on-targeting specificity (right) and reduction in indel probability. **i**, Correlation between experimentally and computationally predicted HDR frequencies for overall (left) and precise WT/R206H (right). Linear regression curves, Pearson's correlation coefficients (r) and P values are shown. **j**, Prediction of diverse HDR events (left) and relative fraction (right). **k**, Detailed distribution of HDR events shown in Fig. 6d; actual clone numbers (left) and indel sequences (right) are shown. Squares indicate codons. In **g** and **h**, P indicates indel probability. See also Extended Data Fig. 7.

hiPSCs (Extended Data Fig. 4a). In contrast to mESCs, even short extension such as [5C] and/or [10C] strongly decreased editing efficiency in hADSCs and hiPSCs. The data collectively suggest that the relationship between Cas9 activity and frequency of bi-/mono-allelic editing is conserved among various cell types, while the editing sensitivity to the [C] length varies. Cell type differences may be associated with various factors including transfection efficiency, proliferation rate and puromycin sensitivity.

We next compared the [C]sgRNA system and other Cas9 inhibition approaches by using anti-CRISPR protein AcrIIA4 (refs. 10–13) and small-molecule inhibitors BRD0539 (ref. 16). In the initial co-transfection experiments, we observed that increasing amounts of AcrIIA4 expression plasmids could inhibit Cas9 activity in a regulated manner but induced cytotoxicity (Extended Data Fig. 4b). When holding the total amounts of plasmids constant to exclude potential DNA toxicity, toxicity was mitigated but still observed for higher doses of

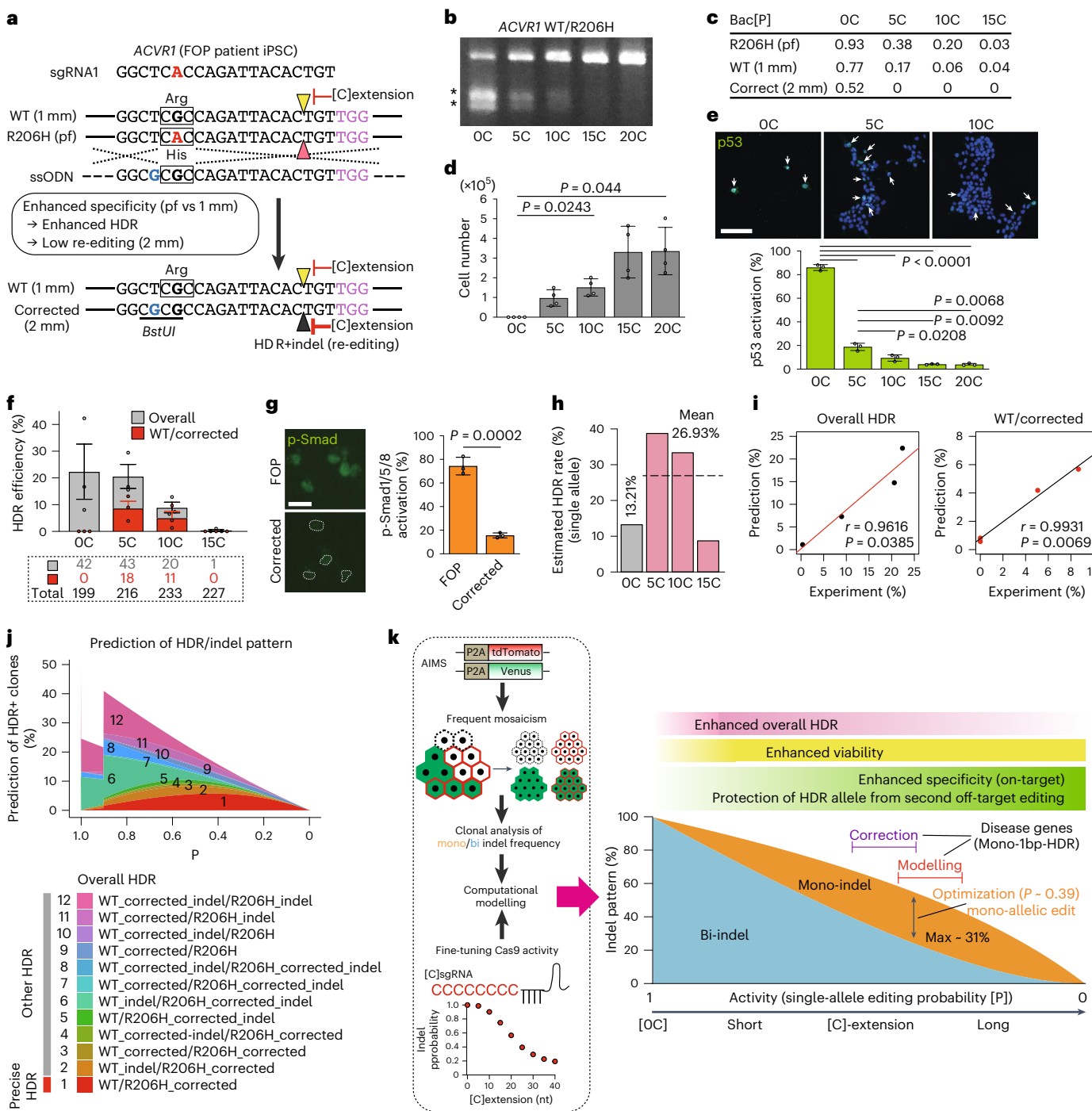

**Fig. 7 | Safe, systematic and precise gene correction in FOP hiPSCs and computational modelling. a**, Schematic of R206H allele-selective precise HDR for A>G correction in FOP iPSCs (WT/R206H). Silent mutation of guanine (G, blue) creates a *BstUI* restriction enzyme site. Pointers indicate DSB sites. Squares indicate codons. pf, perfect match; 1 mm, 1-bp mismatch; 2 mm, 2-bp mismatch. **b**, T7E1 assay results. Asterisks indicate PCR products digested by T7E1. **c**, Bac[P] values for R206H (pf), WT (1 mm) and corrected (2 mm) alleles. **d**, Cytotoxicity was examined by counting cells after all-in-one plasmid transfection and puromycin selection (*n* = 4 biological replicates). **e**, Immuno-cytochemistry results for p53 activation in FOP hiPSCs (*n* = 3 biological replicates). Arrows indicate p53-activated cells. **f**, Clonal analysis of overall and precise HDR (WT/Corrected) efficiencies in FOP hiPSCs (*n* = 3 independent experiments). Overall HDR comprised WT/Corrected and 11 other genotypes,

as shown in Fig. 7j. Numbers of clones analysed are shown in dotted rectangle. **g**, Left: immunocytochemistry results for activin-induced pSmad1/5/8 activation in FOP hiPSCs and a corrected clone (*n* = 3 biological replicates). Right: quantitation of results. **h**, Computational estimation of HDR rates at the single-allele level. **i**, Correlation between experimental HDR frequencies and computational modelling for overall (left) and precise WT/Corrected (right) HDR. Linear regression curves, Pearson's correlation coefficients (*r*) and *P* values are shown. **j**, Simulation of the relationships between indel probability (P) and various HDR outcomes. **k**, Summary of the present study. Statistical significance was assessed using Welch's ANOVA and a post hoc Games–Howell test (**d** and **e**) or two-tailed Student's *t*-test (**g**). Data are means ± s.e.m. (**f**) or means ± s.d. (**d**, **e** and **g**). See also Extended Data Figs. 8 and 9.

AcrIIA4 expression plasmids (Fig. 4c). In contrast, the [C]sgRNA system did not show increased cytotoxicity, instead increasing viability along with [C] extension (Fig. 4d). Alleviation of p53 activation and cytotoxicity in the [C]sgRNA system is further characterized in subsequent experiments. While BRD0539 is reported to inhibit SpCas9 in the eGFP-disruption assay with an $EC_{50}$ of 11.5 µM (ref. 16), we failed to observe Cas9 inhibitory effects of BRD0539 and observed cytotoxicity in mESCs at concentrations of 30 µM (Extended Data Fig. 4c). Thus, both anti-CRISPR protein and the [C]sgRNA system can reduce Cas9 activities in a regulated manner, although the former may be associated with some toxicity especially at higher doses.

We also verified the effects of long-spacer gRNAs, called the self-targeting gRNA (stgRNAs) with more than 20 nt-spacer site[39], using AIMS. When targeting two different sites, Cas9 activity was not decreased by both 30 nt- and 40 nt-spacer sgRNAs (Extended Data Fig. 4d). On the other hand, the spacer-length-dependent inhibition of Cas9 activity was observed when [10C] and [25C] were added to the spacers (Extended Data Fig. 4d). A combination of [C]sgRNAs and other gRNA modification, such as long-spacer gRNAs and mismatch gRNAs[40], may further extend applicability of the fine-tuning approaches.

From the standpoint of biomedical applicability, the [C]sgRNA system can be potentially combined with other CRISPR-related technologies. We tested whether cytosine extension methods are compatible with other CRISPR tools such as CRISPR activator (CRISPRa), CRISPR interference (CRISPRi) and Cas12a (AsCpf1) to finely tune their activities. Indeed, we confirmed that the the [C]sgRNA system can finely tune the activities of CRISPRa and CRISPRi (Fig. 4e,f and Extended Data Fig. 4e). For CRISPRa, we observed that short [C] extension is sufficient for fine tuning of CRISPRa-mediated transcriptional activation (Fig. 4e and Extended Data Fig. 4e). As for Cas12a, we constructed puromycin-selectable PX459-based all-in-one plasmids that express AsCpf1 instead of Cas9 and investigated indel induction using AIMS to investigate bi-/mono-editing outcomes. Since the gRNA structure of 5′-PAM-spacer-3′ for Cas12a is opposite to that of 5′-spacer-PAM-3′ for Cas9, we first addressed defining the position of [C] extension that can efficiently decrease AsCpf1 activity in a length-dependent manner by adding [10C] and [25C] (Fig. 4g, left). Interestingly, the activity was length-dependently suppressed when adding [C] at the 5′-end of sgRNA. On the other hand, adding [C] between hairpin and spacer sites lost Cas12a activity, while 3′ [C] addition did not have suppressive effects. Thus, we chose the 5′-end as the site of [C] addition and examined various lengths of [C] for fine-tuning of AsCpf1 activity. As expected, the activity decreased and the frequency of mono-allelic indels increased in a length-dependent manner (Fig. 4g, right). Taken together, the safeguard [C]sgRNAs can be applied to the Cas12a system by extending [C] at the 5′-end of sgRNA. Of note, TTTN PAM sequence, which is required for AsCpf1 gRNA, could not be set within 66 bp of a $P2A_1$ site; hence, we used a $P2A_2$ site and another sgRNA sequence ($P2A_2$-sgRNA7). In this regard, the AIMS is a powerful tool to investigate modulation of various types of CRISPR-Cas system by altering the P2A sequence with silent mutations that can match various PAM sequences. These results collectively suggest that the [C]sgRNA system can be conveniently applied in various cell lines and for various CRISPR-related technologies.

### Computational modelling of single-cell editing frequency heterogeneity and maximization of mono-allelic editing

Our comparison suggests that [C]sgRNA can conveniently increase mono-allelic editing without cytotoxicity (Fig. 4). Theoretically, mono-allelic editing can be maximized to 50% by setting the indel probability to 50% under the assumption of a homogeneous cell population. However, we found that the actual frequency of mono-allelic indels ($F$(Mono)) was substantially lower than the estimated $F$(Mono), particularly at intermediate AIMS[P] levels (AIMS[P] = ~0.5) (Fig. 5a,b). Therefore, we examined genome editing frequency heterogeneity

at the single-cell level. We used the beta distribution to model editing frequency distribution at the single-cell level, on the basis of the population-level editing frequency; we identified the optimal setting ($\alpha$ = 0.715) (Extended Data Fig. 5a–d and Methods). This model predicted frequencies of bi-, mono- and no-indel induction that were highly consistent with the AIMS data (Fig. 5b,c and Supplementary Table 1). The simulation results indicated that the highest frequency of mono-allelic indel induction was 30.8% for an AIMS[P] of 0.392 (Fig. 5d and Supplementary Table 2). In our experiments, [C] extension between 15 and 30 nucleotides was generally optimal for mono-allelic indel induction (Fig. 2d and Extended Data Fig. 3a). The model showed that Bac[P] and AIMS[P] yielded comparable predictions (Fig. 5e); Bac[P]-based predictions could be applied even when targeting the endogenous *Alb* gene (Fig. 5f). These results collectively suggest that heterogeneity in editing efficiency is an important obstacle for efficient mono-allelic editing; continuous fine-tuning of Cas9 activity is important for determining the optimal range of Cas9 activity.

Next, we applied this model to predict the efficiency of compound heterozygous mutation, which is challenging because of dual mono-allelic indel induction in *trans*-chromosomal configuration (Fig. 5g). Compound heterozygous clones were obtained only with the [25C]sgRNA combination; the frequency of 0.050 (18/363) was almost identical to the predicted frequency of 0.036 (Fig. 5g), supporting high prediction accuracy via integration of population-level editing frequency and single-cell heterogeneity.

### Scarless mono-allelic cassette knock-in by reduction of sgRNA-Cas9 activity

We investigated whether [C] extension allows both mono-allelic KI of large gene cassettes via HDR and protection of non-HDR alleles from indel induction (that is, one-step generation of HDR/WT clones) (Extended Data Fig. 6a–d). In cassette knock-in experiments, the overall HDR frequency, which included HDR/indel clones, gradually decreased along with [C] extension because of reduced indel probability (Extended Data Fig. 6a, middle panels). Although the overall HDR frequency of [30C]sgRNA was 3-fold less than the HDR frequency of [0C]sgRNA, the scarless HDR/WT frequency of [30C]sgRNA was 25-fold higher than the HDR/WT frequency of [0C]sgRNA (Extended Data Fig. 6a, right panel). These data indicate that one scarless HDR/WT clone can theoretically be obtained by picking 40 or 1.6 tdTomato-positive KI clones using [0C] or [30C]sgRNA, respectively. Similar results were obtained in the cassette replacement experiments using AIMS, indicating that one scarless HDR/WT clone can theoretically be obtained by picking 137 or 1.9 G418-resistant KI clones using [0C] or [25C]sgRNA, respectively (Extended Data Fig. 6b–d). Therefore, mono-allelic HDR clones without scars on non-HDR alleles can efficiently be obtained by reducing Cas9 activity via [C]sgRNAs.

### Generation of a heterozygous single-nucleotide polymorphism (SNP) disease model through optimized mono-allelic editing

Scarless mono-allelic single-nucleotide editing is the most challenging type of recombination because it involves a high probability of off-target cleavage against a 1-bp mismatch (1 mm) HDR allele[22]. To address this issue, we focused on a fibrodysplasia ossificans progressiva (FOP) disease model, for which a mono-allelic 617 G>A (R206H) mutation in the human *ACVR1* gene is a causal mutation[41]. We attempted to generate an identical mutation in the mouse *Acvr1* gene in WT/WT mESCs (Fig. 6a). An sgRNA was designed for the region that crossed the G>A editing site (Fig. 6a); indel probability reduction by [C] extension was confirmed by T7E1 and Bac[P] analysis (Fig. 6b,c). After transfection with all-in-one CRISPR plasmids and ssODN as an HDR repair template, the frequencies of overall HDR and precise mono-allelic HDR (WT/R206H) were determined by sequence analysis in puromycin-selected cells.

The overall HDR genotype includes the WT/R206H genotype and various other genotypes that harbour indels. In this study, [0C]sgRNA induced overall HDR in only 4.1% of clones. However, the frequency of overall HDR for [5C]sgRNA increased to 20.5%, suggesting enhancement of HDR rate, as assessed in depth in subsequent computational analyses. Then, the overall HDR frequency gradually decreased in parallel with the reduction in indel probability (Fig. 6d). In contrast, the frequency of precise WT/R206H HDR gradually increased with [C] extension; all clones for [25C] and [30C]sgRNAs exhibited the correct WT/R206H genotype, whereas [0C]–[10C]sgRNAs did not induce precise editing (Fig. 6d).

We confirmed acquisition of the FOP phenotype in the WT/R206H clone in chimaeric mice, according to their contribution to ectopic ossification (Fig. 6e). Therefore, we concluded that the 617 G>A SNP alone faithfully reproduced FOP disease in mice, consistent with a similar result using mESC clones that harboured the same SNP and a PGK-Neo selection cassette in an intron 5 region[42].

### Computational modelling of HDR enhancement, off-target suppression and complex editing outcomes

We further analysed the detailed effects of [C] extension on HDR, off-target and complex editing outcomes. On the basis of overall HDR frequency, we computationally estimated HDR rates after DNA cleavage of a single allele by considering the heterogeneity of single-cell editing efficiency (Fig. 6f). This analysis clearly showed that the low HDR rate (2.07%) increased upon [C] extension; each [C]sgRNA exhibited a similarly high HDR rate (mean, 10.99%), except for [25C]. This result suggests that [C] extension generally recovered the HDR rate, which had presumably been suppressed by the conventional CRISPR-Cas9 system.

Despite a general increase in HDR rates, precise WT/R206H clones were obtained only for long [C] extension ([20C]–[30C]), but not for short [C] extension. We assumed that suppressing Cas9 activity would make 1-nucleotide mismatch (1 mm) targets less responsive to off-target cleavage, thereby protecting HDR alleles from secondary indel induction. As shown in Fig. 6c, the ratio of off-target editing (1 mm) to on-target editing (pf) decreased with [C] extension. To examine this relationship more rigorously, we computationally modelled the ratio of off-target to on-target editing on the basis of the assumption that differences in efficiency between on- and off-target editing reflect differences in their dissociation constants. The results demonstrated that the ratio of off-target editing to on-target editing and the on-target specificity decreased and increased with editing frequency suppression, respectively (Fig. 6g,h and Methods). Thus, the protection of HDR alleles from secondary editing became marked with long [C] extension. Consistent with these observations, a strong off-target inhibitory effect by [C] extension was confirmed for multiple off-target loci of other sgRNAs in HEK293T cells and observed even with short [C] extension (Extended Data Fig. 7a,b).

We used estimated HDR rates and off-target activity to computationally model various HDR outcomes solely according to on-target activity (Bac[P]) (Fig. 6i,j, Extended Data Fig. 7c–g and Methods). The predicted frequencies of overall HDR, WT/R206H HDR and various HDR patterns were highly consistent with the experimental results (Fig. 6i–k). The optimal indel probability for precise WT/R206H HDR was predicted to be 0.313, which was slightly lower than the optimal indel probability of 0.392 for mono-allelic indel induction in mESCs, suggesting the use of [20C]sgRNA and [25C]sgRNA (Extended Data Fig. 7g).

### p53-activation-free systematic precise gene correction in human iPSCs

We finally performed R206H allele-specific gene correction by using [C] extension in FOP patient-derived hiPSCs (WT/R206H)[43] (Fig. 7a). The sgRNA was designed for the R206H (pf) allele and transfected with ssODN that contained a silent mutation as a hallmark, which is necessary to distinguish an HDR-corrected (Corrected) allele from an original WT allele (Methods). Efficient indel induction by [0C]sgRNA and its decrease by [5C]–[20C]sgRNAs were confirmed using a T7E1 assay (Fig. 7b). Consistent with the relative suppression of off-target effects, Bac[P] analysis showed that indel probabilities on the WT allele (1 mm) decreased to a greater extent with [5C]sgRNA than did indel probabilities on the R206H (pf) allele (Fig. 7c). Notably, the Corrected allele (2 mm) was further less sensitive to secondary editing (Fig. 7c).

Cas9-mediated DSBs induce potent p53-dependent cytotoxicity in hiPSCs[6,7]. Indeed, we observed that severe cytotoxicity was induced by a conventional [0C]sgRNA (Fig. 7d); p53 was highly activated in 86% of the surviving cells (Fig. 7e). In contrast, such cytotoxicity and p53 activation were dramatically relieved by the application of [5C]–[20C] sgRNAs (Fig. 7d,e). Cytotoxicity inhibition through [C] extension was confirmed by independent experiments that targeted other genes in hiPSCs (Extended Data Fig. 8a–d), although HEK293T cells were tolerant to the conventional system (Extended Data Fig. 8e–h). We observed a [C] extension-mediated sharp decline in the editing efficiency of hiPSCs compared with mESCs (Fig. 7c), which may partly be explained by higher sensitivity of hiPSCs to p53 activation and selection of hiPSCs with non-successful editing and weaker p53 activation.

Next, we determined the frequencies of overall HDR and precise WT/Corrected HDR. The overall HDR frequency of [5C]sgRNA was comparable with the overall HDR frequency of [0C]sgRNA despite a lower indel probability; overall HDR frequency decreased with longer [C] extension (Fig. 7f). Precise WT/Corrected clones were obtained by [5C]sgRNA and [10C]sgRNA, but not by [0C]sgRNA (Fig. 7f). Activin-A-mediated activation of bone morphogenetic protein-responsive Smad1/5/8 was cancelled in the WT/Corrected clone (Fig. 7g), confirming gene correction consistent with previous findings[44].

### Computational simulation of disease allele-specific SNP correction

We performed similar computational modelling of the gene correction experiments (Extended Data Figs. 7c and 9a–e). The HDR rates of a single allele for [0C]sgRNA and [5C]–[15C]sgRNAs were estimated to be 13.21% and 26.93%, respectively (Fig. 7h). The predicted overall and WT/R206H HDR frequencies were strongly correlated with the experimental results (Fig. 7i and Extended Data Fig. 9c). The computational model estimated the frequency of all 12 possible HDR patterns; the results suggested that two populations were dominant when indel probability was high: WT Corrected indel/R206H_indel (fraction 12) and WT indel/R206H_Corrected indel (fraction 6) (Fig. 7j, upper panel, and Extended Data Fig. 9e, upper panel). These findings suggest that indel probability lowering is necessary to prevent secondary editing and allow single-step precise editing. The optimal indel probability for precise HDR was 0.424, suggesting the use of [5C]sgRNA (Extended Data Fig. 9b,e).

Similar to the mESC results, the HDR rate of [0C]sgRNA was estimated to be lower than that of [5–15C]sgRNAs (Fig. 7h). An additional experiment with a 3-bp replacement in HEK293T cells showed HDR enhancement with [5C]sgRNA (Extended Data Fig. 9f–h). Together, these results indicate that precise heterozygous HDR clones can systematically be obtained by reducing Cas9 activity via multiple mechanisms including enhancement of mono-allelic editing, suppression of p53-dependent cytotoxicity, enhancement of HDR rates and suppression of secondary HDR allele cleavage via off-target suppression (Fig. 7k).

## Discussion

Various approaches, such as anti-Cas9 protein and small-molecule inhibitors, can reduce Cas9 activity[9–16]. However, their roles in precise genome editing and safety have not been thoroughly explored[45]. In this study, we designed an easily tunable system comprising an sgRNA

expression unit with cytosine extension ('safeguard sgRNAs') on the basis of the widely used plasmid-based genome editing approach; the resulting strategy avoids the use of other molecules with unknown adverse effects.

We integrated AIMS-based systematic validation and principle-oriented, equation-based computational simulation to provide an overall snapshot of the relationships between Cas9 activity and its multifaceted functional consequences, including HDR rates, viability, specificity (protection from secondary off-target editing) and bi-allelic vs mono-allelic editing (Fig. 7k). Our computational modelling results indicated that control of the large dynamic range of functional sgRNA-Cas9 complex concentrations (10- to 1,000-fold) is necessary to achieve gradual inhibition of single-allele editing frequency. Although the relationship between [C] extension length and Cas9 activity varies among sgRNA sequences, the weak inhibition of Cas9 activity by short extensions is generally beneficial: it improves HDR rates and cell viability while maintaining bi-allelic editing frequency. Strong inhibition by long extension is appropriate for mono-allelic editing and further improves relative on-target specificity (that is, on-target vs off-target activity), which is important for precise editing. Our comparison of 'safeguard sgRNAs', anti-CRISPR proteins and small-molecule inhibitors suggests that [C] extension is a convenient and safe tool. As for the RNP strategy, it might be useful for inducing mono-allelic indels, but reduction of RNP amount should lead to an increase in non-transfected cells and low cloning efficiency of edited cells. Considering that setting the best concentration of RNP for each target and estimating the required clone number for obtaining edited clones would be very laborious in these conditions, the plasmid-based selection method would be easier for obtaining the desired clones.

In terms of short extension, [10C]sgRNA, having 34-fold lower expression levels than [0C]sgRNA (Fig. 3f), could induce bi-allelic indels in most cells at levels comparable with [0C]sgRNA and [5C]sgRNA (Fig. 2d,f). This result suggests that the expression levels of conventional [0C]sgRNA and Cas9 are excessive for cells; such levels presumably induce frequent p53 activation, cytotoxicity and reduced HDR rates. A previous study reported that p53 activation by CRISPR-Cas9 inhibited HDR frequency by 19-fold in hiPSCs[7]. Artificial inhibition of p53 using p53 siRNA, p53 dominant negative forms and p53 antagonist MDM2 can recover cell viability and HDR frequency[6,7,46]; however, it remains uncertain whether such forced suppression of physiological p53 activation is free from long-term genome instability and unexpected side effects. In contrast, we clearly demonstrated that even short [C] extension (for example, [5C]sgRNA) directly avoided p53 activation without the use of p53 activation-inhibiting molecules in hiPSCs (Fig. 7); it enhanced cell viability and HDR rates. Enhanced HDR rates for [5C]sgRNA could be observed in the situation where the on-target activity is ostensibly saturated in Bac[P] assay (Fig. 6c,d). Even if [0C] and [5C]sgRNAs have the maximal on-target and off-target activities in Bac[P] assays (Fig. 6c), within the cells, the temporal frequency of DNA cleavage events across the genome should be substantially lower for [5C]sgR-NAs, thereby enhancing HDR rates (Fig. 6d,f). Therefore, to avoid long-term deleterious effects of excessive DNA damage on cell phenotypes, it may be reasonable to use sgRNAs with short [C] extensions (for example, [5C]sgRNA) for diverse genome editing applications in mammalian cells, particularly pluripotent stem cells.

In addition, we clarified an inhibitory effect of [C] extension on off-target activity regarding 1 mm targets and enhanced on-target specificity. From the standpoint of enzyme kinetics, our analysis clearly demonstrates that suppression of editing probability is inherently coupled with improved specificity (Fig. 6g,h). This is highly consistent with previous reports suggesting that high specificity of two engineered Cas9 (eCas9 and Cas9-HF1) is achieved by not only mismatch-dependent mechanisms, including inhibition of stable DNA binding to partially matching sequences and mismatch-sensitive alteration of DNA unwinding, but also downregulation of the intrinsic cleavage rate[47,48].

Therefore, this scenario may partly explain the increased on-target specificity of other sgRNA modification approaches, including truncated sgRNAs, hairpin sgRNAs and sgRNAs with a couple of guanine addition to the 5′-end[30,31,49–51]. In fact, among these approaches, some, including hairpin sgRNAs and addition of a few guanines at the 5′-end of sgRNAs, have been reported to reduce Cas9 activities[30,31,51]. Inhibition of Cas9 activities by modifications of constant regions of sgRNAs was also recently reported[40]. On the other hand, given that tunability of these approaches has not been well generalized[51] and that the effects are difficult to predict due to heterogeneity across different target sequences[40], our approach adds an alternative option to reduce the sgRNA activity, which is more predictable and universal for diverse sgRNAs. The proper use of anti-Cas9 protein and 'safeguard sgRNAs' may be beneficial for different purposes including synthetic biology[52].

Mono-allelic genome editing using [C]sgRNAs is typically achieved through longer [C] extension. The improved specificity becomes more remarkable in such conditions where the activities are much reduced, thereby enabling both mono-allelic and precise editing. Together with other precise editing approaches such as prime editing[53], this approach would allow the convenient modelling of heterozygous states of disease mutations and risk variants, as well as the investigation of their downstream effects (such as allele-specific epigenome and gene regulation). In our system, precise homozygous mutations can be obtained by repeated mono-allelic editing. Our approach relieves the necessity of multiple complicated steps for precise editing methods such as CORRECT[22], and limitation of target sequences whose base substitution should otherwise disrupt the PAM sequences to prevent second editing. If the re-cleavage frequency of the donor templates (that is, mismatch targets) can be lowered using [C]sgRNA, HDR design could become much easier even in the CORRECT method. In addition, compatibility between the [C]sgRNA system and CRISPRa/i may facilitate the modelling of weak dosage effects of disease mutations and risk variants.

Our analysis shows that [C] extension reduces the intracellular fitness between gRNA and Cas9 and the formation of effective Cas9-gRNA complexes, possibly via multiple mechanisms (Fig. 3g), providing a promising strategy for controlled Cas9 inhibition. AIMS also revealed frequent mosaicism in primary clones of mESCs, even when bi-allelic editing is induced by the standard Cas9 system. The AIMS mouse developed in this study may be useful to investigate in vivo consequences of the mosaic editing. Also, Cas9 inhibition increased the overall frequency of mosaicism. Overall, our study highlights the importance of the careful dissociation of single cells and subsequent clonal analysis, particularly when the editing frequency is reduced using [C]sgRNAs and other inhibition approaches.

## Methods

### Cell culture
We cultured mESCs in t2iL medium containing Dulbecco's modified eagle medium (DMEM, Nacalai Tesque), 2 mM Glutamax (Nacalai Tesque), 1× non-essential amino acids (Nacalai Tesque), 1 mM sodium pyruvate (Nacalai Tesque), 100 U ml⁻¹ penicillin, 100 μg ml⁻¹ streptomycin (P/S) (Nacalai Tesque), 0.1 mM 2-mercaptoethanol (Sigma) and 15% fetal bovine serum (FBS) (Gibco), supplemented with 0.2 μM PD0325901 (Sigma), 3 μM CHIR99021 (Cayman) and 1,000 U ml⁻¹ recombinant mouse leukaemia inhibitory factor (Millipore)[54]. A higher PD0325901 concentration of 1 μM was used for the 2iL medium. mESC colonies were dissociated with trypsin (Nacalai Tesque) and plated on gelatin-coated dishes. Y-27632 (10 μM, Sigma) was added when cells were passaged. hiPSCs were cultured in mTeSR Plus medium (Veritas). hiPSC colonies were dissociated with Accutase (Nacalai Tesque) and plated on Matrigel-coated dishes (Corning, 3/250 dilution with DMEM). Y-27632 and 1% FBS were added when cells were passaged. WT hiPSCs (409B2, HPS0076) were provided by the RIKEN BioResource Research Centre (BRC)[55]. FOP hiPSCs (HPS0376) were provided by RIKEN BRC through the National BioResource Project of the Japan Ministry of

Education, Culture, Sports, Science and Technology (MEXT) and the Agency for Medical Research and Development (AMED)[43]. Experiments using hiPSCs were approved by the Kyushu University Institutional Review Board for Human Genome/Gene Research. HEK293T cells and mouse embryonic fibroblasts were cultured in 10% FBS medium containing DMEM, 2 mM l-glutamine (Nacalai Tesque), 100 U ml⁻¹ penicillin, 100 μg ml⁻¹ streptomycin (P/S) (Nacalai Tesque) and 10% FBS. hADSCs (Thermo Fisher) were cultured in MesenPRO RS medium (Thermo Fisher). Culture conditions of a HB-AIMS cell line are described in the 'Generation of AIMS cell lines and mice and AIMS analysis' section. Cells were maintained at 37 °C and 5% $CO_2$.

## Animals
In this study, we used C57BL/6 mice (Clea Japan), ICR mice (Clea Japan) and $R26R^{YFP/YFP}$ mice (a gift from Frank Costantini at Columbia University, NY, USA)[56]. The experiments were approved by the Kyushu University Animal Experiment Committee, and the care and use of the animals were in accordance with institutional guidelines.

## Oligonucleotides
All primers, spacer linkers and ssODNs used in the present study are listed in Supplementary Table 3.

## Establishing mESCs
Mouse ES B6-5-2 and B6-D2-4 cell lines were established from E3.5 blastocysts of the C57BL/6 strain using 2iL and t2iL medium, respectively; an $R26R^{YFP/+}$ mESC line was established using t2iL medium. Blastocysts were placed on feeders (mitomycin C-treated mouse embryonic fibroblasts) after removal of the zona pellucida. Inner cell mass outgrowths (passage number 0, p0) were dissociated with trypsin and plated on gelatin-coated plates (p1). After domed colonies formed, they were dissociated and passaged (p2). mESC lines were generated by repeating this procedure.

## Generation of AIMS cell lines and mice and AIMS analysis
Knock-in (KI) template plasmids for Cdh1-AIMS were generated by attaching the 5′ and 3′ arms to plasmids containing P2A₁:Venus or P2A₁:tdTomato cassettes. P2A₁ is identical to a widely used P2A sequence[26]. The 5′ arm was designed such that the coding end was fused in-frame to the P2A sequence to allow independent production of both E-cadherin (CDH1) and fluorescence protein. KI plasmids for Tbx3-AIMS were constructed using the same strategy. The alternative P2A sequence P2A₂ was constructed by introducing silent mutations to each codon of the original P2A sequence. The conventional CRISPR-Cas9 system was used to efficiently knock-in the dual-colour plasmids in a pair of alleles. A spacer linker was designed to induce a DSB downstream of the stop codon, then inserted into the *BpiI* sites of a pSpCas9(BB)-2A-Puro (PX459) V2.0 plasmid (Addgene, 62988; see the 'Plasmid construction' section)[57]. All sgRNAs used in this study were designed using the CRISPR DESIGN (http://crispr.mit.edu/) or CRISPOR tool (http://crispor.tefor.net).

The constructed all-in-one CRISPR plasmids and dual-coloured KI plasmids were co-transfected into mESCs using Lipofectamine 3000 (Thermo Fisher). Dissociated mESCs were plated on gelatin-coated 24-well plates with 500 μl of (t)2iL + Y-27632 medium ((t)2iL + Y). Nucleic acid–Lipofectamine 3000 complexes were prepared in accordance with the standard Lipofectamine 3000 protocol. We added 1 μl of Lipofectamine 3000 reagent to 25 μl Opti-MEM medium; simultaneously, 250 ng of each plasmid (all-in-one, Cdh1-P2A-tdTomato and Cdh1-P2A-Venus plasmid) plus 1 μl of P3000 reagent were mixed with 25 μl of Opti-MEM medium in a different tube. These mixtures were combined and incubated for 5 min at room temperature, then added to the 24-well plate immediately after cells were seeded. At 24 h after transfection, puromycin (1.5 or 2 μg ml⁻¹) was added for 2 d and then washed out. The transiently

treated puromycin-resistant cells were cultured for several days; dual-colour-positive colonies were picked and passaged. Genotypes for the candidate dual KI clones were confirmed by PCR. In this study, transfection experiments for mouse and human cells were performed using this procedure, with passage steps added for an AIMS assay to avoid mosaicism (Fig. 1d). Fluorescence microscopes (BZ-X800 (Keyence) and IX73 (Olympus)) were used to analyse the AIMS data. To extract genomic DNA for clonal sequence analysis, single mESC and hiPSC colonies were suspended in 5–10 μl 50 mM NaOH (Nacalai Tesque) and incubated at 99 °C for 10 min. PCR was performed using the template genomic DNA, and the amplicons were sequenced by Sanger sequencing.

For generation of AIMS mice, the established dual KI mESC clone (Cdh1-P2A₁-tdTomato/Venus AIMS) was dissociated with trypsin and 5–8 cells were injected into 8-cell embryos (E2.5) collected from pregnant ICR mice. Injected blastocysts were transferred into the uteri of pseudo-pregnant ICR mice and chimaeras were generated. Male chimaeras were mated with C57BL/6 females, and Cdh1-P2A₁-tdTomato and Cdh1-P2A₁-Venus KI mouse lines were obtained through germline transmission. After the two genotype mice were mated, homozygous AIMS mice were generated.

HB-AIMS cells were established from the E12.5 dual KI embryos according to the protocol of a previous work[58] with some modifications. Briefly, the whole liver was mechanically dissociated and filtrated, and the dissociated cells were seeded onto a type I collagen-coated plate (Iwaki) with the HB medium. The HB medium is composed of a 1:1 mixture of DMEM and F-12 (Nacalai Tesque), supplemented with 10% FBS (Gibco), 1 μg ml⁻¹ insulin (Wako), 0.1 μM dexamethasone (Sigma-Aldrich), 10 mM nicotinamide (Sigma-Aldrich), 2 mM l-glutamine (Nacalai Tesque), 50 μM β-mercaptoethanol (Nacalai Tesque), 20 ng ml⁻¹ recombinant human hepatocyte growth factor (rhHGF) (PeproTech), 50 ng ml⁻¹ recombinant human epidermal growth factor (rhHGF) (Sigma), penicillin/streptomycin (Nacalai Tesque), and small molecules of 10 μM Y-27632 (Wako), 0.5 μM A8301 (Tocris) and 3 μM CHIR99021 (Tocris). After expansion of HBs, a single-cell-derived HB colony with homogeneous expression of tdTomato and Venus was picked and established as an HB-AIMS cell line.

## Plasmid construction
To generate all-in-one CRISPR plasmids for [5C](3A), [10C](8A), [15C](13A), [20C](18C), [25C](23A) and [30C](28A)sgRNA expression, spacer linkers were inserted into the *BpiI* sites of a PX459 plasmid (Extended Data Fig. 2b). In the plasmids, the 3rd, 8th, 13th, 18th, 23rd or 28th cytosine was replaced with adenine because the overhang sequence of CACC is required for linker ligation. The standard spacer linkers (20 nt) or longer spacer linkers (30 nt or 40 nt) were inserted into the *BpiI* sites of the [0C], [5C](3A), [10C](8A), [15C](13A), [20C](18A), [25C](23A) or [30C](28A) PX459 plasmid, leading to generation of [5C]–[30C] sgRNA-expressing all-in-one Cas9 plasmids applicable for puromycin selection. The same [C] linkers were also inserted into the *BpiI* sites of a PX458 plasmid (Addgene, 62988)[57] for selection of GFP-positive transfected cells.

For the plasmid dilution assay, sgRNA-expressing plasmid was constructed by removing a Cas9-T2A-Puro cassette from a PX459 plasmid using the *KpnI* and *NotI* sites. Different amounts of sgRNA-expressing plasmid (0–250 ng) were co-transfected with an unmodified PX459 plasmid (250 ng). In addition, [5C]–[30C] linkers including *BpiI* sites were inserted into this sgRNA-expressing plasmid to construct [5C]–[30C]sgRNA-expressing plasmids, which were used for the experiments of CRISPRa (Extended Data Fig. 4e) described below.

For the CRISPR inhibition experiments, the pCMVΔAcrIIA4 plasmid was generated from the anti-Cas9 AcrIIA4-expressing pCMV+AcrIIA4 plasmid, pCMV-T7-AcrIIA4-NLS(SV40) (KAC200) (Addgene, plasmid 133801)[59], by truncating the AcrIIA4 cassette using the *NotI* and *AgeI* sites.

For the CRISPRi experiments, the [5C]–[30C] linkers including *BsmBI* sites were inserted into the *BsmBI* sites of an LV hU6-sgRNA hUbC-dCas9-KRAB-T2a-Puro (sgRNA-KRAB-Puro) plasmid (Addgene, 71236)[60] to construct [C]sgRNA-expressing all-in-one CRISPRi plasmids. The sgRNA spacers targeting *BRCA1* and *CXCR4* used in previous studies[61] were inserted into the *BsmBI* sites of the all-in-one plasmids. A puromycin-selectable all-in-one plasmid for CRISPRa was constructed by replacing a GFP cassette of a pLV hU6-gRNA(anti-sense) hUbC-VP64-dCas9-VP64-T2A-GFP (sgRNA-VP64-GFP) plasmid (Addgene, 66707) with a puromycin *N*-acetyl transferase (PuroR) cassette. A synthetic gene encoding VP64-T2A-PuroR (AZENTA) (Supplementary Table 3) was inserted into the sgRNA-KRAB-GFP plasmid using *NheI* and *AgeI* sites, resulting in an sgRNA-VP64-Puro plasmid. In Fig. 4e, the [1C]–[10C] spacer linkers for targeting *ASCL1*[62] were inserted into the sgRNA-VP64-Puro plasmid. In Extended Data Fig. 4e, spacer linkers for targeting *ASCL1* and *TTN*[62] were inserted into the *BpiI* sites of the [0C]–[30]sgRNA-expressing plasmids, and then they were co-transfected with the spacerless all-in-one CRISPRa plasmid.

To construct all-in-one AsCpf1 plasmids enabling puromycin selection, a synthetic DNA fragment encoding U6 promoter and two *BpiI* sites (AZENTA) (Supplementary Table 3) was inserted into a PX459 plasmid while removing a U6-gRNA cassette using *PciI* and *XbaI* sites. Next, a CBh-Cas9 region of the crRNA-Cas9-puro plasmid was replaced with a CBh-AsCpf1 fragment digested from a pY036_ATP1A1_G3_Array plasmid (Addgene, 86619)[63] using *KpnI* and *FseI*, resulting in the construction of an all-in-one crRNA-AsCpf1-puro plasmid (PX459 plasmid backbone). The crRNA linkers (Supplementary Table 3) targeting P2A$_2$ sites of AIMS are composed of 5′ hairpin, 20 nt-spacer and U$_4$AU$_4$ 3′-overhang, which is known to increase editing efficiency of AsCpf1 (ref. 64), and they were inserted into the *BpiI* sites of the crRNA-AsCpf1-puro plasmid.

pSpCas9(BB)-2A-Puro (PX459) V2.0 (Addgene, plasmid 62988; http://n2t.net/addgene:62988; RRID: Addgene_62988) and pSpCas9(BB)-2A-GFP (PX458) (Addgene, plasmid 48138; http://n2t.net/addgene:48138; RRID: Addgene_48138) were gifts from Feng Zhang. The pY036_ATP1A1_G3_Array was a gift from Yannick Doyon (Addgene, plasmid 86619; http://n2t.net/addgene:86619; RRID: Addgene_86619). pLV hU6-sgRNA hUbC-dCas9-KRAB-T2a-Puro was a gift from Charles Gersbach (Addgene, plasmid 71236; http://n2t.net/addgene:71236; RRID: Addgene_71236). pLV hU6-gRNA(anti-sense) hUbC-VP64-dCas9-VP64-T2A-GFP was a gift from Charles Gersbach (Addgene, plasmid 66707; http://n2t.net/addgene:66707; RRID: Addgene_66707). pCMV-T7-AcrIIA4-NLS(SV40) (KAC200) was gifted by Joseph Bondy-Denomy and Benjamin Kleinstiver (Addgene, plasmid 133801; http://n2t.net/addgene:133801; RRID: Addgene_133801)[59].

## Gel shift assay

To detect sgRNAs complexed with Cas9, 1 μl of Cas9 (1 μM) (Alt-R S.p. Cas9 Nuclease V3, IDT) and 1 μl of synthetic sgRNAs (3 μM, 1 μM or 0.3 μM; IDT) were mixed with 8 μl of distilled water (total reaction volume of 10 μl) and reacted on ice for 30 min. Samples were loaded onto Bullet PAGE One Precast gels (6%) (Nacalai Tesque) in Tris-borate-ethylenediaminetetraacetic acid (Tris-Borate-EDTA) buffer. RNA was transferred to a Hybond N+ membrane (GE Healthcare) and cross-linked using CX-2000 (Analytik Jena). An sgRNA tracer probe was labelled with an alkali-labile digoxigenin (DIG)-11-deoxyuridine triphosphate (dUTP) using a PCR DIG Probe Synthesis kit (Roche); DNA fragments were amplified using PCR and primers (Supplementary Table 3). After hybridization, specific bands were visualized with the CDP-Star reagent (Roche) using a luminescent image analyser (LAS-3000, FUJIFILM).

To detect DNA fragments complexed with sgRNA-dCas9, we mixed 1 μl of dCas9 (1 μM) (Alt-R S.p. dCas9 Nuclease V3, IDT) and 1 μl of synthetic sgRNAs (1 μM; IDT) with distilled water for a final reaction volume of 10 μl, then reacted the mixture at room temperature for 10 min. After the reaction, the RNP complex was mixed with 100 ng

of DNA fragment and 1 μl of 10× Cas9 reaction buffer (1 M HEPES, 3 M NaCl, 1 M MgCl$_2$ and 250 mM EDTA (pH 6.5)), then reacted at room temperature for 10 min. The resulting 10 μl samples were loaded onto 2% agarose gels in Tris-acetate-EDTA buffer; DNA bands were detected by staining with ethidium bromide. The target DNA fragment (647 bp) was prepared by PCR amplification from a Tbx3-P2A$_1$-Venus KI plasmid using primers (Supplementary Table 3).

## In vitro DNA cleavage assay

The sgRNA-Cas9-DNA complex was formed using most of the gel shift assay procedure, although its formation also included Cas9 and 3 μM of synthetic sgRNA. The samples were reacted at 37 °C for 90 min, denatured at 70 °C for 10 min and loaded onto Bullet PAGE One Precast gels (6%) (Nacalai Tesque).

## In vitro uncleaved DNA detection assay

A 20 μl sgRNA-Cas9-DNA complex was prepared via the procedure used in the gel shift assay. A cleavage reaction was performed at 37 °C for 30 min; a 10 μl volume was kept on ice while the other 10 μl volume was denatured at 70 °C for 10 min. The products were loaded onto 2% agarose gels.

## Northern blotting

Total RNAs were extracted from mESCs at 68 h after transfection with P2A$_1$-[C]sgRNA1-PX459 plasmids. Transfected cells were selected by 2 d of treatment with puromycin (1.5 μg ml$^{-1}$), then resuspended with ISOGEN II (NIPPON GENE). The samples were incubated for 10 min at room temperature, then heated at 55 °C for 10 min. Total RNA was isolated following the manufacturer's protocol. After reaction at 70 °C for 10 min, 30 μg RNAs were loaded onto Extra PAGE One Precast gels (5–20%) (Nacalai Tesque) in Tris-borate-EDTA buffer. RNA transfer, DIG-probe hybridization and signal detection were performed following the procedure used in the gel shift assay. The DIG probe was labelled by PCR amplification of the DNA fragment (primers shown in Supplementary Table 3). The mU6 DIG-probe was prepared by amplifying the DNA fragment from mESC complementary DNA using specific primers (Supplementary Table 3). cDNA was synthesized using a specific primer that targeted U6 small nuclear RNA[65].

## In vitro transcription (IVT)

Template DNA fragments required for IVT were amplified from a P2A$_1$-gRNA1-PX459 plasmid by PCR (primers shown in Supplementary Table 3). The T7 promoter sequence and cytosine tails were added to the 5′-end of the forward primer. We synthesized [0C], [10C] and [25C] sgRNAs using the T7 RiboMAX Express large-scale RNA production system (Promega) following the manufacturer's protocol.

## Image analysis

FIJI software was used to quantify band signals for the gel shift, DNA cleavage and northern blot assays.

## Indel analysis in hADSCs

The PX458-based all-in-one plasmids (250 ng) for targeting *VEGFA1* gene were transfected into hADSCs using Lipofectamine 3000 upon 80% confluency. Immediately after adding the plasmid:Lipofectamine mixture into the cells, the plates were centrifuged at 700 *g* at 35 °C for 10 min to increase transfection efficiency. The cells were cultured for 7 d without passaging to allow continuous expression of the plasmid, and then GFP-positive single cells were picked using a hand-made capillary and transferred to PCR tubes (1 cell per tube). To enable sequence analysis for a pair of alleles from a single cell, whole genomic DNA were amplified using PicoPLEX (TAKARA) according to the manufacturer's instructions. The genomic locus targeted by Cas9 was amplified by PCR using primers (Supplementary Table 3) and the PCR amplicons were sequenced.

## Cas9 inhibition

At 24 h after transfection with the all-in-one Cas9 plasmid, mESCs were treated with the Cas9 inhibitor BRD0539 (TOCRIS) during puromycin selection and subsequent culture until analysis.

pCMV+AcrIIA4 plasmid was co-transfected with 250 ng of the all-in-one Cas9 plasmid in different amounts (2.5–2,500 ng for 24-well plates). For the BRD0539 and AcrIIA4 experiments, puromycin selection and indel analysis were performed using the same procedure as described above ('Generation of AIMS cell lines and mice and AIMS analysis' section) and in Fig. 1d.

## CRISPR activation and interference

A day before transfection, $3 \times 10^4$ HEK293T cells were seeded onto a 96-well plate. The all-in-one CRISPRa/i plasmids (50 ng, 1/5 scale of the 24-well plate version) were transfected and cultured for 24 h. Then, puromycin (5.0 µg ml$^{-1}$) was treated for 2 d to exclude untransfected cells. After removal of puromycin, the transfected cells were cultured for 1 d and 2 d for CRISPRa and CRISPRi, respectively, and total RNAs were extracted using ISOGEN II as described above ('Northern blotting' section).

## RT–qPCR analysis

The cDNAs were synthesized from total RNAs using SuperScript III Reverse Transcriptase (Thermo Fisher) according to the manufacturer's instructions. RT–qPCR was conducted using a THUNDERBIRD SYBR qPCR Mix (Toyobo) and CFX Connect real-time PCR detection system (BIO RAD) according to the manufacturer's instructions. Primers for *ASCL1, TTN, BRCA1* and *CXCR4* used in previous studies[61,62], and for *GAPDH* are listed in Supplementary Table 3. The values for *GAPDH* were used as normalization controls.

## Scarless mono-allelic KI of tdTomato or P2A$_1$-Neo cassette

A Tbx3-P2A$_1$-tdTomato KI plasmid was co-transfected with Tbx3-sgRNA1-expressing PX459 to the mESCs. After transient puromycin selection, colonies were dissociated and passaged; the resulting colonies were analysed. Colonies with mosaic tdTomato expression were excluded from data analysis. After the colonies had been counted, positive tdTomato colonies were selected and genomic DNA was extracted for sequencing.

The neomycin (Neo) KI plasmid was constructed by replacing the tdTomato cassette of the Tbx3-P2A$_1$-tdTomato KI plasmid with a P2A$_1$-Neo cassette. The KI plasmid was co-transfected with P2A$_1$ sgRNA1-expressing PX459 to a Tbx3-P2A$_1$-AIMS clone. When puromycin was removed, geneticin (400 µg ml$^{-1}$, Gibco) was added to select KI clones. All eight clones were confirmed to possess KI genotypes; geneticin-resistant colonies were identified as KI.

## T7E1 assays

PCR reactions to amplify specific on-target or off-target sites were performed using KOD-Plus-ver.2 DNA polymerase (Toyobo) in accordance with the manufacturer's protocol. The resulting PCR amplicons were denatured and re-annealed in 1× NEB buffer 2 (NEB) in a total volume of 9 µl under the following conditions: 95 °C for 5 min, reduction from 95 °C to 25 °C at a rate of −0.1 °C s$^{-1}$ and indefinite incubation at 4 °C. After re-annealing had been performed, 1 µl of T7 endonuclease I (NEB, 10 U µl$^{-1}$) was added and the product was incubated at 37 °C for 15 min.

## Bac[P] assays

Purified PCR products to amplify specific on-target or off-target sites were inserted into a T-easy vector (Promega) and transformed into DH5-α bacterial cells. For rapid and efficient indel detection, plasmids were directly isolated from each white colony after blue/white screening; the inserted DNA fragment was amplified by PCR. The PCR amplicons were mixed with PCR products amplified from a WT DNA template such as KI plasmid or unedited genomic DNA; a T7E1 assay was then performed. Sanger sequencing was also performed for PCR amplicons that were not digested by T7E1 to determine the total number of colonies that harbour indels. The Bac[P] value was calculated as follows: Bac[P] = Indel/Total.

Bac[P] values for both WT and R206H alleles were determined through indel induction experiments using various [C]sgRNAs in the mESC clone of the FOP model. The targeting sites of both WT and R206H alleles were amplified by PCR, then cloned into a T-easy vector. Sanger sequencing was performed for each PCR product that had been derived from single bacterial clones, as described above. Similarly, Bac[P] values for both R206H (pf) and WT (1 mm) alleles were determined by inducing indels in FOP hiPSCs; a corrected cell line (WT/Corrected) was used to determine the Bac[P] value of the corrected allele (2 mm). Some PCR products did not contain a G/A hallmark because of intermediate-sized deletions (12–50 nucleotides); it was therefore impossible to determine which allele was edited for these PCR products. We observed that the fraction of such products with intermediate-sized deletions was generally constant (~20% in experiments shown in Fig. 6 and 10–20% in experiments shown in Fig. 7) and did not decrease with [C] extension, suggesting that such intermediate-sized deletions are byproducts of the short indel induction processes. Therefore, we assigned products with intermediate-sized deletions to two alleles using the ratio of PCR products with convincingly confirmed origins. For the analysis shown in Fig. 7, we calculated the means of Bac[P] for WT (1 mm) alleles on the basis of comparisons of R206H (pf) to WT (1 mm) alleles and WT (1 mm) to corrected (2 mm) alleles for subsequent computational analyses.

## Cell viability assays

Using the transfection protocol described above ('Generation of AIMS cell lines and mice and AIMS analysis'), $2 \times 10^5$ WT hiPSCs or $4 \times 10^4$ HEK293T cells were seeded onto 48-well plates and transfected with 100 ng of all-in-one CRISPR plasmids (2/5 scale of the 24-well plate version). hiPSCs were dissociated and counted using trypan blue at 3 or 4 d after transient puromycin treatment (1.5 µg ml$^{-1}$); HEK293T cells were counted at 4 d after transient puromycin treatment (3 µg ml$^{-1}$). The data obtained by this procedure are indicated as 'Cell number' in the Figures.

Biochemical assays were also performed using Cell Count Reagent SF reagent according to the manufacturer's instructions (Nacalai Tesque). The Cdh1-P2A1-AIMS mESCs ($2 \times 10^4$ cells) were seeded onto 96-well plates and transfected with 50 ng of all-in-one plasmids (1/5 scale of the 24-well plate version). Two days after puromycin selection, absorbance at 450 nm was measured by Multiskan FC (Thermo Fisher). The data obtained from the biochemical assay are indicated as 'Cell viability (%)' in Fig. 4d and Extended Data Fig. 4c by setting the data for [0C] and 0 mM as a reference value (1.0), respectively.

For the AcrIIA4 experiments (Fig. 4c and Extended Data Fig. 4b), the Cdh1-P2A1-AIMS mESCs ($3 \times 10^4$ cells) were seeded onto 96-well plates and 50 ng of all-in-one plasmids were co-transfected with different amounts of pCMV+AcrIIA4 and/or pCMVΔAcrIIA4 plasmids (1/5 scale of the 24-well plate version). In Fig. 4c and Extended Data Fig. 4b, we observed cytotoxicity for higher doses of AcrIIA4 expression plasmids. Similar cytotoxicity profiles were obtained in the absence of the Cdh1-P2A$_1$-sgRNA1 target sequence in WT mESCs.

## Generation and correction of the FOP model via HDR with ssODNs

The transfection protocol for the 24-well plate experiment was performed as described above ('Generation of AIMS cell lines and mice and AIMS analysis'). For HDR induction in mESCs, WT hiPSCs and HEK293T cells, 1 µl of 10 µM ssODN (Eurofins) was added to the plasmid–Lipofectamine complex; for hiPSC transfection, 1 µl of 3 µM ssODN was added because a concentration of 10 µM induced severe toxicity. After transient puromycin selection, colonies were dissociated and plated at low density to avoid mosaicism. Single colonies

were selected and genomic DNA was extracted. Sequence analysis was performed to identify G to A replacement with or without indels. To correct the FOP hiPSCs, clones that underwent HDR were screened by digesting the PCR product using the *BstUI* restriction enzyme (NEB); *BstUI*-positive PCR products were then sequenced. A silent mutation was inserted into the ssODN to generate the *BstUI* site and to distinguish an HDR-corrected (Corrected) allele from an original WT allele. Without this hallmark, WT/– clones, in which PCR amplicons from the R206H allele cannot to be obtained because of large deletions or more complex genomic rearrangement, would be misidentified as WT/Corrected clones.

## Immunocytochemical analysis

For p53 staining, we performed transfection for HDR induction (1/5 scale of the 24-well plate version), using the protocol described above. In this assay, $6 \times 10^4$ hiPSCs were seeded on a Matrigel-coated 96-well plate in triplicate. Puromycin selection was performed to examine p53 activity solely in transfected cells. The surviving cells were fixed with 4% paraformaldehyde at 2 d after puromycin removal. For pSmad1/5/8 staining, $5 \times 10^3$ cells were plated on a Matrigel-coated 96-well plate without Y-27632 and with 1% FBS. After 2.5 h of culture, activin-A (100 ng ml$^{-1}$) (R & D Systems) was administered for 30 min; cells were fixed with 4% paraformaldehyde. Antibody reactions were performed in accordance with standard protocols. Rabbit polyclonal p53 (FL-393, Santa Cruz, 1:200) and rabbit monoclonal pSmad1/5/8 (D5B10, Cell Signaling Technology, 1:1,000) antibodies were reacted overnight at 4 °C. Donkey anti-rabbit Alexa Fluor 488 secondary antibody (Thermo Fisher, 1:1,000) was reacted at room temperature for 30 min. Data analysis was performed using a cell count application associated with a fluorescent microscope to select cells with p53 and pSmad1/5/8 activation by means of fluorescence intensity thresholds (BZ-X800, Keyence).

## Chimaera generation for FOP model

An mESC clone of an FOP model ($R26R^{YFP/+}$ mESC line) was dissociated with trypsin and 5–8 cells were injected into 8-cell embryos (E2.5) collected from pregnant ICR mice. Injected blastocysts were transferred into the uteri of pseudo-pregnant ICR mice. Chimaeric contribution was confirmed by coat colour and YFP fluorescence. YFP was observed using a fluorescence stereo microscope (M165FC, Leica).

## Computational modelling and analysis of [C] extension, single-cell-level genome editing and HDR efficiency
**Determination of AIMS[P] and effects of [C] extension on the CRISPR-Cas9 system.** In this study, the probability of single-allele editing (P) was determined using AIMS and a Bac[P] assay, on the basis of a T7E1 assay, complemented by sequence validation. AIMS-based P (AIMS[P]) was determined as follows:

$$\text{AIMS}[P] = \frac{(2F(\text{Bi}) + F(\text{Mono}))}{2} \quad (1)$$

where $F$(Bi) and $F$(Mono) are the experimental frequencies of cells with bi-allelic and mono-allelic genome editing, respectively.

The efficiency of the single-allele editing P (P(*pf*), where pf denotes perfect match) can be described as follows:

$$P(pf) = \frac{S}{K + S} \quad (2)$$

where the concentration of effective sgRNA-Cas9 complexes and the dissociation constant between the sgRNA and its target site are defined as $S$ and $K$, respectively. On the basis of high editing efficiency without [C] extension (P = approximately 1), we assumed that the recovery rate from single-site damage was very low; therefore, it was neglected in subsequent analyses. To mechanistically understand the effects of [C] extension and 1 mm, we assumed that [C] extension and 1 mm decreased $S$ and increased $K$, respectively. By setting $S = 1$ for each sgRNA sequence

without [C] extension, we approximated $K$ values for each of eight sgRNA sequences. When P (AIMS[P] or Bac[P]) was 1, P was set to 0.99. Next, the relative $S$ concentrations were determined using $K$ and AIMS[P] for sgRNAs with [C] extension. Despite variation in the relationships between [C] extension and AIMS[P] among sgRNA sequences (Fig. 2f), we found clear and similar inverse relationships between [C] extension and relative S values for different sgRNA sequences (Extended Data Fig. 3d). Linear regression analysis demonstrated a good fit for the logarithm of the ratio of $S$ to the length of [C] extension for all sgRNA sequences (Fig. 2g). Analysis of covariance (ANCOVA) indicated that the linear regression slopes did not significantly differ among various sgRNA sequences (Fig. 2h). This finding suggests that [C] extension exerts uniform suppression effects on diverse sgRNA sequences.

Since we observed that [C] extension modestly decreased target cleavage (Fig. 3c), we also performed similar analysis by gradually increasing $K$ according to the length of [C] extension and observed that [C] extension gradually decreases $S$ in a similar manner. In this setting, the effects on $S$ became weaker. However, we observed that the dynamic range of suppression in northern blot analysis (Fig. 3f, ~6,000-fold change at [30C]) was more comparable to the range of change in $S$ with constant $K$ (~2,000-fold change at [30C]) relative to the range of change in $S$ with increased $K$ (~400-fold and 200-fold change with 5-fold and 10-fold increases in $K$ at [30C], respectively). Therefore, this suggests that the effects on complex formation may be dominant, allowing determination of the single-allele editing probability in the cells.

**Comparison of AIMS[P] and Bac[P].** In the initial phase of this study, we compared matched AIMS[P] and Bac[P] values for nine sgRNAs (that is, Cdh1-P2A1-sgRNA1 with different [C] extension lengths) and observed that AIMS[P] was strongly correlated with Bac[P] (Extended Data Fig. 5a). In our subsequent analyses, we used AIMS[P] to model indel insertion frequency (Figs. 2 and 5, and Extended Data Fig. 6) and Bac[P] to model HDR frequency (Figs. 6 and 7).

**Error frequencies of AIMS[P] and T7E1-based Bac[P].** AIMS error was calculated as the difference between raw AIMS[P] and adjusted AIMS[P] (adjusted AIMS[P] – AIMS[P]) (Fig. 1h). The raw AIMS[P] is simply based on fluorescence patterns. Therefore, in Fig. 1e, rare tdTomato$^+$/Venus$^{indel}$ and tdTomato$^{indel}$/Venus$^+$ heterozygous clones were grouped into mono-allelic clones. To determine the exact number of bi-allelic indel clones, these ostensibly heterozygous clones were analysed for sequencing (Seq-indel data). When sequencing these clones, most (86%) of these ostensibly heterozygous clones turned out to be homozygous. Adjusted AIMS[P] incorporates Seq-indel data together with fluorescence patterns. In most analyses, we used raw AIMS[P].

T7E1 error was calculated as Bac[P] – T7E1:Bac[P] (Fig. 1i,j). T7E1:Bac[P] is the indel probability calculated from the rate of T7E1 sensitive clones, while Bac[P] is the indel probability calculated considering the Seq-indel data. The Seq-indel data were the exact numbers of indel clones that were not digested by T7E1, as determined by sequencing PCR products.

**Genome editing frequency modelling at the single-cell level.**
We performed extensive analyses using a combination of AIMS and sgRNAs with various types of [C] extensions. When editing efficiency was homogeneous across the cell population, we estimated the frequencies of cells with bi-allelic, mono-allelic or no genome editing (that is, $F$(Bi), $F$(Mono) or $F$(No)) as follows:

$$F(\text{Bi}) = \text{AIMS}[P]^2 \quad (3)$$

$$F(\text{Mono}) = 2\text{AIMS}[P](1 - \text{AIMS}[P]) \quad (4)$$

$$F(\text{No}) = (1 - \text{AIMS}[P])^2 \quad (5)$$

Using these equations, we observed that actual $F$(Mono) was lower than estimated $F$(Mono), particularly at intermediate AIMS[P] levels (AIMS[P] = ~0.5). Therefore, we considered genome editing frequency heterogeneity at the single-cell level, which we modelled using a beta distribution. The probability density functions of P and mean P ($E$(P)) were calculated as follows:

$$f(P; \alpha, \beta) = \frac{P^{\alpha-1}(1-P)^{\beta-1}}{B(\alpha, \beta)} \tag{6}$$

$$E(P) = \frac{\alpha}{\alpha + \beta} \tag{7}$$

where the mean P corresponds to AIMS[P] (or Bac[P]) and $\alpha$ and $\beta$ are exponents of P and its complement to 1. Using the beta distribution, $F$(Bi), $F$(Mono) and $F$(No) were described as follows:

$$F(\text{Bi}) = \int_0^1 P^2 f(P; \alpha, \beta) \, dP \tag{8}$$

$$F(\text{Mono}) = \int_0^1 2P(1-P) f(P; \alpha, \beta) \, dP \tag{9}$$

$$F(\text{No}) = \int_0^1 (1-P)^2 f(P; \alpha, \beta) \, dP \tag{10}$$

Using these equations, we determined $\alpha$ values for each experiment that minimized the squared residuals between experimental $F$(Bi), $F$(Mono) and $F$(No), and simulated $F$(Bi), $F$(Mono) and $F$(No) (Extended Data Fig. 5b). As shown in Extended Data Fig. 5b, we observed that optimized $\alpha$ values were generally constant for a wide range of AIMS[P] (0.1 < AIMS[P] < 0.9). Therefore, we used the sum of squared residuals (SSR) as the error function, calculated as follows: SSR = $\sum$(Experimental data – Simulated data)$^2$; we determined a constant $\alpha$ value that minimized SSR (Extended Data Fig. 5c, left, $\alpha$ = 0.715). Probability density functions with different mean P values are shown in Extended Data Fig. 5c (middle). The application of the beta distribution greatly reduced SSRs compared with a homogeneous editing frequency (Extended Data Fig. 5c, right); it adequately explained the experimental $F$(Bi), $F$(Mono) and $F$(No) for diverse AIMS[P] values (Extended Data Fig. 5d). We also tested the normal distribution to approximate genome editing frequency heterogeneity at the single-cell level; we found that the beta distribution was superior to the normal distribution.

**Effect of cytosine extension on CRISPR-Cas9 system specificity.** As described above, 1 mm (or 2 mm) increases $K$ in equation (2). The efficiency of the single-gene editing P on the 1 mm (or 2 mm) target can be described as follows:

$$P(1\text{mm } or \text{ 2mm}) = \frac{S}{mK + S} \tag{11}$$

where $m$ is the ratio of $K$ for the 1 mm target to $K$ for the perfect match target. Thus, the single-gene editing P for 1 mm (or 2 mm) can be expressed as the function of P($pf$), as follows:

$$P(1\text{mm } or \text{ 2mm}) = \frac{P(pf)}{(1-m)P(pf) + m} \tag{12}$$

For the results shown in Figs. 6 and 7, we determined values of $m$ that fit P($pf$) and P(1 mm or 2 mm), using SSR as the error function (Fig. 6g). The ratios of P($pf$) and P(1 mm or 2 mm) can also be described as functions of P($pf$), as follows:

$$\frac{P(1\text{mm } or \text{ 2mm})}{P(pf)} = \frac{1}{(1-m)P(pf) + m} \tag{13}$$

$$\frac{P(pf)}{P(1\text{mm } or \text{ 2mm})} = (1-m)P(pf) + m \tag{14}$$

As shown in Fig. 6h, decreasing P($pf$) contributes to the reduction in relative off-target ratio and enhancement of specificity. Thus, reduction in CRISPR-Cas9 activity through [C] extension is beneficial for reducing the relative off-target activity and enhancing specificity.

**Modelling HDR frequency for homozygous states.** Using the beta distribution, the frequencies of the various HDR clones shown in Fig. 6 were determined as follows (Extended Data Fig. 7c,d):

$$F(\text{WT/R206H}) = \int_0^1 2hP(1-P)(1-(1-h)P')f(P; \alpha, \beta) \, dP \tag{15}$$

$$F(\text{WT/R206H + indel}) = \int_0^1 2h(1-h)P(1-P)P'f(P; \alpha, \beta) \, dP \tag{16}$$

$$F(\text{indel/R206H}) = \int_0^1 2h(1-h)P^2(1-(1-h)P')f(P; \alpha, \beta) \, dP \tag{17}$$

$$F(\text{indel/R206H + indel}) = \int_0^1 2h(1-h)^2 P^2 P' f(P; \alpha, \beta) \, dP \tag{18}$$

$$F(\text{R206H/R206H}) = \int_0^1 h^2 P^2 f(P; \alpha, \beta) \, dP \tag{19}$$

$$F(\text{overall HDR}) = \int_0^1 (-h^2 P^2 + 2hP) f(P; \alpha, \beta) \, dP \tag{20}$$

where the efficiency of HDR on the Cas9-cleaved single allele is defined as $h$. The probability of single-gene editing on the edited (that is, 1 mm) target is P$'$ (Extended Data Fig. 7d), which is described in a manner similar to equation (12), as follows:

$$P' = \frac{P}{(1-m)P + m} \tag{21}$$

where $m$ = 1.723. P is decreased according to the [C] extension length (Extended Data Fig. 7e).

For simplicity, we considered $h$ to be constant across the cell population in each experiment. On the basis of the experimental overall HDR frequency results and equation (20), we estimated $h$ for each [C] extension (Fig. 6f). Although $h$ was very low for sgRNAs without [C] extension (2.07%), $h$ for sgRNAs with [C] extension was generally high (~11%). This result suggests that the conventional system without [C] extension suppresses HDR; [C] extension releases this suppression to allow HDR to reach its upper limit. On the basis of these findings, we used the mean estimated $h$ (10.99%) for [C]-extended sgRNAs; we estimated the frequencies of distinct HDR patterns, overall HDR and precise HDR (Fig. 6i,j). For sgRNAs without [C] extension, we used the estimated $h$ (2.07%). The simulated data adequately fit the experimental results (Fig. 6i–k). To predict continuous HDR outcomes, we designed a hypothetical function for $h$ for the range of P, such that $h$ = 2.07% for P > 0.9 and $h$ = 10.99% for P < 0.9 (Extended Data Fig. 7f); we estimated the frequencies of distinct HDR patterns, overall HDR and precise HDR (Extended Data Fig. 7g). In the simulation, precise HDR reached a maximum at P = 0.313 (Extended Data Fig. 7e,g).

**Modelling of HDR-based gene correction.** Using the beta distribution, we determined the frequencies of the various HDR clones shown in Fig. 7 as follows (Extended Data Figs. 7c and 9a):

$$F(\text{WT/R206H\_Corrected}) = \int_0^1 h P(1-P')(1-P'') f(P;\alpha,\beta)\, dP \quad (22)$$

$$F(\text{WT\_indel/R206H\_Corrected}) = \int_0^1 h(1-h) PP'(1-P'') f(P;\alpha,\beta)\, dP \quad (23)$$

$$F(\text{WT\_Corrected/R206H\_Corrected}) = \int_0^1 h^2 PP'(1-P'')^2 f(P;\alpha,\beta)\, dP \quad (24)$$

$$F(\text{WT\_Corrected\_indel/R206H\_Corrected}) = \int_0^1 h^2 PP'P''(1-P'') f(P;\alpha,\beta)\, dP \quad (25)$$

$$F(\text{WT/R206H\_Corrected\_indel}) = \int_0^1 h P(1-P')P'' f(P;\alpha,\beta)\, dP \quad (26)$$

$$F(\text{WT\_indel/R206H\_Corrected\_indel}) = \int_0^1 h(1-h) PP'P'' f(P;\alpha,\beta)\, dP \quad (27)$$

$$F(\text{WT\_Corrected/R206H\_Corrected\_indel}) = \int_0^1 h^2 PP'P''(1-P'') f(P;\alpha,\beta)\, dP \quad (28)$$

$$F(\text{WT\_Corrected\_indel/R206H\_Corrected\_indel}) = \int_0^1 h^2 PP'P''^2 f(P;\alpha,\beta)\, dP \quad (29)$$

$$F(\text{WT\_Corrected/R206H}) = \int_0^1 h(1-P)P'(1-P'') f(P;\alpha,\beta)\, dP \quad (30)$$

$$F(\text{WT\_Corrected\_indel/R206H}) = \int_0^1 h(1-P)P'P'' f(P;\alpha,\beta)\, dP \quad (31)$$

$$F(\text{WT\_Corrected/R206H\_indel}) = \int_0^1 h(1-h) PP'(1-P'') f(P;\alpha,\beta)\, dP \quad (32)$$

$$F(\text{WT\_Corrected\_indel/R206H\_indel}) = \int_0^1 h(1-h) PP'P'' f(P;\alpha,\beta)\, dP \quad (33)$$

$$F(\text{overall HDR}) = \int_0^1 \left(-h^2 PP' + hP + hP'\right) f(P;\alpha,\beta)\, dP \quad (34)$$

where the efficiency of HDR on Cas9-cleaved single alleles and the probability of single-gene editing on a WT or HDR-corrected (that is, 1 mm or 2 mm) target is defined as $h$ and $P'$ or $P''$, respectively (Extended Data Fig. 9a). $P'$ and $P''$ are described in a manner similar to equation (12), as follows:

$$P' = \frac{P}{(1-m_1)P + m_1} \quad (35)$$

$$P'' = \frac{P}{(1-m_2)P + m_2} \quad (36)$$

where $m_1$ is 3.459 and $m_2$ is 12.0793 (Fig. 6g). The relationship between [C] extension and P is shown in Fig. 7c and Extended Data Fig. 9b.

For simplicity, we assumed $h$ to be constant across the cell population in each experiment; moreover, the HDR rate was presumed to be identical on both R206H and WT alleles. On the basis of the experimental overall HDR frequency results and equation (34), we estimated $h$ for each [C]-extended sgRNA (Fig. 7h). Consistent with the results shown in Fig. 6, $h$ was higher for sgRNAs with [C] extension than for sgRNAs without [C] extension. Together with the results shown in Fig. 6, these findings suggest that the conventional system without [C] extension reduces HDR, probably because of extensive DNA damage and p53 response, as shown in Fig. 7d,e; furthermore, [C] extension releases HDR suppression to allow it to reach its upper limit. Notably, $h$ was generally higher in Fig. 7 than in Fig. 6, perhaps because the cell lines used for the experiment shown in Fig. 7 had only one perfect-match target and thus elicited a weaker suppressive effect on the HDR rate; cell lines used in the experiment shown in Fig. 6 had two such targets. On the basis of these findings, we used the mean of the estimated $h$ (26.93%) for the [C]-extended sgRNAs and estimated the frequencies of overall HDR and precise HDR (Fig. 7i and Extended Data Fig. 9c). For sgRNAs without [C] extension, we used the estimated $h$ (13.21%). The simulated data adequately fit the experimental results (Fig. 7i and Extended Data Fig. 9c). To predict continuous HDR outcomes, we designed a hypothetical function of $h$ for the range of P, such that $h = 13.21\%$ for P > 0.9 and $h = 26.93\%$ for P < 0.9 (Extended Data Fig. 9d); we estimated the frequencies of distinct HDR patterns, overall HDR and precise HDR (Fig. 7j and Extended Data Fig. 9e). In the simulation, precise HDR reached a maximum at P = 0.424 (Extended Data Fig. 9b,e).

**Statistics.** Sample sizes were determined on the basis of our previous experience of performing similar sets of experiments. Statistical tests were performed using JMP v14.2.0 and R v3.2.1 softwares. We verified the equality of variance assumption using the $F$-test or Levene test. As a pre-test for normality, we used the Kolmogorov–Smirnov test. Differences between two groups were analysed using two-tailed Student's $t$-test (Figs. 2e and 7g) or two-tailed Welch's $t$-test (Extended Data Fig. 6a, right). Comparisons among more than two groups were analysed using one-way or two-way analysis of variance (ANOVA), followed by a post hoc Tukey–Kramer test (Figs. 4c,d and 6d, and Extended Data Figs. 2c, 3a, 4b,c and 6c,d) or one-way Welch's ANOVA, followed by a post hoc Games–Howell test (Figs. 1g, 2a, 3b–e, 4e,f and 7d,e, and Extended Data Figs. 4e and 8b,d,f,h). In Fig. 2h, analysis of covariance (ANCOVA) was performed for the data points with [0C]–[30C]-extended sgRNAs. In bar graphs, data are expressed as means ± s.e.m. (Figs. 4c (AIMS[P]),d (AIMS[P]), 6d and 7f, and Extended Data Figs. 1d, 2c,d, 3a, 4b (AIMS[P]) and 6a,c,d) or means ± s.d. (Figs. 2a, 3b–e, 4c (Viability),d (Viability),e,f and 7d,e,g, and Extended Data Figs. 4b (Viability),c,e and 8b,d,f,h). In scatterplots, centre lines indicate the median and whiskers indicate the 25th and 75th percentiles (Fig. 1g). In boxplots, centre lines indicate the median, box limits indicate the 25th and 75th percentiles, and whiskers indicate the minimum and maximum (Fig. 2e).

**Reporting summary**

Further information on research design is available in the Nature Portfolio Reporting Summary linked to this article.

## Data availability

The main data supporting the results in this study are available within the paper and its Supplementary Information. All data generated in this study are available from the authors on reasonable request. Source data for the figures are provided with this paper.

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

## Acknowledgements
We thank F. Costantini for providing the *R26R*^EYFP/YFP mice; A. Miyawaki and H. Miyoshi for sharing reagents; Y. Honda, M. Udono, K. Inoue, N. Goto, M. Tasai, C. Kaieda, T. Kawawada, S. Abe, T. Ishiuchi, H. Sasaki, E. Koba, K. Kageyama and M. Tanaka for excellent technical assistance; and P. A. Sharp for discussion. This work was supported in part by JSPS KAKENHI grants (17K15046, 18H04737, 20H05041 and 21K19232 to M.K.; 19K24694 to H.I.S.; 18H05102, 19H01177, 19H05267, 20H05040, 21K19916, 22H05634, 22H04698 and 22H00592 to A.S.), the AMED (JP20ae0201012h to M.K.; JP21bm0704034h, JP22fk0210116h, and JP22bm1123005h to A.S.), the Center for Clinical and Translational Research of Kyushu University Hospital (to M.K.), the Kyushu University-initiated venture business seed development programme (GAP Fund) (to M.K.), the Fukuoka Financial Group Enterprise Development Foundation (KYUTEC) (to M.K.), the Industrial Support Organization consisting of Fukuoka Prefecture, Kurume City and Kurume Research Park (to M.K.), the Medical Research Center Initiative for High Depth Omics (to M.K. and A.S.), the Takeda Science Foundation (to M.K., H.I.S. and A.S.), the Uehara Memorial Foundation (to H.I.S. and A.S.), the Suzuken Memorial Foundation (to A.S.), the Mitsubishi Foundation (to H.I.S. and A.S.) and the Nagoya University Research Fund (to H.I.S.). Studies conducted by H.I.S. at the Massachusetts Institute of Technology were supported by United States Public Health Service grants R01-GM034277 and R01-CA133404 to P. A. Sharp and P01-CA042063 to T. Jacks from the NIH, as well as Koch Institute Support (core) grant P30-CA14051 from the National Cancer Institute, and supported in part by an agreement between the Whitehead Institute for Biomedical Research and Novo Nordisk.

## Author contributions
M.K. conceived and designed the research. M.K. and R.K. performed experiments. M.K. and H.I.S. analysed the data. H.I.S. designed and performed computational analysis. A.S. supervised the project. M.K., H.I.S. and A.S. wrote the manuscript. All authors read and approved the final manuscript.

## Competing interests
The authors declare no competing interests.

## Additional information
**Extended data** is available for this paper at https://doi.org/10.1038/s41551-023-01011-7.

**Correspondence and requests for materials** should be addressed to Masaki Kawamata, Hiroshi I. Suzuki or Atsushi Suzuki.

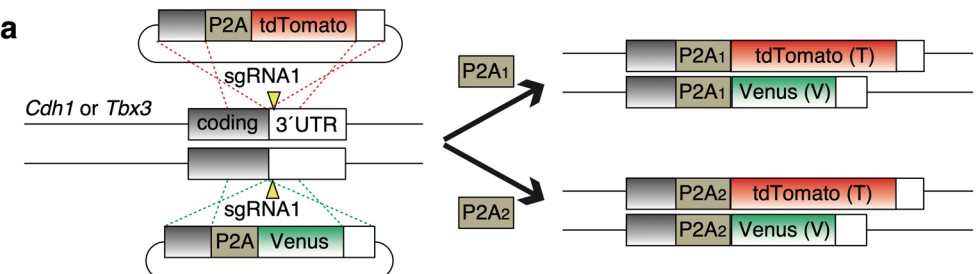

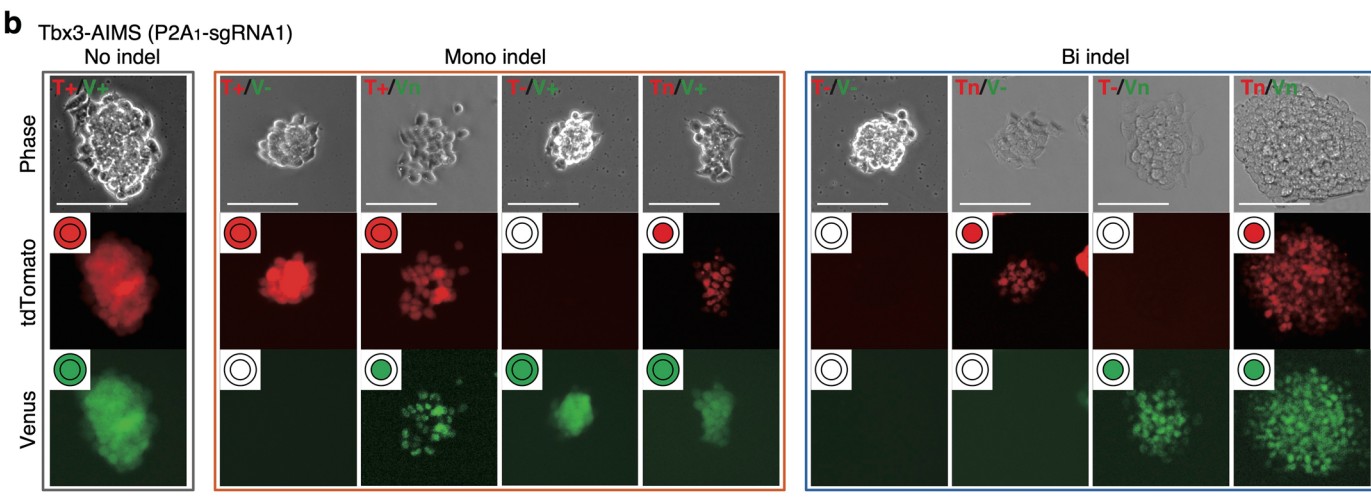

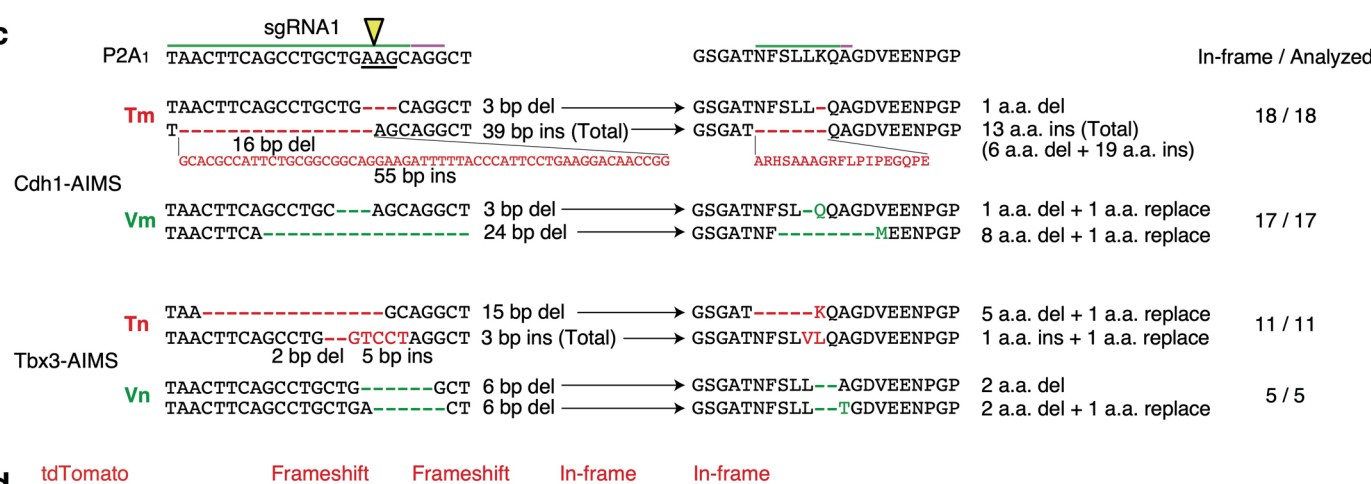

| tdTomato | Frameshift | Frameshift | In-frame | In-frame | | |
|---|---|---|---|---|---|---|
| Venus | Frameshift | In-frame | Frameshift | In-frame | (n) | |
| Tbx3-P2A₁-sgRNA1 | 81.8 ± 3.8 | 8.8 ± 1.9 | 6.3 ± 1.3 | 3.1 ± 0.7 | (%) | 3 |
| Cdh1-P2A₁-sgRNA1 | 73.7 ± 1.8 | 12.1 ± 0.5 | 13.1 ± 1.5 | 1.1 ± 1.1 | (%) | 3 |
| Cdh1-P2A₁-sgRNA2 | 64.2 ± 6.5 | 15.5 ± 1.3 | 15.2 ± 2.6 | 5.0 ± 2.8 | (%) | 3 |
| Tbx3-P2A₁-sgRNA3 | 58.7 ± 2.0 | 18.0 ± 5.0 | 15.6 ± 2.9 | 7.7 ± 3.5 | (%) | 3 |
| Cdh1-P2A₂-sgRNA4 | 47.7 ±11.6 | 18.3 ± 0.5 | 23.4 ± 7.9 | 10.6 ± 4.2 | (%) | 3 |
| Cdh1-P2A₂-sgRNA5 | 59.2 ± 1.9 | 17.6 ± 0.9 | 17.8 ± 1.5 | 4.7 ± 1.0 | (%) | 9 |
| Cdh1-P2A₂-sgRNA6 | 65.4 ± 1.9 | 14.6 ± 2.3 | 16.6 ± 2.0 | 3.3 ± 0.5 | (%) | 6 |
| Total (Fig. 1g) | 63.7 ± 2.1 | 15.5 ± 0.9 | 16.0 ± 1.2 | 4.8 ± 0.8 | (%) | 30 |

**Extended Data Fig. 1 | Allele-specific indel monitor system (AIMS) construction and indel analysis. a**, Schematic of the generation of dual-color knock-in (KI) mouse embryonic stem cell (mESC) clones for AIMS. Two types of targeting plasmids were simultaneously knocked into the two alleles of the *Cdh1* or *Tbx3* locus using CRISPR-Cas9. Pointers indicate double-strand break (DSB) sites. **b**, Results of Tbx3-P2A₁-AIMS in mESCs. Genotypes were determined according to nine combinations of tdTomato/Venus expression and localization in Tbx3-P2A₁-AIMS. T, tdTomato; V, Venus; +, no indel; n, in-frame indel indicated by nuclear localization; –, frameshift indel or large deletion indicated by loss of fluorescence. Scale bar = 100 μm. **c**, Representative DNA and amino acid sequences of in-frame indels. Clones (T + or V + ) were sequenced; the numbers of clones with in-frame indels are shown. **d**, Table shows the percentages of four types of bi-allelic indel patterns. Totals indicate means of all data (*n* = 30), which are shown in Fig. 1g. Data are expressed as means ± SEMs.

**a**

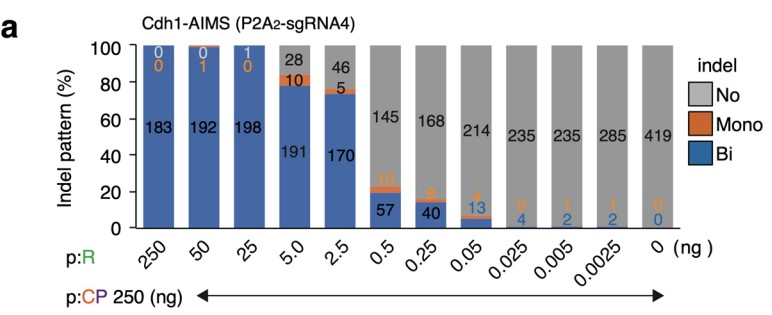

Cdh1-AIMS (P2A₂-sgRNA4)

**b**

sgRNA transcription

5C(3A) linker: 5'-CACCGCCACCGGGTCTTCGAGAAGACCT
                   CGGTGGCCCAGAAGCTCTTCTGGACAAA-5'

10C(8A) linker: 5'-CACCGCCCCCCACCGGGTCTTCGAGAAGACCT
                   CGGGGGGGGTGGCCCAGAAGCTCTTCTGGACAAA-5'

15C(13A) linker: 5'-CACCGCCCCCCCCCCCCACCGGGTCTTCGAGAAGACCT
                   CGGGGGGGGGGGGGGTGGCCCAGAAGCTCTTCTGGACAAA-5'

20C(18A) linker: 5'-CACCGCCCCCCCCCCCCCCCCCACCGGGTCTTCGAGAAGACCT
                   CGGGGGGGGGGGGGGGGGGGTGGCCCAGAAGCTCTTCTGGACAAA-5'

25C(23A) linker: 5'-CACCGCCCCCCCCCCCCCCCCCCCCCCCACCGGGTCTTCGAGAAGACCT
                   CGGGGGGGGGGGGGGGGGGGGGGGGTGGCCCAGAAGCTCTTCTGGACAAA-5'

30C(28A) linker: 5'-CACCGCCCCCCCCCCCCCCCCCCCCCCCCCCCCACCGGGTCTTCGAGAAGACCT
                   CGGGGGGGGGGGGGGGGGGGGGGGGGGGGTGGCCCAGAAGCTCTTCTGGACAAA-5'

**c**

P2A1-sgRNA1

Mosaicism
☐ Before passage
▨ After passage
● AIMS[P]

$P = 0.0116$
$P = 0.0037$
$P = 0.0087$
$P = 1.0$

**d**

| tdTomato Venus | Indel (+) Indel (-) | Indel (-) Indel (+) | (n) |
|---|---|---|---|
| Tbx3-P2A₁-sgRNA1 | 46.6 ± 3.6 | 53.4 ± 3.6 (%) | 12 |
| Cdh1-P2A₁-sgRNA1 | 53.7 ± 3.2 | 46.3 ± 3.2 (%) | 17 |
| Cdh1-P2A₁-sgRNA2 | 53.0 ± 5.2 | 47.0 ± 5.2 (%) | 8 |
| Cdh1-P2A₁-sgRNA3 | 47.4 ± 5.4 | 52.6 ± 5.4 (%) | 9 |
| Cdh1-P2A₂-sgRNA4 | 45.2 ± 4.7 | 54.8 ± 4.7 (%) | 7 |
| Cdh1-P2A₂-sgRNA5 | 44.2 ± 5.0 | 55.8 ± 5.9 (%) | 9 |
| Cdh1-P2A₂-sgRNA6 | 44.6 ± 4.5 | 55.4 ± 4.5 (%) | 11 |
| Total (Fig. 2e) | 48.3 ± 1.6 | 51.7 ± 1.6 (%) | 73 |

**Extended Data Fig. 2 | See next page for caption.**

**Extended Data Fig. 2 | Minor effect of plasmid reduction on mono-allelic indel induction. a**, Indel patterns were analyzed using Cdh1-P2A$_2$-AIMS in mESCs (*n* = 3, independent experiments). The spacerless-PX459 plasmid (p:CP, 250 ng) was co-transfected with different amounts of the P2A$_2$-sgRNA4 expression plasmid (p:R). The total number of colonies analyzed is shown in each column. **b**, Construction of [0 C]–[30 C]sgRNA expressing all-in-one plasmids. Linkers were inserted into the *BpiI* site of the PX459 plasmid. Adenine (A, blue) was inserted at the third position from the 3'-end of the cytosine extension to create an overhang sequence for the insertion of spacer sequences with CCAC overhang. The [5 C]–[30 C]sgRNA-expressing all-in-one plasmids were produced by inserting a

standard 18–20-bp spacer linker between two *BpiI* sites. **c**, Relationship between Cas9 activity (AIMS[P]) and mosaic frequency before and after passage of puromycin-resistant primary colonies, assessed using different [C]sgRNAs in mESCs with Cdh1-P2A1-AIMS. Data are means ± SEMs for three independent experiments performed at different times. Statistical significance was assessed using two-way analysis of variance (ANOVA), followed by a post hoc Tukey–Kramer test. **d**, Table shows percentages of the two types of mono-allelic indel patterns in mESCs. Totals indicate the means of all data (*n* = 73), which are shown in Fig. 2e. Data are means ± SEMs.

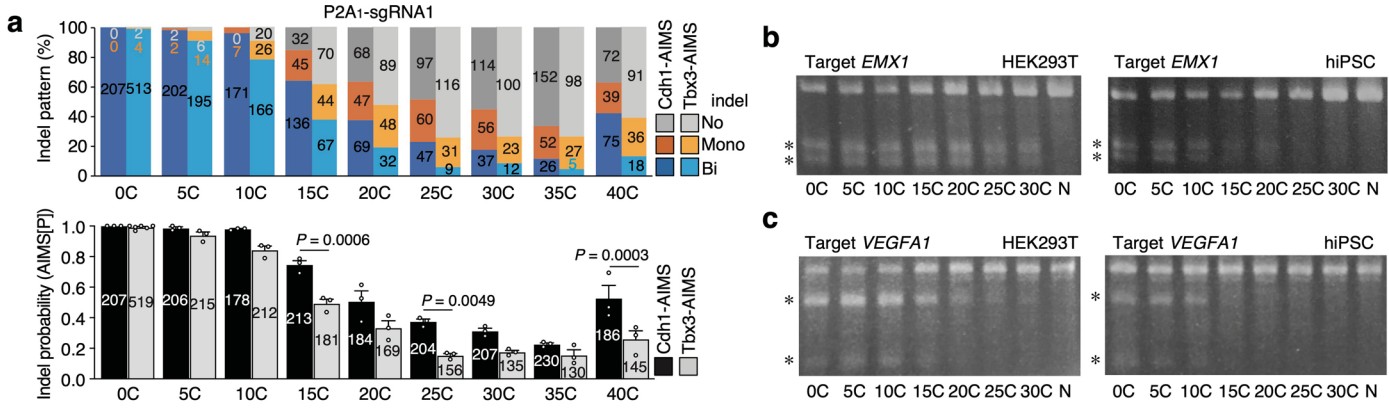

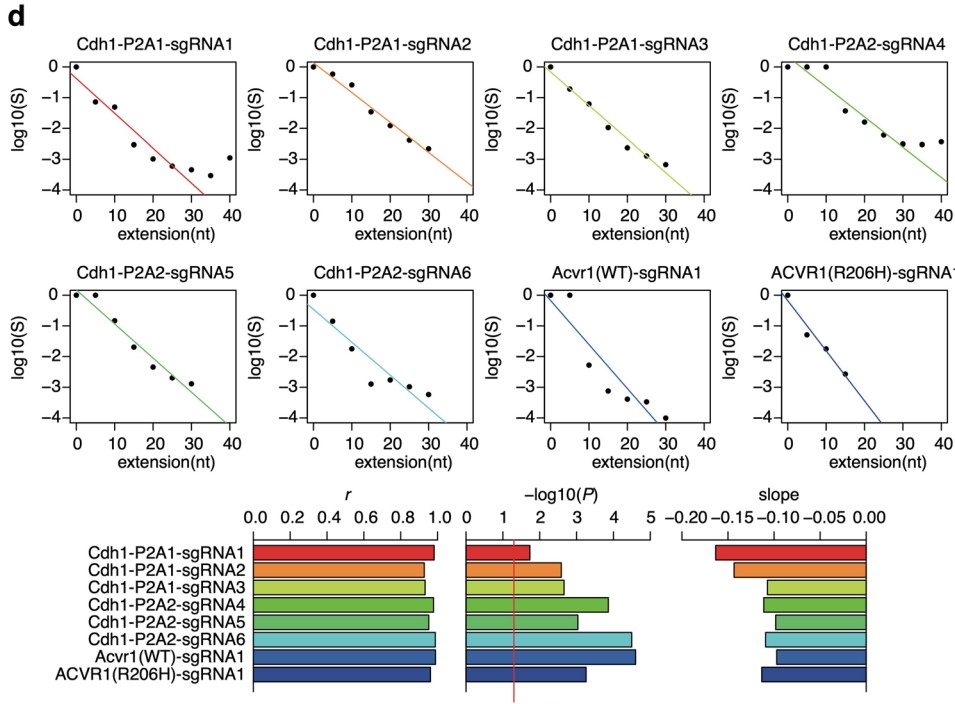

**Extended Data Fig. 3 | Quantitative assessment of suppressive effects of [C] extension for different sgRNAs. a**, Different indel frequencies at different chromosomal loci in mESCs. Indel patterns and probabilities (AIMS[P]) were compared between Cdh1-P2A$_1$-AIMS and Tbx3-P2A$_1$-AIMS in mESCs. Data regarding the indel pattern of Cdh1-AIMS (P2A$_1$-sgRNA1) from Fig. 2d are redisplayed for comparison. Data are means ± SEMs for three independent experiments (except for 0 C in Tbx3-AIMS, $n = 6$) performed at different times. Total colony numbers are shown in each column. Statistical significance was assessed using two-way ANOVA, followed by a post hoc Tukey–Kramer test. **b, c**, Comparison of indel probabilities between HEK293T cells and hiPSCs. *EMX1* (b)

or *VEGFA1* (c) was targeted. Note that editing efficiencies differ between *EMX1* and *VEGFA1* and between HEK293T cells and hiPSCs. Asterisks indicate PCR products digested in the T7E1 assay. N, PX459 plasmid without spacer. These images are also shown in Extended Data Fig. 8a, c. **d**, Relationships between [C] extension length and concentration of effective sgRNA-Cas9 complex (log10(S)) are shown for eight sgRNAs. The three lower panels show linear regression analysis results, including Pearson's correlation coefficients ($r$), $P$ values, and slopes. All eight sgRNAs had similar slope values, suggesting uniform effects of [C] extension.

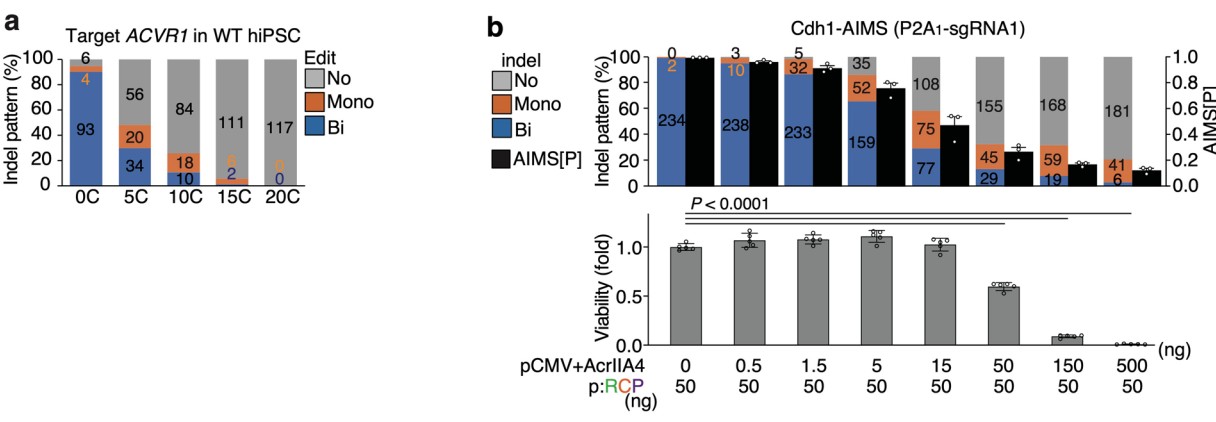

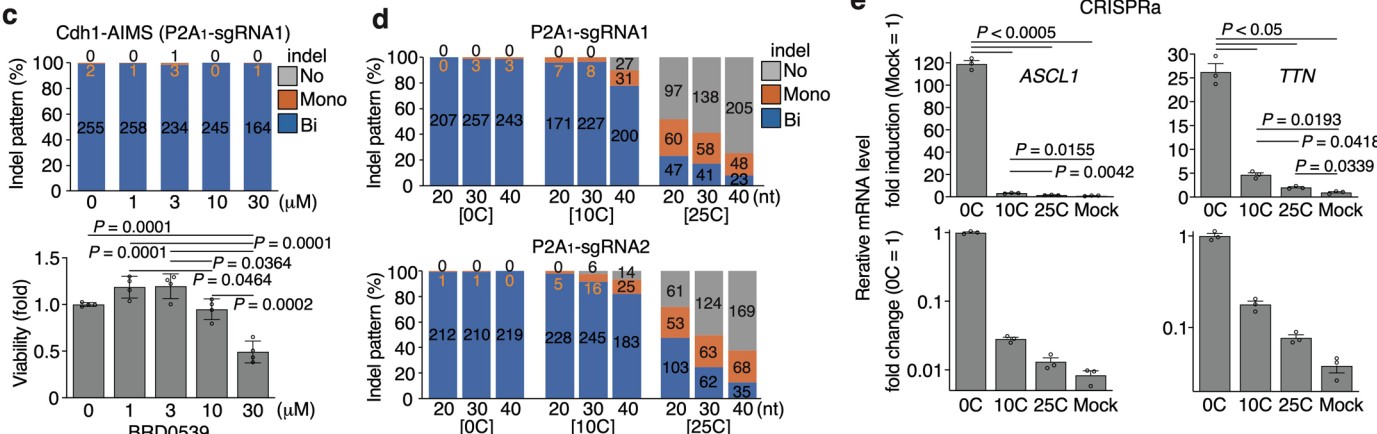

**Extended Data Fig. 4 | Additional verification of applicability of [C]sgRNA system. a**, Performance of [C]sgRNA system in ssODN-mediated R206H SNP knock-in for the *ACVR1* locus in WT hiPSCs (*n* = 4, independent experiments). Edit indicates both HDR and indels. The total number of clones is shown in each column (panels a-d). **b**, Effects of increasing amounts of AcrIIA4 plasmids on Cas9 activity (top, indel pattern analysis and AIMS[P], *n* = 3, independent experiments) and cell viability (bottom, *n* = 5, biological replicates) in mESC-AIMS (Cdh1-P2A₁-sgRNA1). **c**, Effects of Cas9 inhibitor BRD0539 on Cas9 activity (top, *n* = 3, independent experiments) and cell viability (bottom, *n* = 4, biological replicates)

in mESC-AIMS (Cdh1-P2A₁). **d**, Combitnatorial effects of long spacer-sgRNAs and [C] extension in mESC-AIMS (Cdh1-P2A₁) (*n* = 3, independent experiments, left, P2A₁-sgRNA1; right, P2A₁-sgRNA2). **e**, Effects of [C]sgRNAs on CRISPRa platform in HEK293T cells (*n* = 3, technical replicates). Mock indicates a spacer-less all-in-one CRISPRa plasmid and is used as a reference value to define fold induction (top panel). Statistical significance was assessed using one-way ANOVA and a post hoc Tukey–Kramer test (b, c) or Welch's ANOVA and a post hoc Games–Howell test t (e). Data are means ± SEMs (b, top) or means ± SDs (b, bottom, c, e).

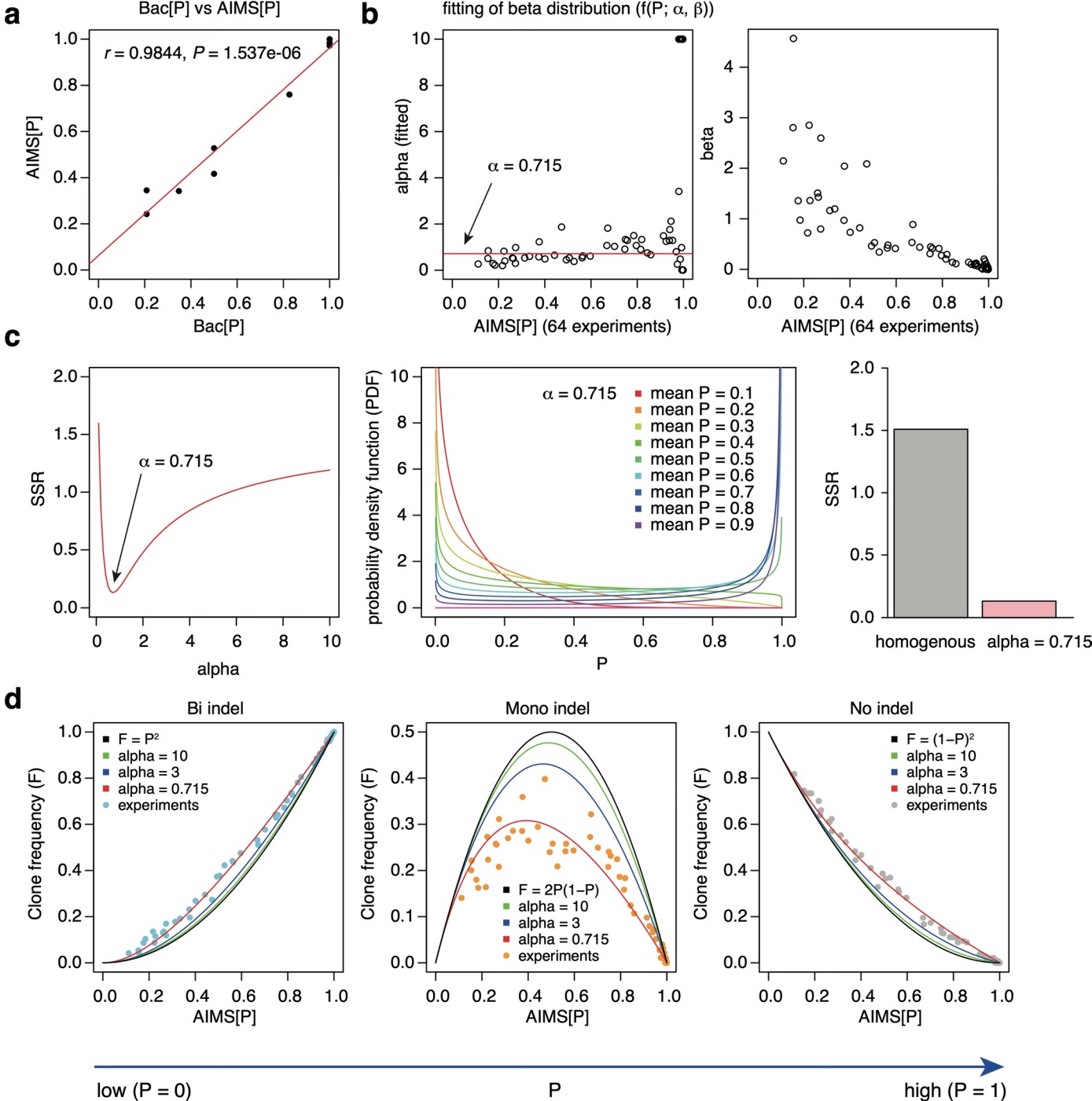

**Extended Data Fig. 5 | Computational modelling of genome editing frequency heterogeneity at the single-cell level using a beta distribution. a,** Correlation between Bac[P] and AIMS[P], which were presented in Fig. 5e. Linear regression curve, Pearson's correlation coefficient (*r*), and *P* value are shown. **b,** Beta distribution α and β values for each experiment that minimized the sum of squared residuals (SSR) between experimental and simulated F(Bi), F(Mono), and F(No). **c,** (Left panel) Identification of the fixed α value that minimized SSR

for all experiments. (Middle panel) Probability density functions for different mean P values are shown. (Right panel) Comparison of SSRs for homogenous or heterogeneous single-cell editing probability. **d,** Correlation between experimental data and predicted clone frequencies of bi-, mono-, or no-indel cases. Black line is for homogenous genome editing probability across the cell population. Additional beta distribution simulations are shown for α = 3 and α = 10. In c and d, P indicates indel probability.

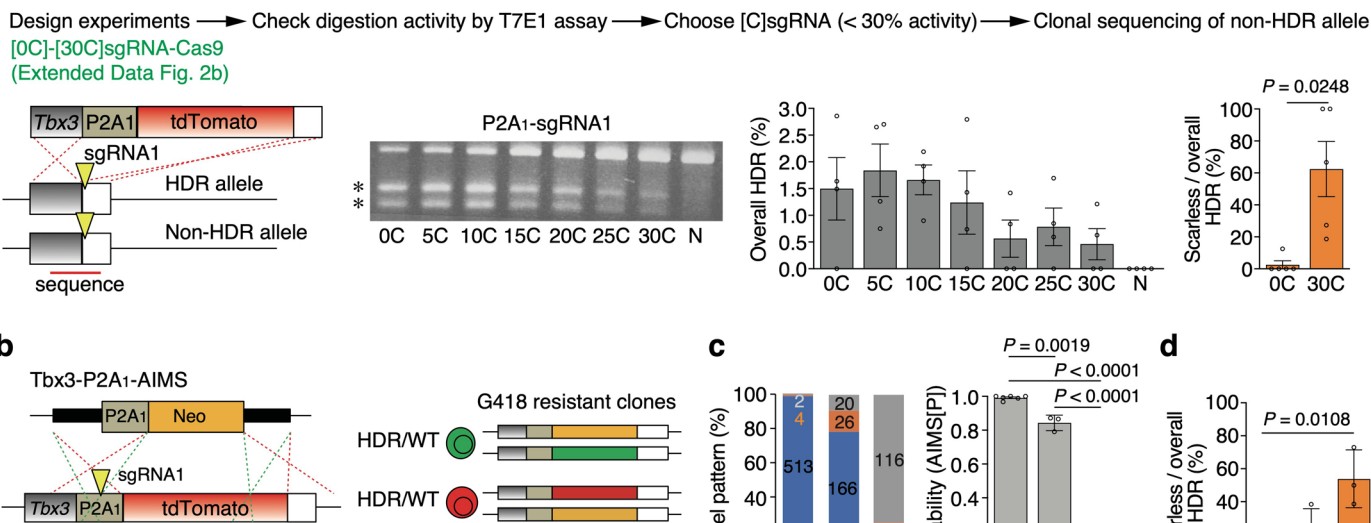

**a**

Design experiments → Check digestion activity by T7E1 assay → Choose [C]sgRNA (< 30% activity) → Clonal sequencing of non-HDR allele
[0C]-[30C]sgRNA-Cas9
(Extended Data Fig. 2b)

**Extended Data Fig. 6 | Induction of scarless mono-allelic gene cassette knock-in through reduction of Cas9 activity. a**, Mono-allelic HDR-based cassette knock-in without indel induction on a non-HDR allele in mESCs. Schematic of HDR (first panel), results of T7E1 assay (second pane), and frequencies of overall (third panel) and scarless (fourth panel) HDR are shown ($n = 4$ or 5, independent experiments, respectively). Pointers indicate DSB sites. Asterisks indicate PCR products digested by T7E1. N, PX459 plasmid without spacer. **b-d**, Scarless

mono-allelic HDR-based cassette replacement using AIMS in mESCs. Schematic of frequency measurement (b), indel pattern (c, left) and probability (c, right, AIMS[P] shown in Extended Data Fig. 3a), frequencies of scarless HDR (d) are shown ($n = 3$, independent experiments). The total colony number is shown in each column. Statistical significance assessed using two-tailed Welch's $t$-test (a) or one-way ANOVA and a post hoc Tukey–Kramer test (c, d). In panels a, c, and d, data are means ± SEMs. See also Extended Data Figs. 2 and 3.

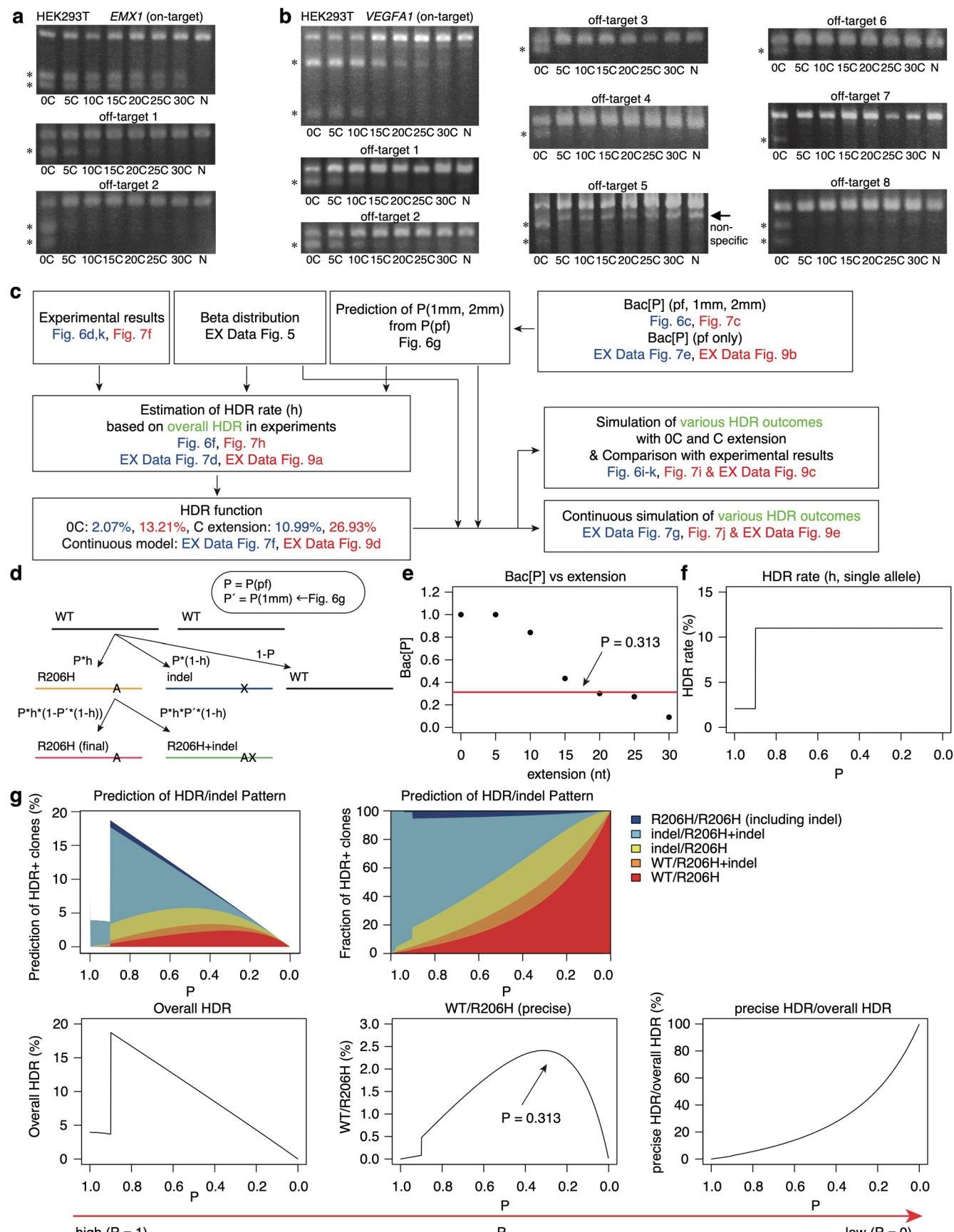

**Extended Data Fig. 7 | See next page for caption.**

**Extended Data Fig. 7 | Computational simulation of increased specificity coupled with off-target suppression and precise mono-allelic HDR. a, b,** Indel probabilities of other sgRNAs in HEK293T cells. A T7E1 assay was performed to investigate on-target and off-target indel probabilities for (a) *EMX1* and (b) *VEGFA1* targeting sgRNAs. These images are also shown in Extended Data Fig. 8e, g. **c,** Flowchart of the computational simulation of precise mono-allelic HDR to generate a fibrodysplasia ossificans progressiva (FOP) disease model in mESCs (Fig. 6) or to correct a single-nucleotide polymorphism (SNP) in FOP hiPSCs (Fig. 7 and Extended Data Fig. 9). **d,** Scheme of HDR-mediated generation of an FOP model (WT/R206H) from a wild-type (WT/WT) genotype. **e,** Relationship

between [C] extension length and Bac[P] shown in Fig. 6c. Red line indicates the value at which precise WT/R206H HDR reaches a maximum (P = 0.313). **f,** Hypothetical function of HDR rate (*h*) for the range of indel probability (P), based on data shown in Fig. 6f and described in detail in Methods. **g,** Simulated editing outcomes in the presence of HDR templates. Top panels show the relationships between indel probability (P) and the frequencies of various HDR clones (top, left) and their ratios (top, right). Bottom panels show the frequencies of overall HDR and precise WT/R206H editing and the ratio of WT/R206H clones to overall HDR. Arrow indicates the predicted maximum value (P = 0.313) at which precise WT/R206H HDR clones are generated. In c-g, P indicates indel probability.

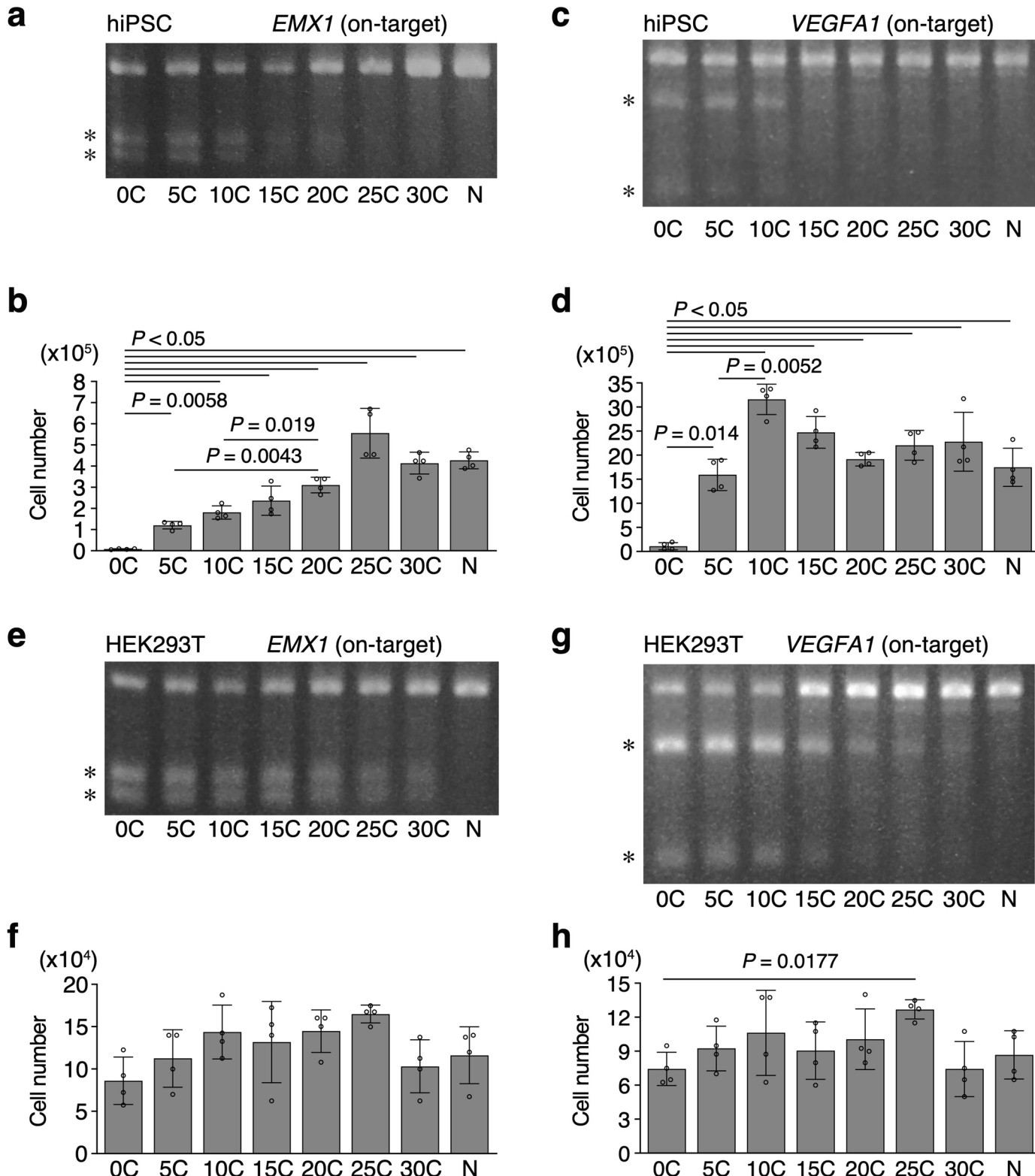

**Extended Data Fig. 8 | Suppression of cytotoxicity by [C] extension in hiPSCs.**
**a-d**, Indel probabilities (a and c) and cytotoxicities (b and d) of other sgRNAs
were investigated in hiPSCs. A T7E1 assay was performed to investigate on-target
indel probabilities for sgRNAs targeting (a) *EMX1* and (c) *VEGFA1*. These images
are also shown in Extended Data Fig. 3b and c. **e-h**, Indel probabilities (e and g)
and cytotoxicities (f and h) were investigated in HEK293T cells. A T7E1 assay was

performed to investigate on-target and off-target indel probabilities for sgRNAs
targeting (e) *EMX1* and (g) *VEGFA1*. These images are also shown in Extended
Data Fig. 7a and b. N, PX459 plasmid without spacer (a–h). Asterisks indicate PCR
products digested in the T7E1 assay (a, c, e and g). Statistical significance was
assessed using Welch's ANOVA and a post hoc Games–Howell test (b, d, f and h).
Data are means ± SDs.

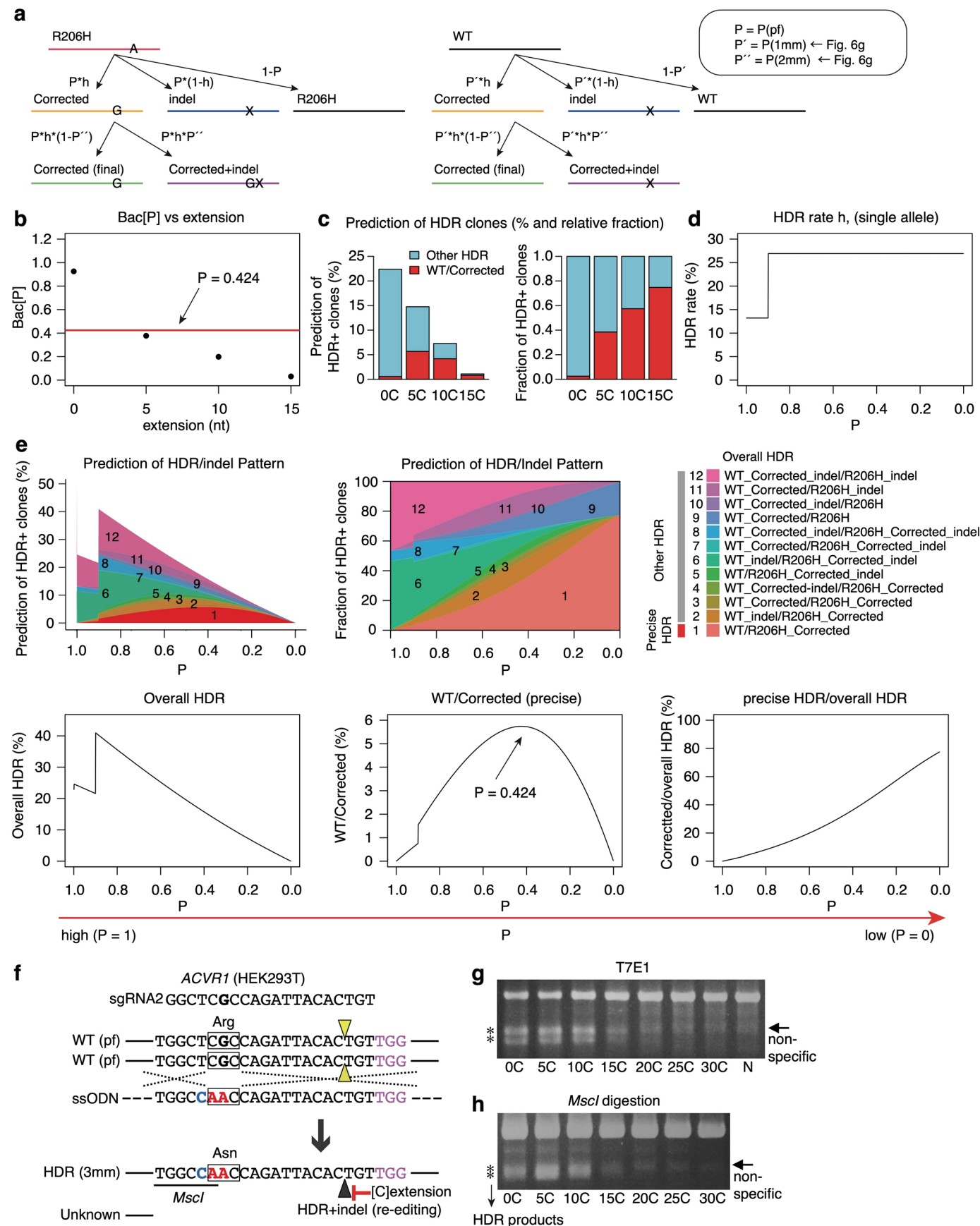

**Extended Data Fig. 9 | See next page for caption.**

**Extended Data Fig. 9 | Computational simulation of precise disease gene correction. a**, Scheme of HDR-mediated correction of an FOP model (WT/R206H) to a WT/Corrected genotype. **b**, Relationship between [C] extension length and Bac[P] for the R206H allele, as shown in Fig. 7c. Red line indicates the value at which precise WT/Corrected HDR reaches a maximum (P = 0.424). **c**, Prediction of precise WT/Corrected clones and (left) other HDR clones and (right) their ratios. **d**, Hypothetical function of HDR rate ($h$) for the range of indel probability (P); $h$ was determined based on data shown in Fig. 7h and described in detail in Methods. **e**, Simulated relationship between indel probability (P) and its ratio. The top left panel is also shown in Fig. 7j. Bottom panels show the frequencies of overall HDR and WT/Corrected HDR and the ratio of WT/Corrected clones to overall HDR. Arrow indicates the predicted maximum value at which precise WT/Corrected HDR clones are generated (P = 0.424). **f-h**, HDR frequency measurement in HEK293T cells. Schematic of HDR for 3-bp substitution in exon 5 of *ACVR1* (f). Silent mutation with cytosine (c, blue) and missense mutations with two adenines (AA, red) created a *MscI* restriction enzyme site, which allowed rapid quantification of HDR frequency. Squares indicate codons. pf, perfect match; 3 mm, 3-bp mismatches. Pointers indicate DSB sites. Asterisks indicate PCR products digested by (g) T7E1 or (h) *MscI* restriction enzyme. N, PX459 plasmid without spacer. In a, b, d and e, P indicates indel probability.

# nature research

# Reporting Summary

Nature Research wishes to improve the reproducibility of the work that we publish. This form provides structure for consistency and transparency in reporting. For further information on Nature Research policies, see our Editorial Policies and the Editorial Policy Checklist.

## Statistics

For all statistical analyses, confirm that the following items are present in the figure legend, table legend, main text, or Methods section.

| n/a | Confirmed | |
|---|---|---|
| ☐ | ☒ | The exact sample size (*n*) for each experimental group/condition, given as a discrete number and unit of measurement |
| ☐ | ☒ | A statement on whether measurements were taken from distinct samples or whether the same sample was measured repeatedly |
| ☐ | ☒ | The statistical test(s) used AND whether they are one- or two-sided<br>*Only common tests should be described solely by name; describe more complex techniques in the Methods section.* |
| ☐ | ☒ | A description of all covariates tested |
| ☐ | ☒ | A description of any assumptions or corrections, such as tests of normality and adjustment for multiple comparisons |
| ☐ | ☒ | A full description of the statistical parameters including central tendency (e.g. means) or other basic estimates (e.g. regression coefficient) AND variation (e.g. standard deviation) or associated estimates of uncertainty (e.g. confidence intervals) |
| ☐ | ☒ | For null hypothesis testing, the test statistic (e.g. *F*, *t*, *r*) with confidence intervals, effect sizes, degrees of freedom and *P* value noted<br>*Give P values as exact values whenever suitable.* |
| ☒ | ☐ | For Bayesian analysis, information on the choice of priors and Markov chain Monte Carlo settings |
| ☒ | ☐ | For hierarchical and complex designs, identification of the appropriate level for tests and full reporting of outcomes |
| ☐ | ☒ | Estimates of effect sizes (e.g. Cohen's *d*, Pearson's *r*), indicating how they were calculated |

*Our web collection on statistics for biologists contains articles on many of the points above.*

## Software and code

Policy information about availability of computer code

| Data collection | Fluorescent microscopes (BZ-X800, Keyence and IX73, Olympus); Luminescent image analyzer (LAS-3000, FUJIFILM); Fluorescence stereo microscope (M165FC, Leica); Real-Time System (CFX Connect, BIO-RAD). |
|---|---|
| Data analysis | R version 3.2.1, JMP 14.2.0., FIJI, Graphpad Prism 8.4.3, Microsoft Excel v16. Measurement application software (Keyence, BZ-H4M). |

For manuscripts utilizing custom algorithms or software that are central to the research but not yet described in published literature, software must be made available to editors and reviewers. We strongly encourage code deposition in a community repository (e.g. GitHub). See the Nature Research guidelines for submitting code & software for further information.

## Data

Policy information about availability of data

All manuscripts must include a data availability statement. This statement should provide the following information, where applicable:
- Accession codes, unique identifiers, or web links for publicly available datasets
- A list of figures that have associated raw data
- A description of any restrictions on data availability

The main data supporting the results in this study are available within the paper and its Supplementary Information. Source data for the figures are provided with this paper. All data generated in this study, are available from the authors on reasonable request.

April 2020

# Field-specific reporting

Please select the one below that is the best fit for your research. If you are not sure, read the appropriate sections before making your selection.

☒ Life sciences ☐ Behavioural & social sciences ☐ Ecological, evolutionary & environmental sciences

For a reference copy of the document with all sections, see nature.com/documents/nr-reporting-summary-flat.pdf

# Life sciences study design

All studies must disclose on these points even when the disclosure is negative.

| | |
|---|---|
| Sample size | We did not use a statistical method to determine sample sizes. We chose sample size (at least n = 3) on the basis of prior experience. |
| Data exclusions | No data were excluded. |
| Replication | All attempts to reproduce the results were successful. |
| Randomization | Randomization was not performed. Sample randomization is not applicable for cell-culture experiments where batches of homogeneous cultures can be tested in parallel. |
| Blinding | The investigators were not blinded to sample identity. We did not perform any analysis entailing subjective group allocation. |

# Reporting for specific materials, systems and methods

We require information from authors about some types of materials, experimental systems and methods used in many studies. Here, indicate whether each material, system or method listed is relevant to your study. If you are not sure if a list item applies to your research, read the appropriate section before selecting a response.

## Materials & experimental systems

| n/a | Involved in the study |
|---|---|
| ☐ | ☒ Antibodies |
| ☐ | ☒ Eukaryotic cell lines |
| ☒ | ☐ Palaeontology and archaeology |
| ☐ | ☒ Animals and other organisms |
| ☒ | ☐ Human research participants |
| ☒ | ☐ Clinical data |
| ☒ | ☐ Dual use research of concern |

## Methods

| n/a | Involved in the study |
|---|---|
| ☒ | ☐ ChIP-seq |
| ☒ | ☐ Flow cytometry |
| ☒ | ☐ MRI-based neuroimaging |

## Antibodies

| | |
|---|---|
| Antibodies used | Rabbit polyclonal p53 antibody (Santa Cruz, Catalog number SC-6243, Clone name FL-393, Lot number I0705); Rabbit monoclonal pSmad1/5/8 antibody (Cell Signaling Technology, Catalog number 13820, Clone name D5B10, Lot number 1); Alexa Fluor 488 donkey anti-rabbit secondary antibody IgG (H+L) (Thermo Scientific, Catalog number A21206, Lot number 1796375). |
| Validation | The validation of the p53 and pSmad1/5/8 antibodies in human cells is stated in the manufacture's website and they have been repeatedly used in previous reports. |

## Eukaryotic cell lines

Policy information about cell lines

| | |
|---|---|
| Cell line source(s) | Mouse ESC lines (B6-5-2, B6-D2-4, and R26R YFP/+ mESC) were established for this study. Human iPS cell lines (409B2, HPS0076, and FOP, HPS0376) were provided by the RIKEN BRC. HEK293T cells were kindly provided by Dr. Miyoshi (Keio University). Human adipose-derived stem cells (hADSCs) were purchased (Thermo Fisher Scientific). |
| Authentication | Cell-line authentication was performed based on cell morphology, growth condition, and specific properties. |
| Mycoplasma contamination | All cell lines tested were negative for mycoplasma contamination. |
| Commonly misidentified lines (See ICLAC register) | No commonly misidentified cell lines were used. |

## Animals and other organisms

Policy information about studies involving animals; ARRIVE guidelines recommended for reporting animal research

| | |
|---|---|
| Laboratory animals | C57BL/6 mice, male and female (Clea Japan, Tokyo, Japan), ICR mice, male and female (Clea Japan, Tokyo, Japan),R26R YFP/YFP mice, male (a gift from Frank Costantini, Columbia University, New York, NY) and Cdh1-AIMS mice (generated in this work) were used. |
| Wild animals | The study did not involve wild animals. |
| Field-collected samples | The study did not involve samples collected from the field. |
| Ethics oversight | The study were approved by the Kyushu University Animal Experiment Committee, and the care and use of the animals were performed in accordance with institutional guidelines. |

Note that full information on the approval of the study protocol must also be provided in the manuscript.

