## [Peer Review File · Nature Biomedical Engineering]

Optimization of Cas9 activity through the addition of cytosine extensions to single-guide RNAs

Corresponding author: Masaki Kawamata

Editorial note

This document includes relevant written communications between the manuscript's corresponding author and the editor and reviewers of the manuscript during peer review. It includes decision letters relaying any editorial points and peer-review reports, and the authors' replies to these (under 'Rebuttal' headings). The editorial decisions are signed by the manuscript's handling editor, yet the editorial team and ultimately the journal's Chief Editor share responsibility for all decisions.

Any relevant documents attached to the decision letters are referred to as **Appendix #**, and can be found appended to this document. Any information deemed confidential has been redacted or removed. Earlier versions of the manuscript are not published, yet the originally submitted version may be available as a preprint. Because of editorial edits and changes during peer review, the published title of the paper and the title mentioned in below correspondence may differ.

Correspondence

Tue 08 Feb 2022

Decision on Article nBME-21-2632

Dear Dr Kawamata,

Thank you again for submitting to *Nature Biomedical Engineering* your manuscript, "Rational optimization of versatile genome editing applicability by wide-range programmable Cas9 inhibition with safeguard sgRNAs", and for your patience in waiting for the reviewer reports and our feedback. As noted in previous e-mail correspondence, the manuscript has been seen by three experts, whose reports you will find at the end of this message.

You will see that the reviewers appreciate aspects of the work. However, they articulate concerns about the degree of support for some of the claims and about the advance that the work represents over relevant published studies, and provide useful suggestions for improvement. We hope that with significant further effort you can address the criticisms, increase the level of significance of the study, and convince the reviewers of its merits. In particular, we would expect that a revised version of the manuscript provides:

- * Head-to-head comparisons of the performance of cytosine extension at the 5' end with alternative strategies, as per the comments of Reviewers #3 and #1.
- * Evidence of advantageous biomedically relevant applicability of the approach, beyond the alleviation of p53 activation and cytotoxicity in induced pluripotent stem cells.
- * Extended evidence of the efficiency of monoallelic and biallelic editing in additional cell lines, as per the questions and suggestions of Reviewer #2.

When you are ready to resubmit your manuscript, please upload the revised files, a point-by-point rebuttal to the comments from all reviewers, the reporting summary, and a cover letter that explains the main improvements included in the revision and responds to any points highlighted in this decision.Please follow the following recommendations:

- * Clearly highlight any amendments to the text and figures to help the reviewers and editors find and understand the changes (yet keep in mind that excessive marking can hinder readability).
- * If you and your co-authors disagree with a criticism, provide the arguments to the reviewer (optionally, indicate the relevant points in the cover letter).
- * If a criticism or suggestion is not addressed, please indicate so in the rebuttal to the reviewer comments and explain the reason(s).
- * Consider including responses to any criticisms raised by more than one reviewer at the beginning of the rebuttal, in a section addressed to all reviewers.
- * The rebuttal should include the reviewer comments in point-by-point format (please note that we provide all reviewers will the reports as they appear at the end of this message).
- * Provide the rebuttal to the reviewer comments and the cover letter as separate files.

We hope that you will be able to resubmit the manuscript within 20 weeks from the receipt of this message. If this is the case, you will be protected against potential scooping. Otherwise, we will be happy to consider a revised manuscript as long as the significance of the work is not compromised by work published elsewhere or accepted for publication at *Nature Biomedical Engineering*.

We hope that you will find the referee reports helpful when revising the work. Please do not hesitate to contact me should you have any questions.

Best wishes,

Pep

Pep Pàmies
Chief Editor, Nature Biomedical Engineering

Reviewer #1 (Report for the authors (Required)):

Kawamata et al. developed methods to achieve intended editing using Cas9 nuclease. First, the authors generated an allele-specific indel monitoring system (AIMS) by adding P2A-tdTomato and P2A-Venus sequences to genes encoding a membrane protein (Chd1) and a nuclear protein (Tbx3). Using these fluorescent reporters, the authors visualized allele-specific genome editing events and found that bi-allelic DNA cleavage was dominant when appropriate sgRNAs were designed. In addition, the authors observed that AIMS was more accurate than the T7E1 assay. Then, the authors attenuated Cas9 activity by adding [C] extensions at the 5'-ends of sgRNA spacer sequences. Next, the authors developed a computational model to maximize mono-allelic editing. Based on these results, the authors generated a heterozygous single-nucleotide polymorphism disease model through optimized mono-allelic editing. The authors also performed computational modeling to maximize the frequency of intended HDR-mediated editing and optimized the method for inducing precise gene corrections in human iPSCs.

This study is well-design and carefully conducted. However, the impact and novelty of the current study are limited, especially given that prime editing technology is available to induce precise gene corrections. Thus, the current study based on Cas9 nuclease would be suitable for a more specialized journal.

Major comments

1. As stated above, the precise genome editing that the authors intended can be induced using prime

editing, probably in a more efficient and specific manner. Unless the authors show evidence that their approach is superior to the approach based on prime editors, the impact and implications of the current study, especially with respect to biomedical applications of genome editing, do not reach the level of Nature Biomedical Engineering.

2. The authors attenuated sgRNA efficiencies by adding [C] extensions to the 5' ends of sgRNAs. Attenuation of sgRNA efficiencies by adding an extension to the 5' end has been published (Perli et al. Science 2016, DOI: 10.1126/science.aag0511), a reference that the authors do not cite. In addition, a method for guide RNA activity attenuation was also previously published (Jost et al. Nat. Biotechnol. 2020, <https://doi.org/10.1038/s41587-019-0387-5>), which is not cited in the current manuscript, either.

3. As conducted in Jost et al. Nat. Biotechnol. 2020, the authors could develop a machine learning-based model that predicts sgRNA activity, which would strengthen the manuscript.

Minor comment

1. The authors stated "editing outcomes at each allele are stochastic" based on their results on a handful of sgRNAs. However, testing at thousands of target sequences has shown that editing outcomes are not stochastic (Shen et al. Nature 2018, doi: 10.1038/s41586-018-0686-x; Allen et al., Nat. Biotechnol. 2018, <https://doi.org/10.1038/nbt.4317>). The authors may want to tone down their statement considering these previous studies, perhaps saying "partially stochastic".

Reviewer #2 (Report for the authors (Required)):

Gene editing with CRISPR-Cas9 is widely used to alter genome sequence for gene function study and clinical application. When expressed in high levels, mainly through plasmid-based expression systems, sgRNA-Cas9 frequently induces biallelic editing. To induce monoallelic editing, the level of sgRNA-Cas9 will need to be carefully modulated with different approaches. However, these approaches require optimization to balance between mono vs. biallelic editing. How to modulate the intracellular level of sgRNA-Cas9 is therefore important to determine whether a desirable outcome of gene editing experiments can be achieved. The authors in this manuscript use an allele-specific indel monitor system (AIMS) to analyze editing patterns of a pair of alleles in a large number of clones without sequencing analysis. Using AIMS, they discover that adding cytosine stretches to the 5'-end of a sgRNA reduces the formation of the effective sgRNA-Cas9 complex, which affects the frequency of monoallelic and biallelic editing events. By controlling the level of effective sgRNA-Cas9 complexes via the C stretch, the authors are able to demonstrate improved efficiency of getting clones with gene knockin/wt and cassette replacement/wt. They also generate a SNP/wt clone that recapitulates a disease model in mice. Finally, the authors display the advantage of using the C stretch in a gene correction study with human patient-derived iPSCs. They also show that p53 activation by DNA damage responses is prevented with the C stretch, reducing cell toxicities caused by excessive Cas9-sgRNA-induced DNA double-strand breaks. Although the length of the cytosine stretch needs to be optimized with different sgRNAs, the authors argue that their approach likely represents an easier and more efficient strategy to achieve the desirable outcome of gene editing relative to other approaches trying to modulate the intracellular level of sgRNA-Cas9.

The authors have designed and performed extensive experiments to support their hypothesis and most of the conclusions can be supported by the data. The finding that extended C residues in sgRNA can modulate the sgRNA-Cas9 activity represents a novel and important advancement in the field of gene editing. If proven to be generally applicable in more than just mESCs, this finding will have an impact on the current state of the art in gene editing.

Issues need to be addressed to clarify some of their claims.

1. Most of the experiment were carried out in mESCs. The frequency of biallelic editing is surprisingly high as shown in Fig. 1e and 1f. Even with 100 fold less DNA at 2.5 ng, the efficiency of biallelic editing remains unchanged (Fig. 2a). This is somewhat surprising as such a drastic reduction in the amount of the transfected DNA is expected to have an impact on the frequency of the monoallelic and biallelic gene editing event. Can the authors explain this? The question is whether this outcome depends on the cell line used. Will cell lines with a reduced transfection efficiency generate lower levels of the effective sgRNA-Cas9 and

thus altering the frequency of monoallelic and biallelic editing? Performing a similar study in human iPSCs or ESCs known to have low gene editing efficiency and slower proliferation rate would address whether the ratio between monoallelic and biallelic gene editing varies, depending on different cell lines. Similarly, using the preformed

sgRNA-Cas9 RNP complex instead of the expression plasmid should significantly reduce the gene editing efficiency and alter the ratio between monoallelic and biallelic editing. The authors should discuss the advantage of the current finding over the RNP strategy for sgRNA-Cas9 modulation.

2. The way of deriving AIMS(P) and adjusted AIMS(P) in Fig. 1h and 1j needs to be described in more details as it is not obvious from the Methods section how the authors derive the frequency. It is not clear whether these so-called rare tdTomato+/Venusindel and tdTomatoindel/Venus+ heterozygous clones are grouped into either monoallelic or biallelic clones shown in Fig. 1e.

3. The point of extended Fig. 3b and 3c (line 164) is not clear. The effect of extended Cs on gene editing is clear in both HEK293T and hiPSC. However, as the two cell lines have very different transfection efficiencies, comparing indel probability between them as the figure title suggests is not appropriate.

4. One issue with the experiments in the manuscript is the frequent missing of the cell line the authors use. For example, the cell lines used in experiments Fig. 4g and Fig. 5 are not specified. The authors need to provide this information clearly either in the text or in the figure legends.

5. The claim (line 348) that “[C] extension generally recovered the HDR rate which had presumably been suppressed by the conventional CRISPR-Cas9 system” needs further clarification. Will the authors provide references or data to support such a claim?

6. Both Fig. 6 and Fig. 7 show that the Cas9 activity would make single nucleotide mismatch less sensitive to re-cleavage. However, it is unclear whether this is dependent on the position of the mismatch nucleotide (1mm) in the sgRNA. If the mismatch occurs at a different position, will the difference between pf and 1mm disappear? Can the authors comment on this? It is also useful if the authors can comment on the effect of incorporating PAM mutations as described in reference 22 into their donor template on re-cleavage of the HDR allele in their system.

7. In line 384, the authors explain “We observed a sharp decline in the editing efficiency of hiPSCs, compared with mESCs (Fig. 7c), which may be partly explained by our selection of hiPSCs with weak p53 activation.” Why does the use of hiPSCs with weak p53 activation reduce gene editing efficiency in hiPSCs? Activation of p53 counteracts gene editing. Weak p53 activation should therefore improve gene editing instead of reducing it.

8. A couple of mistakes need to be corrected;
line 448, Fig. 3h. There is no Fig. 3h;
line 462, Fig. 5g, h. There are no Fig. 5g and h.

Reviewer #3 (Report for the authors (Required)):

The manuscript reports a method to reduce the editing efficiency and increase monoallelic editing using a modified Cas9 guide RNA design. The authors first developed an allele-specific indel monitor system (AIMS) to visualize on the single cell level the outcomes of editing: no indel, in frame indel, frameshift indels/large deletions. They used the system to show appending C's to the 5' end of sgRNA can reduce the biallelic editing and increase monoallelic editing. They applied the method to generate monoallelic single-nucleotide polymorphism (SNP) gene editing in mESC and hiPSC to create a fibrodysplasia ossificans progressiva (FOP) mutation.

Modifying the 5' end of sgRNA to reduce its editing efficiency is not totally surprising, as previous studies have observed similar effects (PMID: 32496535, 28985763, 30988504). The nice part of the work is its detailed characterization for monoallelic editing using the modified sgRNA, which is a methodological advance. The finding may provide a simple and useful strategy for the gene editing and gene therapy field. But it currently misses a key comparison with alternative approaches. The manuscript has several interesting components, but the overall organization of the story should be improved to increase its impact in the field.

Major comments:

1. Comparison with other technologies. As a technology, it is important to know how this method compares to alternative methods and what the advantages are. Is this C-extension method superior compared to previous methods including anti-Cas9 protein and small molecule inhibitors? Have the authors performed comparative studies to prove the superiority of the method given anti-Cas9 (PMID: 30643127) or small molecules (31051099) are reported? This is important as this information will provide key data for other researchers to decide which approach to use.

2. Rationale for the technology. The authors need to provide better rationale at the beginning of the story for why they need to limit editing efficiency in the abstract and the introduction. The authors wrote in the abstract, "Controlled inhibition of CRISPR-Cas9 is important for precise and safe genome editing". I am not sure if this statement is totally accurate. The gene editing field desires technologies with high editing efficiency and less toxicity and less off-targets. Methods that can achieve high precision and safety while sacrificing efficiency may or may not be chosen. The question is whether their safeguard sgRNA method is better in terms of this tradeoff compared to alternative methods.

3. Application. The authors demonstrated using their method to generate fibrodysplasia ossificans progressiva (FOP) monoallelic mutation in stem cells. Is there an application that can be uniquely addressed by their approach? For example, can the use of the safeguard sgRNA increase gene editing specificity? In Kocak Nat Biotech 2019 work where they introduced a hairpin structure to the 5' end of sgRNA, they observed greatly increased editing specificity. Can the authors demonstrate such a nice feature for gene editing that is important to the applications?

4. The AIMS system is interesting as it provides single-cell readout of various editing outcomes. However, it also requires precisely inserting the dual-color reporters into cells and selecting single clones for testing sgRNAs. Can the authors comment on the broader utility of the AIMS system to study gene editing? Can the authors apply the system to anti-CRISPR and small molecule inhibitors for a fair comparison with their safeguard sgRNA approach? Can the authors apply AIMS to other Cas molecules such as Cas12a? It has been reported that Cas12a may have different monoallelic vs. biallelic editing activities from Cas9. Performing such experiments can enhance the significance of the story.

5. How are off-targets in Fig. 6 measured? Have the authors performed standard assays such as GUIDE-seq to quantify the off-target effects?

6. Also in Fig. 6, it is confusing to see that [0C]sgRNA induced HDR in only 4.1% of clones, much less than [5C]sgRNA which is 20.5%. This is inconsistent with their conclusions in Fig. 2g. How do authors explain this observation?

Minor comments:

7. Some figures including Fig. 3 and some computational modeling figures can be moved to Extended Data figures. The text describing these biochemical assays and detailed computation modeling can be moved to supplemental text. This can increase overall readability of the story, helping the readers to focus on the main findings.

Thu 25 Aug 2022

Decision on Article nBME-21-2632A

Dear Dr Kawamata,

Thank you for your revised manuscript, "Rational optimization of versatile genome editing applicability by wide-range programmable Cas9 inhibition with safeguard sgRNAs", which has been seen by the original reviewers. In their reports, which you will find at the end of this message, you will see that the reviewers acknowledge the improvements to the work and that Reviewer #3 raises a few additional technical and stylistic points that I am hoping you will be able to address. In particular, please do make clearer the advantages of your approach to reducing sgRNA activity.

As before, when you are ready to resubmit your manuscript, please upload the revised files, a point-by-point rebuttal to the comments from all reviewers, the reporting summary, and a cover letter that explains the main improvements included in the revision and responds to any points highlighted in this decision.

We hope that you will be able to resubmit the manuscript within 6 weeks from the receipt of this message. If this is the case, you will be protected against potential scooping. Otherwise, we will be happy to consider a revised manuscript as long as the significance of the work is not compromised by work published elsewhere or accepted for publication at *Nature Biomedical Engineering*.

We look forward to receive a further revised version of the work. Please do not hesitate to contact me should you have any questions.

Best wishes,

Pep

Pep Pàmies
Chief Editor, Nature Biomedical Engineering

Reviewer #1 (Report for the authors (Required)):

I think that the revised manuscript is now suitable for publication.

Reviewer #2 (Report for the authors (Required)):

After reviewing the response from the authors, the previous comments for the manuscript have been addressed satisfactorily. No additional issues from this reviewer.

Reviewer #3 (Report for the authors (Required)):

The authors have addressed some concerns. Thanks to their efforts. While new experiments and further computational analysis answered or clarified some questions, there are major remaining issues that need to be addressed.

•A general comment: it is much easier to reduce the activity of gene editing than to improve its activity. It is easy to conceive that there are many possible ways to reduce the activity of a sgRNA, such as by removing a few nucleotides, mutating a few nucleotides in the scaffold, or appending other sequences that can disrupt its structure. The strategy presented here is just one of numerous options to reduce the sgRNA activity.

•Regarding their response to #1, comparison with C-sgRNA with Acr and small molecules, two things to note here.

1) Comparing the viability of having more C's on the sgRNA to the viability of transfecting more Acr plasmid is not a fair comparison. It is unfair to claim that Acr is more toxic, as it could be that there is naked DNA toxicity from increasing Acr plasmid while the C-extension guide plasmid mass remains the same across all experiments. The toxicity is likely due to naked DNA not the Acr. A fairer comparison would be to create an analogous plasmid to the extended sgRNA using a suite of promoters (e.g. similar to Figure 1d, [promoter] – Acr – BFP – CBh – SpCas9 – T2A – puro), transfecting in the DNA, and seeing how Acr expression (as a function of BFP) correlates to viability and mono-allelic efficacy.

2) Acr seems to be just as good if the goal is to get mono-allelic (see the 25ng and 75ng Acr case compared to 20, 30, and 35 extension). The manuscript is heavy on 'tunability', so it is still unclear why C-extension guide strategy is better than Acr. It is likely that C extension is easier to implement, but Acr gives more synthetic biology options.

•Regarding their response to #2: the authors changed the abstract to "precise regulation" instead of "controlled inhibition". They should explain better what 'precise regulation' means in the Discussion. In general, the paper is very dense on method details and stats but does not give enough context on why it is important.

•Regarding their response to #6: I don't understand they answered the question. It doesn't seem to explain why [0C] sgRNA has a low efficiency.

•For their off-target detection approach (comment #3, #5): T7E1-based method is less comprehensive as compared to GUIDE-seq for revealing a full profile of off-targets.

•The paper is dense and very hard to read. Much of it includes models that don't help genome engineers to apply what they are learning. More empirical and applications-based evidence would have been better than dense probability modeling.

Tue 04 Oct 2022

Decision on Article nBME-21-2632B

Dear Dr Kawamata,

Thank you for your revised manuscript, "Rational optimization of versatile genome editing applicability by wide-range programmable Cas9 inhibition with safeguard sgRNAs". Having consulted with Reviewer #3 (whose brief comments you will find at the end of this message), I am pleased to write that we shall be happy to publish the manuscript in *Nature Biomedical Engineering*.

We will be performing detailed checks on your manuscript, and in due course will send you a checklist detailing our editorial and formatting requirements. You will need to follow these instructions before you upload the final manuscript files.

Best wishes,

Pep

Pep Pàmies
Chief Editor, Nature Biomedical Engineering

Reviewer #3 (Report for the authors (Required)):

The authors provided more data and detailed clarification to address my concerns. Although not all experiments (such as off-targets) were performed using most comprehensive approaches, the reviewer understands such assays can be tedious to perform at the stage. The available data and explanation provided evidence to support their claims. Therefore this reviewer agrees that this version of manuscript is suitable for consideration for publication.

Rebuttal 1

Summary of the revision (nBME-21-2632):

“Rational optimization of versatile genome editing applicability by wide-range programmable Cas9 inhibition with safeguard sgRNAs”

We thank reviewers for evaluating our manuscript. Please find our summary of the revised manuscript and point-by-point response to reviewers’ criticisms below.

All changes are highlighted in red in the revised manuscript.

<< Overall advances in the revised manuscript >>

Summary

According to the reviewers’ suggestions, we performed (1) comparison of cytosine extension vs. other Cas9 modulation approaches, (2) analyses of extended applicability of cytosine extension methods in CRISPRa/i and Cas12a platforms, and (3) evaluation of cytosine extension methods in other cell lines besides mESCs.

First, we compared the cytosine extension method, anti-Cas9 small molecule inhibitor, and anti-Cas9 proteins. Among them, the cytosine extension method enables both wide-range Cas9 inhibition and reduced toxicity, while anti-Cas9 proteins can suppress Cas9 activities but also yield considerable toxicity along with increasing dose. In our assay, anti-Cas9 small molecule inhibitors did not work well to suppress Cas9 activities. These results highlight the significance of our approach without the use of other molecules with unknown adverse effects.

Second, we demonstrated that cytosine extension methods are compatible with CRISPRa/i and Cas12a platforms to finely tune their activities, suggesting the wide-range applicability in CRISPR-related technologies.

Third, we confirmed that cytosine extension methods work well in other cell lines including human iPSCs, adipose-derived stem cells (ADSCs), and mouse hepatoblasts derived from AIMS mice, which we have newly generated.

These results collectively reinforce versatile applicability of wide-range programmable Cas9 inhibition with safeguard sgRNAs.

Our response to reviewers is subdivided into two sections:

Section 1: Summary of new experiments and analyses (Responses to three major comments from the editor)

Section 2: Point by point responses to reviewers’ comments.

*** Section 1 covers the comments from the editor and raised by more than one reviewer.**

Section 1: Summary of new experiments and analyses:

(1) Comparison of [C]sgRNA and other Cas9 modulation platforms

(2) Expansion of biomedical applicability of the [C]sgRNA platform

(3) Performance of [C]sgRNA in other cell lines

(1) Comparison of [C]sgRNA and other Cas9 modulation platforms

(Editor comment No. 1)

** Head-to-head comparisons of the performance of cytosine extension at the 5’ end with alternative strategies, as per the comments of Reviewers #3 and #1.*

(Reviewer #3, major comment No. 1)

1. Comparison with other technologies. As a technology, it is important to know how this method compares to alternative methods and what the advantages are. Is this C-extension method superior compared to previous methods including anti-Cas9 protein and small molecule inhibitors? Have the authors performed comparative studies to prove the superiority of the method given anti-Cas9 (PMID: 30643127) or small molecules (31051099)

are reported? This is important as this information will provide key data for other researchers to decide which approach to use.

(Reviewer #3, major comment No. 4)

The AIMS system is interesting as it provides single-cell readout of various editing outcomes. Can the authors apply the system to anti-CRISPR and small molecule inhibitors for a fair comparison with their safeguard sgRNA approach?

(Reviewer #1, major comment No. 2)

2. The authors attenuated sgRNA efficiencies by adding [C] extensions to the 5' ends of sgRNAs. Attenuation of sgRNA efficiencies by adding an extension to the 5' end has been published (Perli et al. Science 2016, DOI: 10.1126/science.aag0511), a reference that the authors do not cite.

(REPLY)

According to the reviewer #3's comments, we first compared the cytosine extension method, anti-Cas9 small molecule inhibitor (BRD0539, PMID 31051099), and anti-Cas9 proteins (AcrIIA4 expression plasmid, PMID 30643127). For this purpose, the comparative experiments were conducted using AIMS.

BRD0539 is reported to inhibit SpCas9 in the eGFP-disruption assay with an EC50 of 11.5 mM (PMID 31051099). However, as shown in Fig. R1, the small molecule BRD0539 showed no Cas9 inhibitory effects; but on the contrary, induced cytotoxicity in mESCs at concentrations of 30 mM (Fig. R1).

Fig. R1. Effects of Cas9 inhibitor BRD0539 on Cas9 activity and cell viability in mESC-AIMS (Cdh1-P2A1).

On the other hand, the anti-Cas9 AcrIIA4 protein could inhibit Cas9 activity and was effective to induce mono-allelic indels in AIMS experiments (Fig. R2, top). However, stronger inhibition by the addition of large amount of AcrIIA4-expressing plasmids induced severe cytotoxicity (Fig. R2, bottom). Thus, AcrIIA4 appears to have a narrow window for fine-tuning inhibition of Cas9 activity. In contrast, [C]gRNAs reciprocally inhibited cytotoxicity along with longer [C] and exhibits wide-range tuning (Fig. R3, bottom). These results are consistent with the original results in Fig. 7d, e and Extended Data Fig. 8b,d,f, and h in the revised manuscript.

Therefore, the safeguard [C]sgRNAs are better than AcrIIA4-expressing plasmids for efficient and easy fine-tuning of Cas9 activity without concomitant increase in cytotoxicity.

Fig. R2. Effects of AcrIIA4 on Cas9 activity and cell viability in mESC-AIMS.

Fig. R3. Effects of [C]sgRNA on Cas9 activity and cell viability in mESC-AIMS.

Next, as the reviewer #1 suggested, Perli et al. reported delay of sequential indel induction in the self-targeting gRNA (stgRNAs) with more than 20nt-spacer site (Perli et al. Science 2016, shown in Fig. S7). Although this is suggestive of suppressive effects of longer spacers on Cas9 activity, it has not been established and generalized in depth how to optimize the design of such longer spacer for each target. In fact, they reported a complete loss of Cas9 activities for a stgRNA containing a randomly chosen 30-nt spacer, and computationally designed stgRNAs that were predicted to avoid undesirable secondary structures. In their report, Fig. 1E showed almost the same frequency of indel induction among 30nt-, 40nt- and 70nt-stgRNAs.

Based on the reviewer #1's comments, we have verified the long-spacer gRNAs using AIMS. We observed that Cas9 activity was not decreased by both 30nt- and 40nt-spacer sgRNAs at all, which was shown by targeting two different sites (Fig. R4). Interestingly, the spacer-length-dependent inhibition of Cas9 activity was observed when [10C] and [25C] are added to the spacers. In these conditions, mono-allelic indel clones could be obtained. The results suggest that the longer spacer actually have potential of lowering Cas9 activity, but its effect is limited to certain conditions, for example, a condition with low Cas9 activity. A combination of [C]sgRNAs and long-spacer gRNAs may further extend the applicability of fine-tuning approaches. Our systematic validation provides a strong foundation for the usefulness of the safeguard [C]sgRNAs.

Fig. R4. Combinatorial effects of long spacer-sgRNAs and [C] extension in mESC-AIMS (Cdh1-P2A1).

We would like to thank the reviewer for valuable comments which helped us to make this discovery. These results are now included in Fig. 4 and Extended Data Fig.4 in the revised manuscript. We have also updated the references according to the reviewer #1's comments.

(2) Expansion of biomedical applicability of the [C]sgRNA platform (Editor comment No. 2)

** Evidence of advantageous biomedically relevant applicability of the approach, beyond the alleviation of p53 activation and cytotoxicity in induced pluripotent stem cells.*

(REPLY)

From the standpoint of biomedical applicability, the safeguard [C]sgRNAs can be potentially combined with other CRISPR-related technologies. To investigate this possibility, we have tested whether cytosine extension methods are compatible with CRISPRa/i and Cas12a platforms to finely tune their activities. This plasmid-based system can be easily adopted to AAV- or other virus-based experimental and clinical applications.

We first confirmed fine-tuning effect of [C]sgRNA on CRISPR activator (CRISPRa) and CRISPR interference (CRISPRi) (Fig.R5). This tuning effect is very important for biomedical tool because processes of many complex diseases are associated with a slight change in gene expression, typically induced by single nucleotide polymorphisms. While CRISPRa/i can be modulated by the choice of different sgRNAs, the use of mismatch sgRNA, and anti-Cas9 approaches, our method provides a simple and easy way for large-dynamic-range regulation.

Fig. R5. Effects of [C]sgRNAs on CRISPRa (ASCL1 induction) and CRISPRi (BRCA1 and CXCR4 suppression) platform in HEK293T cells.

According to the reviewer #3's comments (the major comment No. 4), we further examined the utility of the safeguard [C]sgRNAs in Cas12a system. We constructed puromycin-selectable PX459-based all-in-one plasmids that express AsCpf1 (Cas12a) instead of Cas9 and investigated indel induction using AIMS to investigate bi-/mono-editing outcomes. Since the gRNA structure of 5'-PAM-spacer-3' for Cas12a is opposite to that of 5'-spacer-PAM-3' for Cas9, we first addressed to define the position of [C] extension that can efficiently decrease AsCpf1 activity in a length-dependent manner by adding [10C] and [25C] (Fig. R6, left). Interestingly, the activity was length-dependently suppressed when adding [C] on a 5' sgRNA site. On the other hand, adding [C] between hairpin and spacer sites lost Cas12a activity, while 3' [C] addition did not have suppressive effect at all. Thus, we chose the 5' end as the site of [C] addition, and next examined various length of [C] for fine-tuning of AsCpf1 activity (Fig. R6, right). As expected, the activity decreased, and the frequency of mono-allelic indels increased in a length-dependent manner. Taken together, the safeguard [C]sgRNAs can be applied to Cas12a system by extending [C] at 5' end of sgRNA.

This analysis also suggested that AIMS is also useful for assessing the performance of Cas12a system. Of note, AsCpf1 gRNA requiring TTTN PAM sequence could not be set within 66 bp of a P2A₁ site, thereby, we used a P2A₂ site. In this regard, the AIMS is a powerful tool to investigate various types of CRISPR-Cas system by altering P2A sequence with silent mutations that can match various PAM sequences.

Fig. R6. Effects of [C]-extension site (left) and length (right) on Cpf1 platform in mESC-AIMS.

These results are now included in Fig. 4 and Extended Data Fig.4 in the revised manuscript. We further validated the biomedical applicability by testing the usefulness of our approach in other cell lines besides mESCs, as summarized in the next section.

(3) Performance of [C]sgRNA in other cell lines

(Editor comment No. 3)

* Extended evidence of the efficiency of monoallelic and biallelic editing in additional cell lines, as per the questions and suggestions of Reviewer #2.

(Reviewer #2, major comment No. 1)

1. The question is whether this outcome depends on the cell line used. Will cell lines with a reduced transfection efficiency generate lower levels of the effective sgRNA-Cas9 and thus altering the frequency of monoallelic and

biallelic editing? Performing a similar study in human iPSCs or ESCs known to have low gene editing efficiency and slower proliferation rate would address whether the ratio between monoallelic and biallelic gene editing varies, depending on different cell lines.

(REPLY)

According to the reviewer #2's comments, we confirmed that cytosine extension methods work well in other cell lines including human iPSCs, primary human adipose-derived stem cells (ADSCs), and mouse hepatoblasts derived from AIMS mice, which we have newly generated.

First, in the initial manuscript, as for EMX1 and VEGFA1, we compared the differences in editing efficiency (indel probability) between HEK293T and hiPSC (Extended Data Fig.3b,c and Extended Data Fig.8a,c,e,g). The results indicated that editing efficiency decreased in hiPSC more rapidly than HEK293T by [C] extension, which corresponds to the reviewer #2's comment. To further investigate the outcomes of bi-/mono-allelic editing, we conducted clonal analysis in hiPSC (Fig. R7) and human adipose-derived stem cells (hADSCs) (Fig. R8). In both cell lines, suppression of Cas9 activities by cytosine extension was successfully confirmed. In contrast to mESCs, even short extension such as [5C] and [10C] strongly decreased editing efficiency, and mono-allelic editing was efficiently induced. In addition, we have established a method to sequence a pair of alleles from single cells of hADSCs through whole genome amplification (WGA). Especially in the primary cells, this WGA process is necessary because enough genomic DNA for PCR amplification for sequencing cannot be obtained from colonized cells due to loss of proliferative potential under single cell culture for cloning.

Fig. R7. Performance of [C]sgRNA system in ssODN-mediated R206H SNP knock-in for the ACVR1 locus in WT hiPSCs.

Fig. R8. Single-cell analysis of indel patterns for endogenous VEGFA1 targeting in hADSCs.

We further generated AIMS mouse and established hepatoblast (HB) cell line (HB-AIMS) to compare editing outcomes under the same conditions of AIMS-mESC experiment and analysis protocols (Fig. R9). HB was chosen because Cdh1 gene is homogeneously expressed. The HB-AIMS also showed bi- to mono-indel shift, while the Cas9 activity rapidly decreased in HB-AIMS comparing to mESC-AIMS with [C] extension.

Fig. R9. Establishment of a HB-AIMS cell line from Cdh1-P2A1-AIMS embryos at E12.5 (left) and indel analysis using P2A1-sgRNA1 (right).

The data collectively suggest that the role of relationship between Cas9 activity and frequency of bi-/mono-allelic editing is conserved among cell types, while the editing sensitivity to the [C] length varies in them. In addition, this finding is very important for biomedical applications since these cell lines can contribute to regenerative medicine and gene therapy. Regulation of Cas9 activity for selecting bi-/mono-editing and fine-tuning of endogenous gene expression by CRISPRa/i system in these cell lines will increase the number of diseases to be targeted.

These results are now included in Fig. 4 and Extended Data Fig.4 in the revised manuscript.

Section 2: Point by point responses to the reviewers' comments.

Responses to Reviewer #1

>>> *Reviewer #1 (Report for the authors (Required)):*

Kawamata et al. developed methods to achieve intended editing using Cas9 nuclease. First, the authors generated an allele-specific indel monitoring system (AIMS) by adding P2A-tdTomato and P2A-Venus sequences to genes encoding a membrane protein (Chd1) and a nuclear protein (Tbx3). Using these fluorescent reporters, the authors visualized allele-specific genome editing events and found that bi-allelic DNA cleavage was dominant when appropriate sgRNAs were designed. In addition, the authors observed that AIMS was more accurate than the T7E1 assay. Then, the authors attenuated Cas9 activity by adding [C] extensions at the 5'-ends of sgRNA spacer sequences. Next, the authors developed a computational model to maximize mono-allelic editing. Based on these results, the authors generated a heterozygous single-nucleotide polymorphism disease model through optimized mono-allelic editing. The authors also performed computational modeling to maximize the frequency of intended HDR-mediated editing and optimized the method for inducing precise gene corrections in human iPSCs. This study is well-design and carefully conducted. However, the impact and novelty of the current study are limited, especially given that prime editing technology is available to induce precise gene corrections. Thus, the current study based on Cas9 nuclease would be suitable for a more specialized journal.

(REPLY)

We appreciate your time and effort in reviewing our manuscript. Your valuable comments provided significant help in improving our manuscript. We have modified our former version to address your requests. As indicated in the point-by-point responses below, we have taken all of your comments and suggestions into account in the revision.

Major comments 1.

As stated above, the precise genome editing that the authors intended can be induced using prime editing, probably in a more efficient and specific manner. Unless the authors show evidence that their approach is superior to the approach based on prime editors, the impact and implications of the current study, especially with respect to biomedical applications of genome editing, do not reach the level of Nature Biomedical Engineering.

(REPLY)

Thank you very much for important comments. As the reviewer suggested, the prime editing (PE) might be the best editing tool for 1 bp replacement at higher frequency. On the other hand, the PE studies has not clarified allele selectivity of the 1 bp replacement at the single cell levels together with the cocerns of editing errors. Of note, the editing errors, which are composed of alleles with both 1 bp replacement and additional mutations (impure prime edits) and alleles without 1 bp replacement but with other mutations (by-product edits) are very frequent, reaching around 50% of alleles (Petri 2022 Nat.Biotechnol., PMID 33927418). Given the editing error issue, it remains uncertain at this time whether PE will be only the sole solution for a practical gene therapy in the future. In addition, considering CRISPR-Cas9 is already being used in clinical trials by multiple therapiutic companies, we think that it is very important to expand the options of CRISPR-related technoloigoies for wider applications. In fact, our study focuses not only on precise 1 bp replacemenet, but also on modulation of selectivity of bi-/mono-allelic editing, off-target effects, HDR efficiencies, and cytotoxicity through development of the safeguard [C]sgRNAs. In addition, systematic validation of the safeguard [C]sgRNA system is supported by establishment of mathematical simulation models. These improvements could mitigate multiple issues in the current CRISPR-Cas9 platforms and may contribute to the ongoing Cas9-mediated gene therapy. Furthermore, as discussed in Section 1 (2), we

demonstrated that cytosine extension methods are compatible with CRISPRa/i and Cas12 platforms to finely tune their activities, suggesting the wide-range applicability in CRISPR-related technologies. Based on these findings, we believe that the present study is of great significance for biomedical applications of genome editing.

We reorganized the relevant texts in the revised manuscript by adding new Fig.4 and Extended Data Fig.4.

2. The authors attenuated sgRNA efficiencies by adding [C] extensions to the 5' ends of sgRNAs. Attenuation of sgRNA efficiencies by adding an extension to the 5' end has been published (Perli et al. Science 2016, DOI: 10.1126/science.aag0511), a reference that the authors do not cite. In addition, a method for guide RNA activity attenuation was also previously published (Jost et al. Nat. Biotechnol. 2020, https://doi.org/10.1038/s41587-019-0387-5), which is not cited in the current manuscript, either.

(REPLY)

We appreciate the reviewer’s suggestions and comments. We have cited the two works in the revised manuscript (page8, 267; page 8, line 271).

As for the spacer-mismatch method for attenuation of Cas9 activity (Jost et al. Nat. Biotechnol. 2020), although the mismatch introduction is a well-known method to reduce Cas9 activity, it potentially generates new on- and off-targets. In contrast, the [C]sgRNA method do not generate them, but rather suppresses all off-targets through a mechanism that downsizes the Cas9 activity itself.

As discussed in Section 1 (1), we verified the long-spacer gRNAs, which are described by Perli et al. (Perli et al. Science 2016). Perli et al. reported delay of sequential indel induction in the self-targeting gRNA (stgRNAs) with more than 20nt-spacer site (Perli et al. Science 2016, shown in Fig. S7). Although this is suggestive of suppressive effects of longer spacers on Cas9 activity, it has not been established and generalized in depth how to optimize the design of such longer spacer for each target. In fact, they reported a complete loss of Cas9 activities for a stgRNA containing a randomly chosen 30-nt spacer, and computationally designed stgRNAs that were predicted to avoid undesirable secondary structures. In their report, Fig. 1E showed almost the same frequency of indel induction among 30nt-, 40nt- and 70nt-stgRNAs (Fig.1E).

Based on the reviewer’s suggestions, we have experimentally verified the long-spacer gRNAs using AIMS. We observed that Cas9 activity was not decreased by both 30nt- and 40nt-spacer sgRNAs at all, which was shown by targeting two different sites (Fig. R4, redisplayed). Interestingly, the spacer-length-dependent inhibition of Cas9 activity was observed when [10C] and [25C] are added to the spacers. In these conditions, mono-allelic indel clones could be obtained. The results suggest that the longer spacer actually have potential of lowering Cas9 activity, but its effect is limited to certain conditions, for example, a condition with low Cas9 activity. A combination of [C]sgRNAs and long-spacer gRNAs may further extend the applicability of fine-tuning approaches. Our systematic validation provides a strong foundation for the usefulness of the safeguard [C]sgRNAs.

These results are now shown in Extended Data Fig.4 in the revised manuscript.

Fig. R4. Combinatorial effects of long spacer-sgRNAs and [C] extension in mESC-AIMS (Cdh1-P2A1). (Redisplayed)

3. As conducted in Jost et al. Nat. Biotechnol. 2020, the authors could develop a machine learning-based model that predicts sgRNA activity, which would strengthen the manuscript.

(REPLY)

The effects of mismatch introduction into spacer site are highly heterogenous depending on the converted nucleotides (for example, from T to G, C, or A), positions, and target sequences. In such complicated conditions, the machine learning is a powerful approach to establish the prediction by examining huge numbers of mismatch gRNAs with different target sites. On the other hand, our simple [C]sgRNA method can uniformly decrease Cas9 activity in a [C]-length-dependent manner, and this effect was seen for all sgRNAs tested. Based on these pervasive features in the relationships between [C]-length and Cas9 activity, we took principle-oriented, equation-based simulation approach as an alternative way, which could predict various editing outcomes based on systematic but small scale datasets.

We reorganized the relevant texts in the revised manuscript.

Minor comment

1. The authors stated “editing outcomes at each allele are stochastic” based on their results on a handful of sgRNAs. However, testing at thousands of target sequences has shown that editing outcomes are not stochastic (Shen et al. Nature 2018, doi: 10.1038/s41586-018-0686-x; Allen et al., Nat. Biotechnol. 2018, <https://doi.org/10.1038/nbt.4317>). The authors may want to tone down their statement considering these previous studies, perhaps saying “partially stochastic”.

(REPLY)

Thank you very much for careful reading. We apologize to you for causing the misunderstanding. As the reviewer suggested, the sequence outcomes are not stochastic. In the initial manuscript, the meaning of ‘stochastic’ is about choice of a pair of allele (paternal or maternal allele) but not about indel sequence.

We have corrected the sentence in the text (page 4, line 137).

Responses to Reviewer #2

>>> Reviewer #2 (Report for the authors (Required)):

Gene editing with CRISPR-Cas9 is widely used to alter genome sequence for gene function study and clinical application. When expressed in high levels, mainly through plasmid-based expression systems, sgRNA-Cas9 frequently induces biallelic editing. To induce monoallelic editing, the level of sgRNA-Cas9 will need to be carefully modulated with different approaches. However, these approaches require optimization to balance between mono vs. biallelic editing. How to modulate the intracellular level of sgRNA-Cas9 is therefore important to determine whether a desirable outcome of gene editing experiments can be achieved. The authors in this manuscript use an allele-specific indel monitor system (AIMS) to analyze editing patterns of a pair of alleles in a large number of clones without sequencing analysis. Using AIMS, they discover that adding cytosine stretches to the 5'-end of a sgRNA reduces the formation of the effective sgRNA-Cas9 complex, which affects the frequency of monoallelic and biallelic editing events. By controlling the level of effective sgRNA-Cas9 complexes via the C stretch, the authors are able to demonstrate improved efficiency of getting clones with gene knockin/wt and cassette replacement/wt. They also generate a SNP/wt clone that recapitulates a disease model in mice. Finally, the authors display the advantage of using the C stretch in a gene correction study with human patient-derived iPSCs. They also show that p53 activation by DNA damage responses is prevented with the C stretch, reducing cell toxicities caused by excessive Cas9-sgRNA-induced DNA double-strand breaks. Although the length of the cytosine stretch needs to be optimized with different sgRNAs, the authors argue that their approach likely represents an easier and more efficient strategy to achieve the desirable outcome of gene editing relative to other approaches trying to modulate the intracellular level of sgRNA-Cas9.

The authors have designed and performed extensive experiments to support their hypothesis and most of the conclusions can be supported by the data. The finding that extended C residues in sgRNA can modulate the sgRNA-Cas9 activity represents a novel and important advancement in the field of gene editing. **If proven to be generally applicable in more than just mESCs, this finding will have an impact on the current state of the art in gene editing.**

Issues need to be addressed to clarify some of their claims.

(REPLY)

Thank you very much for your helpful comments and positive evaluation on our manuscript. These comments are very helpful to improve our manuscript. We agreed with your suggestions, performed a series of experiments and re-analyses, and revised our manuscript following your comments. Especially, validation in other cell lines is summarized in Section 1 (3). Please find our point-by-point response below.

Major comments 1.

1. Most of the experiment were carried out in mESCs. The frequency of biallelic editing is surprisingly high as shown in Fig. 1e and 1f. Even with 100 fold less DNA at 2.5 ng, the efficiency of biallelic editing remains unchanged (Fig. 2a). This is somewhat surprising as such a drastic reduction in the amount of the transfected DNA is expected to have an impact on the frequency of the monoallelic and biallelic gene editing event. Can the authors explain this? The question is whether this outcome depends on the cell line used. Will cell lines with a reduced transfection efficiency generate lower levels of the effective sgRNA-Cas9 and thus altering the frequency of monoallelic and biallelic editing? Performing a similar study in human iPSCs or ESCs known to have low gene editing efficiency and slower proliferation rate would address whether the ratio between monoallelic and biallelic gene editing varies, depending on different cell lines.

Similarly, using the preformed sgRNA-Cas9 RNP complex instead of the expression plasmid should significantly reduce the gene editing efficiency and alter the ratio between monoallelic and biallelic editing. The authors should discuss the advantage of the current finding over the RNP strategy for sgRNA-Cas9 modulation.

(REPLY)

We appreciate the reviewer’s suggestions and comments.

First, as the reviewer suggested, the frequency of biallelic editing is high even with 100 fold less DNA at 2.5 ng (Fig. 2a in the revised manuscript). This can be explained by the condition of puromycin selection, which selected only the cells with enough expression levels of Cas9 and sgRNA. When using the standard Cas9 system, sufficiently expressed Cas9 and sgRNA, which are selected by puromycin, resulted in bi-allelic indels in most cells even with lower DNA dose. In fact, the plasmid dose-dependent decrease in the number of colony results from the removal of non-transfected cells or cells with low expression levels of sgRNA and Cas9 by low transfection efficiency (Fig. 2a, left). We modified the relevant texts in the revised manuscript.

Second, we validated cytosine extension methods in other cell lines including human iPSCs, primary adipose-derived MSCs (hADSCs), and mouse hepatoblasts derived from AIMS mice, which we have newly generated. This point is also summarized in Section 1 (2).

In the initial manuscript, as for EMX1 and VEGFA1, we compared the differences in editing efficiency (indel probability) between HEK293T and hiPSC (Extended Data Fig.3b,c and Extended Data Fig.8a,c,e,g). The results indicated that editing efficiency decreased in hiPSC more rapidly than HEK293T by [C] extension. To further investigate the outcomes of bi-/mono-allelic editing, we conducted clonal analysis of a pair of alleles in hiPSC (Fig. R7) and human adipose-derived stem cells (hADSCs) (Fig. R8) by sanger sequencing (Fig. R7). In contrast to mESCs, even short extension such as [5C] and [10C] strongly decreased editing efficiency, and mono-allelic editing was efficiently induced. In addition, we have established a method to sequence a pair of alleles from single cells of hADSCs through whole genome amplification (WGA). Especially in the primary cells, this WGA process is necessary because enough genomic DNA for PCR amplification for sequencing cannot be obtained from colonized cells due to loss of proliferative potential under single cell culture for cloning.

Fig. R7. Performance of [C]sgRNA system in ssODN-mediated R206H SNP knock-in for the ACVR1 locus in WT hiPSCs. (Redisplayed)

Fig. R8. Single-cell analysis of indel patterns for endogenous VEGFA1 targeting in hADSCs. (Redisplayed)

We further generated AIMS mouse and established hepatoblast (HB) cell line (HB-AIMS) to compare editing outcomes under the same conditions of AIMS-mESC experiment and analysis protocols (Fig. R9). HB was chosen because Cdh1 gene is homogeneously expressed. The HB-AIMS also showed bi- to mono-indel shift, while the Cas9 activity rapidly decreased in HB-AIMS comparing to mESC-AIMS with [C] extension.

Fig. R9. Establishment of a HB-AIMS cell line from Cdh1-P2A1-AIMS embryos at E12.5 (left) and indel analysis using P2A1-sgRNA1 (right). (Redisplayed)

The data collectively suggest that the role of relationship between Cas9 activity and frequency of bi-/mono-allelic editing is conserved among cell types, while the editing sensitivity to the [C] length varies in them.

These results are now included in Fig. 4 and Extended Data Fig.4 in the revised manuscript.

Third, as for RNP strategy, RNP strategy might be useful to induce mono-allelic indels, but reduction of RNP amount should lead to increase in non-transfected cells. Moreover, cloning efficiency for edit cells would become very low if the transfection efficiency decreases. Therefore, setting the best concentration of RNP in each target and estimating the picked clone number for obtaining edited ones would be a very laborious task. From this standpoint, the plasmid (puromycin)-based method is easy to obtain the desired clones and estimate the picked clone number for obtaining edited ones.

We added the relevant texts in the revised manuscript (page 16, line, 519).

2. The way of deriving AIMS(P) and adjusted AIMS(P) in Fig. 1h and 1j needs to be described in more details as it is not obvious from the Methods section how the authors derive the frequency. It is not clear whether these so-called rare tdTomato+/Venusindel and tdTomatoindel/Venus+ heterozygous clones are grouped into either monoallelic or biallelic clones shown in Fig. 1e.

(REPLY)

Thank you very much for careful reading. Raw AIMS(P) is simply based on fluorescence patterns. Thus, in Fig. 1e, rare tdTomato+/Venusindel and tdTomatoindel/Venus+ heterozygous clones are grouped into **monoallelic** clones. When sequencing these clones, most (86%) of these ostensibly heterozygous clones were turned out to be homozygous. Adjusted AIMS(P) incorporates the sequence analysis of ostensibly heterozygous clones together with fluorescence patterns. In most analyses, we used raw AIMS(P).

We have updated the method section in the revised manuscript.

3. The point of extended Fig. 3b and 3c (line 164) is not clear. The effect of extended Cs on gene editing is clear in both HEK293T and hiPSC. However, as the two cell lines have very different transfection efficiencies, comparing indel probability between them as the figure title suggests is not appropriate.

(REPLY)

While the transfection efficiencies are different in HEK293T cells and hiPSCs, we only analyzed the cells selected with puromycin treatment in Fig.3b. and 3c.

We modified the relevant texts and Figure legends of Extended Fig. 3b. and 3c.

4. One issue with the experiments in the manuscript is the frequent missing of the cell line the authors use. For example, the cell lines used in experiments Fig. 4g and Fig. 5 are not specified. The authors need to provide this information clearly either in the text or in the figure legends.

(REPLY)

We appreciate the reviewer's important suggestions. We have included the description of cell lines used in each of the figure legends.

5. The claim (line 348) that "[C] extension generally recovered the HDR rate which had presumably been suppressed by the conventional CRISPR-Cas9 system" needs further clarification. Will the authors provide references or data to support such a claim?

(REPLY)

A previous study reported that p53 activation by CRISPR-Cas9 inhibited HDR frequency by 19-fold in hiPSCs (Ref. 7). Artificial inhibition of p53, using p53 siRNA, p53 dominant negative forms, and p53 antagonist MDM2, can recover cell viability and HDR frequency in Cas9-based genome editing experiments (REF 6,7, 46).

We updated the relevant texts in the revised manuscript (page 14, line, 454; page 16, line 528). Thank you very much for important comments.

6. Both Fig. 6 and Fig. 7 show that the Cas9 activity would make single nucleotide mismatch less sensitive to re-cleavage. However, it is unclear whether this is dependent on the position of the mismatch nucleotide (1mm) in the sgRNA. If the mismatch occurs at a different position, will the difference between pf and 1mm disappear? Can the authors comment on this? It is also useful if the authors can comment on the effect of incorporating PAM mutations as described in reference 22 into their donor template on re-cleavage of the HDR allele in their system.

(REPLY)

Thank you very much for important suggestions. The relative differences of Cas9 activities between perfect match and 1mm targets depend on "the ratio of K for the 1mm target to K for the perfect match target" (m in Fig. 6g and 6h, please compare red and blue in these figures). Since the relative affinity for the 1mm target depends on the position of the mismatch nucleotide and other contexts including the converted nucleotides (for example, from T to G, C, or A) and target sequences, therefore, the relative differences of Cas9 activities between perfect match and 1mm targets depend on the position of the mismatch nucleotide. However, in most cases where mismatch causes substantial decrease in binding affinity of [OC] sgRNA, decreasing Cas9 activities by the [C] sgRNAs decreases and increases the ratio of off-target editing to on-target editing and the on-target specificity, respectively (Fig. 6h). In fact, even in case where we could not observe the clear differences between perfect match and 1mm for [OC] sgRNA (Fig. 6c), the differences between perfect match and 1mm became clearer along with editing frequency suppression. In addition, when mutating each nucleotide position of sgRNA (P2A1-sgRNA1), we observed [C] length-dependent off-target repression at most positions (Fig. R10 (Reviewer only, data not

shown). Together, we think in theory that 1mm off-target with strong affinity can be effectively suppressed by the use of lower Cas9 activity with long-[C]sgRNA.

Fig. R10. (Reviewer only). Relationships between the position of the mismatch nucleotide (1mm, x-axis) and [C] extension-mediated off-target repression in mESC-AIMS (Tbx1-P2A1). pf, perfect match. 1-20, 1mm position from PAM.

In CORRECT platform in Ref. 22, the donor template should be protected from re-cleavage through PAM mutations. This makes a limitation on HDR design, where the replacement position should be set in the GG of PAM site. Considering the major situation where the target 1bp-replacement is set in the spacer site, extra replacement needs to be introduced in the PAM site to block re-cleavage as reported in the ref. 22. However, this may cause amino acid change, and also alterations in translation efficiency of the targeted gene, even with silent mutation. If the re-cleavage frequency of the donor templates (i.e., mm targets) can be lowered using [C] sgRNA, HDR design could be much easier to design and achieve precise editing.

We updated the relevant method sections and included a part of this discussion in the revised manuscript (page 17, line, 555).

7. In line 384, the authors explain “We observed a sharp decline in the editing efficiency of hiPSCs, compared with mESCs (Fig. 7c), which may be partly explained by our selection of hiPSCs with weak p53 activation.” Why does the use of hiPSCs with weak p53 activation reduce gene editing efficiency in hiPSCs? Activation of p53 counteracts gene editing. Weak p53 activation should therefore improve gene editing instead of reducing it.

(REPLY)

Since the sensitivity of iPSCs to p53 activation is much higher than mESCs, we think that our selection methods enriched hiPSCs with weaker p53 activation, where DSB-indels cause at lower frequency, compared to a number of removed cells in which p53 is highly activated by frequent DSB-indels. According to the reviewer’s comments, we have modified the relevant texts as follows:

We observed a [C] extension-mediated sharp decline in the editing efficiency of hiPSCs, compared with mESCs (Fig. 7c), which may be partly explained by higher sensitivity of hiPSCs to p53 activation and selection of hiPSCs with non-successful editing and weaker p53 activation.

*8. A couple of mistakes need to be corrected;
line 448, Fig. 3h. There is no Fig. 3h;
line 462, Fig. 5g, h. There are no Fig. 5g and h.*

(REPLY)

Thank you very much for careful reading. We have corrected the relevant parts as follows:

Line 526 in the revised manuscript, Fig. 3h -> Fig. 3f

Line 540 in the revised manuscript, Fig. 5g, h -> Fig. 6g, h

Responses to Reviewer #3

>>> Reviewer #3 (Report for the authors (Required)):

The manuscript reports a method to reduce the editing efficiency and increase monoallelic editing using a modified Cas9 guide RNA design. The authors first developed an allele-specific indel monitor system (AIMS) to visualize on the single cell level the outcomes of editing: no indel, in frame indel, frameshift indels/large deletions. They used the system to show appending C's to the 5' end of sgRNA can reduce the biallelic editing and increase monoallelic editing. They applied the method to generate monoallelic single-nucleotide polymorphism (SNP) gene editing in mESC and hiPSC to create a fibrodysplasia ossificans progressiva (FOP) mutation.

Modifying the 5' end of sgRNA to reduce its editing efficiency is not totally surprising, as previous studies have observed similar effects (PMID: 32496535, 28985763, 30988504). The nice part of the work is its detailed characterization for monoallelic editing using the modified sgRNA, which is a methodological advance. The finding may provide a simple and useful strategy for the gene editing and gene therapy field. But it currently misses a key comparison with alternative approaches. The manuscript has several interesting components, but the overall organization of the story should be improved to increase its impact in the field.

(REPLY)

Thank you very much for your helpful comments and positive evaluation on our manuscript. These comments are very helpful to improve our manuscript. We agreed with your suggestions, performed a series of experiments and re-analyses, and revised our manuscript following your comments. Especially, comparison with alternative approaches is summarized in Section 1 (1). Please find our point-by-point response below.

Major comments:

1. Comparison with other technologies. As a technology, it is important to know how this method compares to alternative methods and what the advantages are. Is this C-extension method superior compared to previous methods including anti-Cas9 protein and small molecule inhibitors? Have the authors performed comparative studies to prove the superiority of the method given anti-Cas9 (PMID: 30643127) or small molecules (31051099) are reported? This is important as this information will provide key data for other researchers to decide which approach to use.

(REPLY)

We appreciate the reviewer's suggestions and comments. This point is also summarized in Section 1 (1). We compared the cytosine extension method, anti-Cas9 small molecule inhibitor (BRD0539, PMID 31051099), and anti-Cas9 proteins (AcrIIA4 expression plasmid, PMID 30643127). For this purpose, the comparative experiments were conducted using AIMS.

BRD0539 is reported to inhibit SpCas9 in the eGFP-disruption assay with an EC50 of 11.5 mM (PMID 31051099). However, as shown in Fig. R1, the small molecule BRD0539 showed no Cas9 inhibitory effects; but on the contrary, induced cytotoxicity in mESCs at concentrations of 30 mM (Fig. R1).

Fig. R1. Effects of Cas9 inhibitor BRD0539 on Cas9 activity and cell viability in mESC-AIMS (Cdh1-P2A1). (Redisplayed)

On the other hand, the anti-Cas9 AcrIIA4 protein could inhibit Cas9 activity and was effective to induce mono-allelic indels in AIMS experiments (Fig. R2, top). However, stronger inhibition by the addition of large amount of AcrIIA4-expressing plasmids induced severe cytotoxicity (Fig. R2, bottom). Thus, AcrIIA4 appears to have a narrow window for fine-tuning inhibition of Cas9 activity. In contrast, [C]gRNAs reciprocally inhibited cytotoxicity with longer [C] and exhibits wide-range tuning (Fig. R3, bottom). These results are consistent with the original results in Fig. 7d, e and Extended Data Fig. 8b,d,f, and h in the revised manuscript.

Therefore, the safeguard [C]sgRNAs are better than AcrIIA4-expressing plasmids for efficient and easy fine-tuning of Cas9 activity without concomitant increase in cytotoxicity.

Fig. R2. Effects of AcrIIA4 on Cas9 activity and cell viability in mESC-AIMS. (Redisplayed)

Fig. R3. Effects of [C]sgRNA on Cas9 activity and cell viability in mESC-AIMS. (Redisplayed)

These results are included in Fig. 4 and Extended Data Fig.4 in the revised manuscript. We also included a part of this discussion in the revised manuscript (page 16, line, 519).

2. Rationale for the technology. The authors need to provide better rationale at the beginning of the story for why they need to limit editing efficiency in the abstract and the introduction. The authors wrote in the abstract, "Controlled inhibition of CRISPR-Cas9 is important for precise and safe genome editing". I am not sure if this statement is totally accurate. The gene editing field desires technologies with high editing efficiency and less toxicity and less off-targets. Methods that can achieve high precision and safety while sacrificing efficiency may or may not be chosen. The question is whether their safeguard sgRNA method is better in terms of this tradeoff compared to alternative methods.

(REPLY)

Thank you very much for important suggestions. Although several approaches have been developed, "high editing efficiency" frequently accompanies with "high toxicity and high off-targets".

The fundamental mechanism of Cas9 action involves targeted induction of mutagenic DNA lesions, leading to the possibility of off-target editing, genotoxicity, translocations, and malignancy. Therefore, limiting genome editing action only to the desired target sequence, time duration, and spatial location is desirable. In such context, "Controlled inhibition of CRISPR-Cas9 is important for precise and safe genome editing", and anti-Cas9 proteins and small molecular inhibitors have been proposed as described in the introduction section.

In the trade-off relationships between “editing efficiency” and “toxicity and less off-targets”, the degree of suppression of “editing efficiency” matters and the importance of the degree depends on the application context, for example, bi-allelic vs. mono-allelic editing.

In both cases of engineered Cas9 (eCas9 and Cas9-HF1) and modified sgRNAs (truncated sgRNAs, hairpin sgRNAs, and sgRNAs with a couple of guanine addition to the 5'-end), high specificity is associated with tolerable decrease in on-target activities (discussed in the discussion section). We provide the first description of equation-based explanation of this trade-off relationship between on-target and off-target activities, as shown in Fig. 6g, 6h, and method section. In our system, short cytosine extension corresponds to this scenario: short cytosine extension reduces cytotoxicity and enhances homology-directed repair, while maintaining bi-allelic editing capacity.

In the setting of mono-allelic editing, the suppression of “editing efficiency” should be further potentiated. In such context, long extension further decreases on-target activity and facilitates mono-allelic and precise editing. Therefore, the key point of “safeguard sgRNA” method is an easily tunable platform for diverse applications, including safe bi-allelic editing, mono-allelic editing, and HDR-based generation and correction of disease-associated single nucleotide substitutions free from p53 activation.

According to the reviewer’s suggestions, we have updated the relevant description in the abstract.

3. Application. The authors demonstrated using their method to generate fibrodysplasia ossificans progressiva (FOP) monoallelic mutation in stem cells. Is there an application that can be uniquely addressed by their approach? For example, can the use of the safeguard sgRNA increase gene editing specificity? In Kocak Nat Biotech 2019 work where they introduced a hairpin structure to the 5' end of sgRNA, they observed greatly increased editing specificity. Can the authors demonstrate such a nice feature for gene editing that is important to the applications?

(REPLY)

In FOP mutation experiments, our method enable one-step generation of disease-associated single-nucleotide substitutions, in contrast to the necessity of multiple complicated steps for precise editing (Ref 22). We demonstrate that the safeguard sgRNA increase gene editing specificity (Figure 6) as well as hairpin sgRNAs described by Kocak et al (Nat Biotech 2019). In addition, we provide the first description of equation-based explanation of the trade-off relationship between on-target and off-target activities, as shown in Fig. 6g, 6h, and method section. In contrast to hairpin sgRNAs, our system can continuously tune the Cas9 activity by modulation [C] length. In the present study, the integration of a fluorescence-based allele-specific indel monitor system (AIMS), computational simulation, and systematic validation established distinct optimal windows of Cas9 activity for diverse applications, as summarized in Fig. 7k. As summarized in Section 1 (2), we further demonstrated that cytosine extension methods are compatible with CRISPRa/i and Cas12a platforms to finely tune their activities, suggesting the wide-range applicability in CRISPR-related technologies.

We have updated the relevant description in the revised manuscript.

4. The AIMS system is interesting as it provides single-cell readout of various editing outcomes. However, it also requires precisely inserting the dual-color reporters into cells and selecting single clones for testing sgRNAs. Can the authors comment on the broader utility of the AIMS system to study gene editing? Can the authors apply the system to anti-CRISPR and small molecule inhibitors for a fair comparison with their safeguard sgRNA approach? Can the authors apply AIMS to other Cas molecules such as Cas12a? It has been reported that Cas12a may have different monoallelic vs. biallelic editing activities from Cas9. Performing such experiments can enhance the significance of the story.

(REPLY)

We appreciate the reviewer’s suggestions and comments. The comparison with anti-Cas9 small molecular inhibitors and proteins are shown in Section 1 (1) and the reponse to the reviewer #3’s major comment No. 1.

We appreciate that you are interested in the AIMS system and its broader utility. In order to expand the utility of AIMS for many researchers who can also investigate editing outcomes in various cell types both in vitro and in vivo, we have generated AIMS mice. In the revised manuscript, we have presented data from a primary hepatoblast cell line that homogenously expresses Cdh1 gene as a representative example of its utility for examing editing outcomes in a similar manner to mESC-AIMS (summarized in Section 1 (3)). Thus, in the future, the AIMS mouse will be used to investigate editing outcomes of other primary tissue and tumor cell lines, and in cells of tissues directly by in vivo editing. We included a part of this discussion in the revised manuscript.

As summarized in Section 1 (2), we examined the utility of the safeguard [C]sgRNAs in Cas12a system. We constructed puromycin-selectable PX459-based all-in-one plasmids that express AsCpf1 (Cas12a) instead of Cas9 and investigated indel induction using AIMS to investigate bi-/mono-editing outcomes. Since the gRNA structure of 5’-PAM-spacer-3’ for Cas12a is opposite to that of 5’-spacer-PAM-3’ for Cas9, we first addressed to define the position of [C] extension that can efficiently decrease AsCpf1 activity in a length-dependent manner by adding [10C] and [25C] (Fig. R6, left). Interestingly, the activity was length-dependently suppressed when adding [C] on a 5’ sgRNA site. On the other hand, adding [C] between hairpin and spacer sites lost Cas12a activity, while 3’ [C] addition did not have suppressive effect at all. Thus, we chose the 5’ end as the site of [C] addition, and next examined various length of [C] for fine-tuning of AsCpf1 activity (Fig. R6, right). As expected, the activity decreased, and the frequency of mono-allelic indels increased in a length-dependent manner. Taken together, the safeguard [C]sgRNAs can be applied to Cas12a system by extending [C] at 5’end of sgRNA.

This analysis also suggested that AIMS is also useful for assessing the performance of Cas12a system. Of note, AsCpf1 gRNA requiring TTTN PAM sequence could not be set within 66 bp of a P2A₁ site, thereby, we used a P2A₂ site. In this regard, the AIMS is a powerful tool to investigate various types of CRISPR-Cas system by altering P2A sequence with silent mutations that can match various PAM sequences.

These results are included in Fig. 4 and Extended Data Fig.4 in the revised manuscript. We also included a part of this explanation in the revised manuscript (page 8, line, 243; page 16, line, 519).

Fig. R6. Effects of [C]-extension site (left) and length (right) on Cpf1 platform in mESC-AIMS. (Redisplayed)

5. How are off-targets in Fig. 6 measured? Have the authors performed standard assays such as GUIDE-seq to quantify the off-target effects?

(REPLY)

In Fig. 6, the on-target and off-target activities are measured by T7E1-based Bac[P] assays. We have updated the figure legends.

6. Also in Fig. 6, it is confusing to see that [0C]sgRNA induced HDR in only 4.1% of clones, much less than [5C]sgRNA which is 20.5%. This is inconsistent with their conclusions in Fig. 2g. How do authors explain this observation?

(REPLY)

In Fig. 6d, [0C]sgRNA induced overall HDR in only 4.1% of clones. However, the frequency of overall HDR for [5C]sgRNA increased to 20.5%. This is due to suppression of HDR rates by [0C] sgRNAs (Fig. 6f in the revised manuscript).

Minor comments:

7. Some figures including Fig. 3 and some computational modeling figures can be moved to Extended Data figures. The text describing these biochemical assays and detailed computation modeling can be moved to supplemental text. This can increase overall readability of the story, helping the readers to focus on the main findings.

(REPLY)

We have moved Figure 5 in the initial manuscript to Extended Data Figure. On the other hand, we have included new datasets related to Section 1 (1), (2), and (3) in Figure 4. Since the computational modeling is a vital part of our study, we further reorganized the manuscript to improve the overall readability.

Rebuttal 2

Summary of the revision (NBME-21-2632A):

“Rational optimization of versatile genome editing applicability by wide-range programmable Cas9 inhibition with safeguard sgRNAs”

We thank reviewers for evaluating our manuscript. Please find point-by-point response to reviewers’ criticisms below.

Point by point responses to the reviewers’ comments.

Responses to Reviewer #1

*>>> Reviewer #1 (Report for the authors (Required)):
I think that the revised manuscript is now suitable for publication.*

(REPLY)

We appreciate your time and effort in reviewing our manuscript. Thank you very much for positive evaluation.

Responses to Reviewer #2

*>>> Reviewer #2 (Report for the authors (Required)):
After reviewing the response from the authors, the previous comments for the manuscript have been addressed satisfactorily. No additional issues from this reviewer.*

(REPLY)

We appreciate your time and effort in reviewing our manuscript. Thank you very much for positive evaluation.

Responses to Reviewer #3

*>>> Reviewer #3 (Report for the authors (Required)):
The authors have addressed some concerns. Thanks to their efforts. While new experiments and further computational analysis answered or clarified some questions, there are major remaining issues that need to be addressed.*

(REPLY)

Thank you very much for your helpful comments and positive evaluation on our manuscript. These comments are very helpful to improve our manuscript. We agreed with your suggestions, performed additional experiments, and revised our manuscript following your comments. Please find our point-by-point response below.

•A general comment: it is much easier to reduce the activity of gene editing than to improve its activity. It is easy to conceive that there are many possible ways to reduce the activity of a sgRNA, such as by removing a few nucleotides, mutating a few nucleotides in the scaffold, or appending other sequences that can disrupt its structure. The strategy presented here is just one of numerous options to reduce the sgRNA activity.

(REPLY)

We appreciate the reviewer’s comments. As the reviewer suggested, some modifications, including addition of a few guanines at the 5’-end of sgRNAs (Mullally et al. NAR. 2020. PMID: 32496535), hairpin sgRNAs (Kocak

et al. Nat Biotechnol. 2019. PMID: 30988504), and modification of constant regions (Jost et al. Nat Biotechnol. 2020. PMID: 31932729), are reported to reduced the activity of a sgRNA.

On the other hand, tunability of these approaches have not been established well. In fact, in the report by Jost et al. (Nat Biotechnol. 2020. PMID: 31932729), the authors found that modified constant regions of sgRNAs can be used to titrate gene expression (activity of CRISPRi), but finally concluded that the activity of a given constant region and targeting sequence pair is difficult to predict, due to high heterogeneity of the effects of constant-region variants across different target sequences. Thus, they focused on sgRNAs with mismatches in the targeting region. In addition, while hairpin sgRNAs with different hairpin sizes appeared to yield regulated suppression (Kocak et al. Nat Biotechnol. 2019. PMID: 30988504), this analysis was only performed for a limited number of target sequences. In such context, our approach adds a novel option to reduce the sgRNA activity, which is more predictable and universal for diverse sgRNAs.

We included a part of this explanation in Discussion section in the revised manuscript (page 17, line 557).

•Regarding their response to #1, comparison with C-sgRNA with Acr and small molecules, two things to note here. 1) Comparing the viability of having more C's on the sgRNA to the viability of transfecting more Acr plasmid is not a fair comparison. It is unfair to claim that Acr is more toxic, as it could be that there is naked DNA toxicity from increasing Acr plasmid while the C-extension guide plasmid mass remains the same across all experiments. The toxicity is likely due to naked DNA not the Acr. A fairer comparison would be to create an analogous plasmid to the extended sgRNA using a suite of promoters (e.g. similar to Figure 1d, [promoter] – Acr – BFP – CBh – SpCas9 – T2A – puro), transfecting in the DNA, and seeing how Acr expression (as a function of BFP) correlates to viability and mono-allelic efficacy.

(REPLY)

Thank you very much for important suggestions. As the reviewer suggested, the effects of Acr plasmid coexpression in our previous experiments could be compromised by DNA toxicity. According to the reviewer's comments, we re-performed similar experiments by holding the total amounts of plasmids constant. In a new experiment shown in Fig. R1, the toxicity of Acr was mitigated, but still observed for higher doses of Acr plasmids. Similar cytotoxicity profiles were obtained in the absence of the Cdh1-P2A1-sgRNA1 target sequence in WT mESCs (data not shown). Thus, the toxicity of Acr in the previous revision resulted from DNA toxicity and Acr effect. We sincerely appreciate important comments from the reviewer. Thus, both Acr and C-extension can reduce Cas9 activities in a regulated manner, although the former may be associated with some toxicity especially for higher doses. Based on these findings, we included new results in Fig. 4c and updated method section in the revised manuscript and overall toned down the description about Acr.

Fig. R1. Effects of AcrIIA4 on Cas9 activity and cell viability in mESC-AIMS. The total amount of 3 plasmids (550 ng) are constant.

2) Acr seems to be just as good if the goal is to get mono-allelic (see the 25ng and 75ng Acr case compared to 20, 30, and 35 extension). The manuscript is heavy on 'tunability', so it is still unclear why C-extension guide strategy is better than Acr. It is likely that C extension is easier to implement, but Acr gives more synthetic biology options.

(REPLY)

We agree with the reviewer's suggestions. A new experiment suggested that both Acr and C-extension can reduce Cas9 activities in a regulated manner, while the former may be associated with some toxicity especially for higher doses. We overall toned down the description about Acr, and added a discussion about the use of Acr for synthetic biology options (Nakamura et al. Nat Commun. 2019. PMID: 30643127) in the revised manuscript (page 17, line 563).

•Regarding their response to #2: the authors changed the abstract to "precise regulation" instead of "controlled inhibition". They should explain better what 'precise regulation' means in the Discussion. In general, the paper is very dense on method details and stats but does not give enough context on why it is important.

(REPLY)

We appreciate the reviewer's suggestions and comments. "Precise regulation" means predictable and programmable inhibition of "Cas9 activities" by C-extension approach (not "precise editing"). According to the reviewer's suggestion, we have updated the relevant parts in Abstract and Discussion section (page 1, line 21; page 17, line 562).

•Regarding their response to #6: I don't understand they answered the question. It doesn't seem to explain why [0C] sgRNA has a low efficiency.

(REPLY)

Thank you very much for important suggestions. As the reviewer suggested, while [0C] and [5C] sgRNAs have the same on-target and off-target activities in Fig. 6c ($P = 1$), the frequency of overall HDR is clearly different between [0C] and [5C] sgRNAs. This can be explained by ostensible saturation of detected activities of [0C] and [5C] sgRNAs in Bac[P] assays in Fig. 6c. Even if [0C] and [5C] sgRNAs have the maximal on-target and off-target activities in Bac[P] assays (Fig. 6c), within the cells, the temporal frequency of DNA cleavage events across the genome should be substantially lower for [5C]sgRNAs. This should enhance HDR rates. These results are consistent with the results in human iPS cells (Fig. 7). According to the reviewer's suggestion, we have updated the relevant texts (page 17, line 542).

•For their off-target detection approach (comment #3, #5): T7E1-based method is less comprehensive as compared to GUIDE-seq for revealing a full profile of off-targets.

(REPLY)

We appreciate the reviewer's suggestions and comments. Due to the limitations in time and research resource, we expanded our off-target analysis by testing multiple other well-established and representative off-target loci of VEGFA1 sgRNA. In Fig. R2 (see the red box), we confirmed that off-target effects of even [5C] sgRNAs sharply dropped off for all of newly tested off-target loci, thereby reinforcing our results.

These results are now shown in Extended Data Fig. 7b in the revised manuscript. We also updated the relevant texts.

Fig. R2. Off-target analysis by T7E1 assay in HEK293T. The red square indicates additional data showing almost loss of off-targets by [5C]sgRNA.

•The paper is dense and very hard to read. Much of it includes models that don't help genome engineers to apply what they are learning. More empirical and applications-based evidence would have been better than dense probability modeling.

(REPLY)

We apologize for the inconvenience of tough reading. On the other hand, as one of the practical genome engineers, the models presented here is important for us to set up many high-grade genome engineering experiments and help other engineers. We appreciate that you understand the necessity of the models to generalize our experimental results.